# A neurodevelopmental epigenetic programme mediated by SMARCD3–DAB1–Reelin signalling is hijacked to promote medulloblastoma metastasis

Han Zou [1,2,3,4,5], Bradley Poore[4,5], Emily E. Brown[6], Jieqi Qian[5], Bin Xie[7], Evridiki Asimakidou[4,5], Vladislav Razskazovskiy[4,5], Deanna Ayrapetian[4,5], Vaibhav Sharma[4,5], Shunjin Xia[2], Fei Liu[8], Apeng Chen[4,5], Yongchang Guan[4,5], Zhengwei Li[4,5], Siyi Wanggou[2], Olivier Saulnier [9], Michelle Ly[9], Wendy Fellows-Mayle[4], Guifa Xi [10], Tadanori Tomita[10], Adam C. Resnick[11], Stephen C. Mack [12], Eric H. Raabe[13], Charles G. Eberhart[14], Dandan Sun[15], Beth E. Stronach [16], Sameer Agnihotri[4,5,17], Gary Kohanbash[4,5,17], Songjian Lu[18], Karl Herrup[19], Jeremy N. Rich[15,17], George K. Gittes[5,20,21], Alberto Broniscer[5,17,21], Zhongliang Hu[7], Xuejun Li[2,3], Ian F. Pollack[4,5,17], Robert M. Friedlander [4], Sarah J. Hainer [6,17] ✉, Michael D. Taylor [9] ✉ & Baoli Hu [4,5,17,21] ✉

How abnormal neurodevelopment relates to the tumour aggressiveness of medulloblastoma (MB), the most common type of embryonal tumour, remains elusive. Here we uncover a neurodevelopmental epigenomic programme that is hijacked to induce MB metastatic dissemination. Unsupervised analyses of integrated publicly available datasets with our newly generated data reveal that SMARCD3 (also known as BAF60C) regulates Disabled 1 (DAB1)-mediated Reelin signalling in Purkinje cell migration and MB metastasis by orchestrating *cis*-regulatory elements at the *DAB1* locus. We further identify that a core set of transcription factors, enhancer of zeste homologue 2 (EZH2) and nuclear factor I X (NFIX), coordinates with the *cis*-regulatory elements at the *SMARCD3* locus to form a chromatin hub to control SMARCD3 expression in the developing cerebellum and in metastatic MB. Increased SMARCD3 expression activates Reelin–DAB1-mediated Src kinase signalling, which results in a MB response to Src inhibition. These data deepen our understanding of how neurodevelopmental programming influences disease progression and provide a potential therapeutic option for patients with MB.

Organism development is precisely orchestrated in time and space, during which dysregulation of biological factors may influence diseases such as medulloblastoma (MB). MB is the most common type of embryonal tumour arising in the cerebellum, and it causes a high rate of morbidity and mortality in children[1,2]. Molecular characterizations of MB have revealed disease heterogeneity associated with four major subgroups[3,4]: WNT, SHH, group 3 and group 4. Group 3 MB (hereafter referred to as G3), which accounts for 25–30% of all MB cases, is the

most aggressive and malignant, characterized by frequent metastasis at diagnosis and the worst prognosis[5]. Metastatic tumours, rather than primary tumours or recurrent tumours at the primary site, have a particularly high mortality rate in patients with MB[6,7]. Despite rarely spreading to extraneural organs, MB metastasizes almost exclusively to the spinal and intracranial leptomeninges through the cerebrospinal fluid and/or the bloodstream[6,8,9]. How MB cells acquire mobility for metastatic dissemination is poorly understood.

G3 is thought to arise from Nestin[+] early neural stem cells that give rise to GABAergic and glutamatergic neurons, the two major lineages of the cerebellum[10]. Decades of studies describing the morphological, cellular and molecular features of the developing cerebellum have implicated abnormal cerebellar development as a major determining factor for neurological diseases, including MB[11–13]. Yet the cellular and molecular mechanisms of MB tumour metastatic dissemination remain elusive.

In this study, we identify a molecular circuit that regulates the migration and positioning of Purkinje cells (PCs), a principal GABAergic neuron population in cerebellar development. Of note, MB cells hijack this molecular circuit using an abnormal epigenetic programme to promote tumour metastasis. These findings shed light on the mechanisms associated with tumour dissemination and potential targeted therapies for this childhood cancer.

## Results

### SMARCD3 is increased in G3 and related to tumour metastasis

Given that epigenetic deregulation plays a crucial role in the development and progression of MB[14], we explored the epigenetic regulators involved in MB aggressiveness, focusing on the oncobiology of G3. We defined G3-associated differentially expressed genes (DEGs) by analysing the transcriptomes of 1,350 MB samples from patients and 291 cerebellum samples from unaffected individuals[15] (Fig. 1a). G3-associated DEGs were then intersected with epigenetic related genes from the EpiFactors database, which contains 720 DNA-modifying, RNA-modifying, histone-modifying and chromatin-modifying enzymes and their cofactors[16]. Notably, *SMARCD3* was the sole G3-associated DEG related to epigenetic modifications (Fig. 1b). An analysis of two transcriptomics datasets[15,17] revealed that *SMARCD3* mRNA expression levels were significantly higher in G3 relative to other MB subgroups and unaffected tissues (Fig. 1c and Extended Data Fig. 1a). An analysis of single-cell RNA sequencing (scRNA-seq) data[18] demonstrated that the majority of G3 cells (40.98%) expressed SMARCD3 compared with cells in the other subgroups (G4, 15.67%; SHH, 5.43%; WNT, 13.14%) (Fig. 1d and Extended Data Fig. 1b). Consistently, higher levels of SMARCD3 expression were observed in G3 than in the other MB subgroups in a proteomics dataset[19] (Fig. 1e). Moreover, higher levels of *SMARCD3* mRNA expression were significantly correlated with poorer prognosis of patients with MB across all subgroups, which was independent of age and sex (Fig. 1f and Extended Data Fig. 1c). Notably,

a slight trend in the correlation between patient survival and *SMARCD3* mRNA expression levels was observed in G3 only. This result might be due to the high but small variation in SMARCD3 expression levels among each patient in this aggressive MB subgroup (Fig. 1f and Extended Data Fig. 1d). Immunohistochemistry (IHC) analysis using human MB tissue microarrays revealed that high SMARCD3 levels were associated with worse patient outcomes in all MB subgroups, but a trend for worse survival in G3 (Fig. 1g). These results suggest that SMARCD3 is highly expressed in G3 and may play a crucial role in MB aggressiveness.

A gene ontology (GO) analysis based on SMARCD3-associated genes using a MB transcriptomics dataset[4] (Supplementary Table 1) revealed that SMARCD3 is involved in biological processes for regulating cell membrane projection and organization related to cell motility and migration (Fig. 1h). To examine the positive correlation between high SMARCD3 expression levels and increased tumour metastasis, analyses of transcriptomics and proteomics datasets[4,19] revealed that patients with metastases from all types of MB and G3 exhibited higher levels of *SMARCD3* mRNA and protein expression than those in patients without metastases (Fig. 1i and Extended Data Fig. 1e). Moreover, patients with higher SMARCD3 levels had a higher frequency of tumour metastasis (Extended Data Fig. 1f). A gene distribution analysis revealed that SMARCD3 is in the top 7.331% of the 1,937 genes that are highly expressed in G3 tumours with metastasis and the top 8.584% of the 3,984 genes that are highly expressed in all MB subgroups with metastasis compared with MB types without metastasis ($P < 0.05$, $\log_2$(fold change) > 0) (Extended Data Fig. 1g). Experimentally, G3 cell lines with higher SMARCD3 levels exhibited increased migratory abilities in Transwell assays and a higher metastatic capacity in the brain and spine of mice bearing MB xenografts (Fig. 1j,k and Extended Data Fig. 1h). Together, these data demonstrate a strong correlation between SMARCD3 expression levels and MB metastasis.

### SMARCD3 drives MB cell migration and tumour metastasis

To examine whether SMARCD3 promotes MB metastatic dissemination, we generated two CRISPR–Cas9-mediated SMARCD3 knockout (KO) G3 cell lines: MED8A and D341. These cell lines exhibited decreased cell migration in scratch-wound healing and in Transwell assays (Fig. 2a,b and Extended Data Fig. 2a–d). Bioluminescence imaging (BLI) of mice bearing orthotopic xenografts of SMARCD3 KO MED8A cells showed a decreasing percentage of spinal metastasis compared with control mice bearing wild-type (WT) cells (Fig. 2c and Extended Data Fig. 2e). Moreover, SMARCD3 was highly expressed in the tumour margin compared with the tumour centre (Fig. 2d), which suggests that MB cells with high SMARCD3 levels tend to spread from the primary tumour site.

Notably, SMARCD3 expression levels in the metastatic tumour cell line D458 were higher than those in the matched primary tumour cell line D425 (ref. [20]) (Fig. 1j). Therefore, we performed loss-of-function and gain-of-function studies using these paired cell lines. SMARCD3 deletion decreased D458 cell migration and spinal metastasis in

**Fig. 1 | High levels of SMARCD3 expression in G3 correlate with MB metastasis. a**, A heatmap of gene expression in the four MB subgroups (G3, group 4 (G4), SHH and WNT) and in unaffected (normal) tissues. Twofold change; false discovery rate (FDR) < 0.05. **b**, Venn diagram showing the overlapping *SMARCD3* expression between G3-associated genes and epigenetic genes. **c**, Violin plot showing *SMARCD3* mRNA expression using transcriptomics data from patients with MB. ANOVA, analysis of variance. **d**, Uniform manifold approximation and projection (UMAP) visualization (left) and violin plot (right) showing *SMARCD3* mRNA expression based on scRNA-seq data from 25 patients with MB. **e**, Boxplot showing levels of SMARCD3 expression ($n_{G3} = 14$, $n_{G4} = 13$, $n_{SHH} = 15$, $n_{WNT} = 3$). **f**, Kaplan–Meier survival curve of patients comparing all MB subgroups (left) and G3 only (right) based on *SMARCD3* mRNA expression level. **g**, Left, representative images of IHC staining for SMARCD3 levels in MB tissue microarrays. Right, log-rank test for survival fraction of patients comparing all MB subgroups and G3 only based on SMARCD3 level. **h**, Top ten biological

pathways of the SMARCD3-associated genes in MB by GO analysis. **i**, Density plots (top) and boxplots (bottom) showing the association between metastasis status (0, no metastasis; 1+, metastasis at diagnosis) and *SMARCD3* mRNA ($n_0 = 397$, $n_{1+} = 176$) and protein ($n_0 = 23$, $n_{1+} = 20$) expression levels in primary MB samples. **j**, RT–qPCR (top) and immunoblotting (bottom) analyses showing *SMARCD3* mRNA ($n = 3$) and protein levels in six G3 MB cell lines. **k**, Representative haematoxylin and eosin (H&E) images showing primary tumours (yellow dashed lines) and brain and spinal metastatic tumours (red dashed lines) in six orthotopic xenograft models derived from G3 MB cell lines. Images are representative of three independent mice, with similar results obtained (**k**). Each dot represents one bulk sample (**c,e,i**) or one cell (**d**). *n* represents the number of human patients (**a,c,e,f,g,i**) or biologically independent samples (**j**). Data are presented as the mean ± s.d. *P* values were calculated using two-tailed Welch's *t*-test with FDR correction (**c,e,i**) or two-tailed accumulative hypergeometric distribution (**h**).

mice (Fig. 2e,f and Extended Data Fig. 2f,g). Circulating tumour cells (CTCs) in peripheral blood are considered to mediate MB leptomeningeal metastasis[6]. Accordingly, we observed fewer mice with green-fluorescent-protein-positive (GFP+) D458 CTCs after SMARCD3 deletion (Fig. 2g,h). Consistently, overexpression (OE) of SMARCD3 in D425 cells increased cell migration, spinal metastasis and the percentage of tumour-bearing mice with CTCs (Fig. 2i–k and Extended Data Fig. 2h,i). Moreover, SMARCD3 OE D425-derived GFP+ mice had enhanced tumour dissemination in the spinal cord and the local brain

compared with WT D425-derived GFP+ mice (Fig. 2l,m). These results indicate that SMARCD3 has a pivotal role in the phenotypic determination of MB cell migration and metastasis.

We next sought to directly observe and characterize the migratory behaviour of tumour cells. Time-lapse imaging of MED8A cells from in vitro scratch-wound healing assays and ex vivo brain slice models showed that SMARCD3 deletion decreased cell movement, including directional cell migration velocity and non-directional cell motility speed (Supplementary Videos 1–3 and Extended Data Fig. 2j,k).

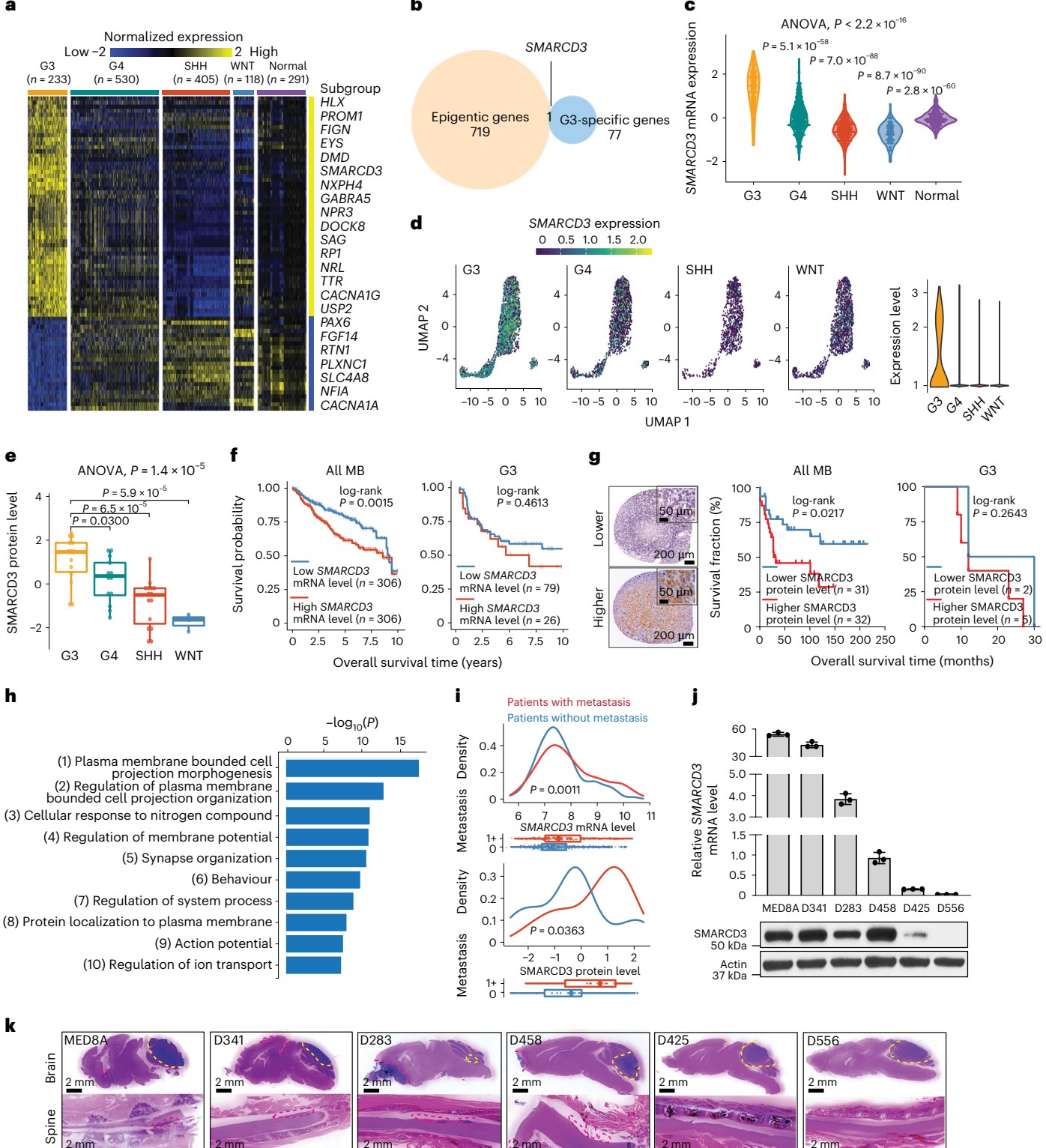

To better understand how SMARCD3 influences MB growth besides metastasis, we performed bromodeoxyuridine and cell proliferation assays using MED8A and D458 cells. No significant differences in cell viability and growth were observed following genetic alteration of SMARCD3 expression (Extended Data Fig. 3a,b). Mice bearing orthotopic xenograft tumours with SMARCD3 KO or OE exhibited moderate survival differences compared with the controls (Extended Data Fig. 3c). This result suggests that SMARCD3 may have a moderate influence on tumour cell proliferation, which leads to continued growth of the primary tumours. However, when we grouped these mice to increase the cohort size, we found a significantly decreased survival time in mice with high SMARCD3 levels (MED8A, D458 and D425-SMARCD3 OE) compared with mice with low SMARCD3 levels (MED8A-SMARCD3 KO, D458-SMARCD3 KO and D425) (Fig. 2n). These data provide evidence to indicate that SMARCD3-induced metastasis, rather than proliferation, contributes to a worse prognosis in these mouse models. This result was further supported by the lack of correlation between proliferating cell nuclear antigen scores and metagene scores[21] and SMARCD3 expression levels in patients with MB (Extended Data Fig. 3d).

To determine whether increased SMARCD3 levels contribute to MB development, we used virus-induced spontaneous tumour formation in postnatal C57BL/6J mice. Notably, OE of constitutively active MYC[S62D] alone and MYC[S62D] + SMARCD3 induced tumour formation; however, SMARCD3 OE alone did not (Extended Data Fig. 3e,f). Although a significant difference between the two groups was not obtained, there was a trend in shorter survival times in mice bearing SMARCD3 + MYC[S62D]-induced tumours compared with MYC[S62D]-induced tumours (Extended Data Fig. 3f). Furthermore, GFP fluorescence analyses showed no obvious differences in tumour size of MYC[S62D]-induced tumours with or without SMARCD3 OE (Extended Data Fig. 3g). By contrast, MYC[S62D]-induced tumours promoted by SMARCD3 OE led to spinal metastases (Extended Data Fig. 3h). Histopathology and IHC analyses revealed that both MYC[S62D]-induced and SMARCD3 + MYC[S62D]-induced tumours showed the typical features of G3, but no differences in the cell proliferation index (based on Ki-67 staining levels) were observed between these two tumour groups (Extended Data Fig. 3i). In a human cerebellar neural stem cell (hcNSC) line with low malignant potential for MB formation[22], SMARCD3-induced tumour formation was not observed for up to 90 days. However, MYC[S62D] OE in hcNSCs substantially increased tumour formation in orthotopic SCID mouse models (Extended Data Fig. 3j). Significant differences in mouse survival and tumour sizes were not observed between MYC[S62D]-induced and SMARCD3 + MYC[S62D]-induced tumours; however, SMARCD3 OE promoted tumour spinal metastasis of MYC[S62D]-induced tumours (Extended Data Fig. 3k–m). Collectively, our in vitro and in vivo loss-of-function and gain-of-function studies together with the patient data analysis suggest that SMARCD3 acts as the main driver in tumour metastatic dissemination in the evolution of MB.

## SMARCD3 upregulates DAB1 for MB cell migration

To delineate the molecular mechanisms of how SMARCD3 promotes MB metastasis, we performed RNA-seq of SMARCD3 KO cells and WT MED8A cells. Ingenuity pathway analyses (IPA) based on the 44 downregulated and 67 upregulated DEGs (fourfold change; $P < 0.05$) showed that Reelin signalling in neurons was the most significantly enriched (Fig. 3a and Supplementary Table 2). Reelin plays a pivotal part in cell migration and positioning throughout the central nervous system by binding to its receptors the very-low-density lipoprotein receptor (VLDLR) and/or the apolipoprotein E receptor-2 (ApoER2, encoded by *LRP8*)[23]. Reelin also promotes downstream activation of DAB1 signalling through the phosphorylation of key tyrosine residues (for example, Y232)[23,24]. Gene expression of key Reelin signalling components (*Reln*, *Vldlr*, *Dab1* and *Dcc*) was decreased in SMARCD3 KO MED8A cells (Fig. 3b).

We further validated that *DAB1* mRNA expression is decreased in SMARCD3 KO MED8A and D458 cells but increased in SMARCD3-overexpressed MED8A, D425 and D556 cells (Fig. 3c,d). Integrated analysis of transcriptomic and proteomics data of samples from patients with MB[19] revealed that *DAB1* mRNA expression was correlated with translational and post-translational modifications of DAB1, including phosphorylation on serine, threonine or tyrosine (pSTY), particularly Y232 (Extended Data Fig. 4a). Analyses of datasets of samples from patients with MB[15,19] showed that *DAB1* mRNA and protein levels were significantly higher in G3 compared with other MB subgroups and unaffected cerebellum tissues (Fig. 3e,f and Extended Data Fig. 4b). Given the relatively small variation in *SMARCD3* and *DAB1* mRNA expression in G3 compared with the other MB subgroups (Extended Data Figs. 1d and 4c), we analysed the datasets of all patients with MB and found positive correlations between SMARCD3 and DAB1 at the transcriptional, translational and post-translational levels[4,19] (Fig. 3g,h and Extended Data Fig. 4d). Experimental validation revealed that DAB1 deletion in MED8A cells decreased cell migration (Fig. 3i,j). Moreover, an analysis of a patient dataset[4] showed that DAB1 expression was associated with MB metastasis across all subgroups (Fig. 3k,l). These results suggest that SMARCD3 upregulates Reelin–DAB1 signalling to promote cell migration and MB metastasis.

## SMARCD3 regulates Reelin signalling in cerebellar development

We asked whether a positive correlation between SMARCD3 and DAB1 exists in other human cancers or healthy organs. Pan-cancer analyses using The Cancer Genome Atlas datasets revealed that the levels of

---

**Fig. 2 | SMARCD3 promotes cell migration and tumour metastasis. a**, IB for SMARCD3 expression in MED8A cells with control (WT) and SMARCD3 KO using two independent single-guide RNAs (sgRNAs; KO-1 and KO-2). **b**, Representative images (left) and quantification (right) showing cell migration of MED8A cells with SMARCD3 WT ($n = 5$), KO-1 ($n = 5$) or KO-2 ($n = 5$) in Transwell assays. **c**, Representative luminescence images (left) and pie charts (right) showing mice bearing MED8A cells with SMARCD3 WT or KO-1 after implantation. **d**, Representative IHC staining of SMARCD3 in MED8A-derived xenograft MB tumours. High-magnification images show a part of the tumour margin and core areas. **e**, IB for SMARCD3 expression in D458 cells with SMARCD3 WT or KO-1. **f**, Representative luminescence images (left) and pie charts (right) showing mice bearing D458 cells with SMARCD3 WT or KO-1 after implantation. **g**, Representative bright-field and fluorescence microscopy images of mouse brains bearing D458 cells with SMARCD3 WT or KO. **h**, Flow cytometry (left) and pie chart (right) analysis of GFP⁺ CTCs from peripheral blood mononuclear cells (PBMCs) of mice bearing D458 cells with SMARCD3 WT or KO (GFP⁺ ≥ 0.01%). **i**, RT–qPCR (top) and IB (bottom) for the *SMARCD3* mRNA and protein expression levels in D425 cells with vector ($n = 4$) or SMARCD3 OE ($n = 4$). **j**, Representative luminescence images (left) and pie charts (right) showing mice bearing D425 cells with vector or SMARCD3 OE after implantation. **k**, Flow cytometry (left) and pie chart (right) analysis of GFP⁺ CTCs from PBMCs of mice bearing D425 cells with vector or SMARCD3 OE. **l**, Representative bright-field and fluorescence microscopy images of the spinal cords from mice bearing D425 cells with vector or SMARCD3 OE. **m**, Left: representative fluorescence stereoscopic images of mouse brain tumours derived from D425 cells with vector ($n = 5$) or SMARCD3 OE ($n = 5$). Insets: high-magnification images were donated. Right: histograms showing the number of brain metastases. **n**, Kaplan–Meier survival curve of the grouped mice bearing cells with high (MED8A, D458, D425-SMARCD3 OE) or low (MED8A-SMARCD3 KO, D458-SMARCD3 KO, D425) levels of SMARCD3 expression. The red arrow denotes the metastatic tumour observed by in vivo (**c,f,j**) or fluorescence (**l**) imaging. $n$ represents the number of biologically independent samples (**b,i**) or mice (**m**). Data are presented as the mean ± s.d. $P$ values were calculated using one-way ANOVA with Dunnett's multiple comparison test (**b**) or one-tailed unpaired $t$-test (**i,m**). ****$P < 0.0001$. At least five (**a,d,e,g,m**) or four (**l**) replicates per experiment were repeated independently, with similar results obtained.

*SMARCD3* and *DAB1* mRNA expression were not correlated ($R = 0.17$, $P < 2.2 \times 10^{-16}$), including no positive correlation in low-grade glioma and glioblastoma ($R = -0.11$, $P = 0.0023$) (Extended Data Fig. 4e,f). A gene expression correlation analysis of various human healthy organs revealed that SMARCD3 and DAB1 were significantly correlated and highly expressed in the brain compared with other organs, especially

in the cerebellar hemisphere and cerebellum (Extended Data Fig. 4g,h). An analysis of gene-specific patterns of expression variation across organs and species[25] revealed that SMARCD3 and DAB1 expression varied considerably across organs but varied little across species (Extended Data Fig. 4i), which indicated a potential evolutionary conservation of organ-specific gene expression in vertebrates. These data

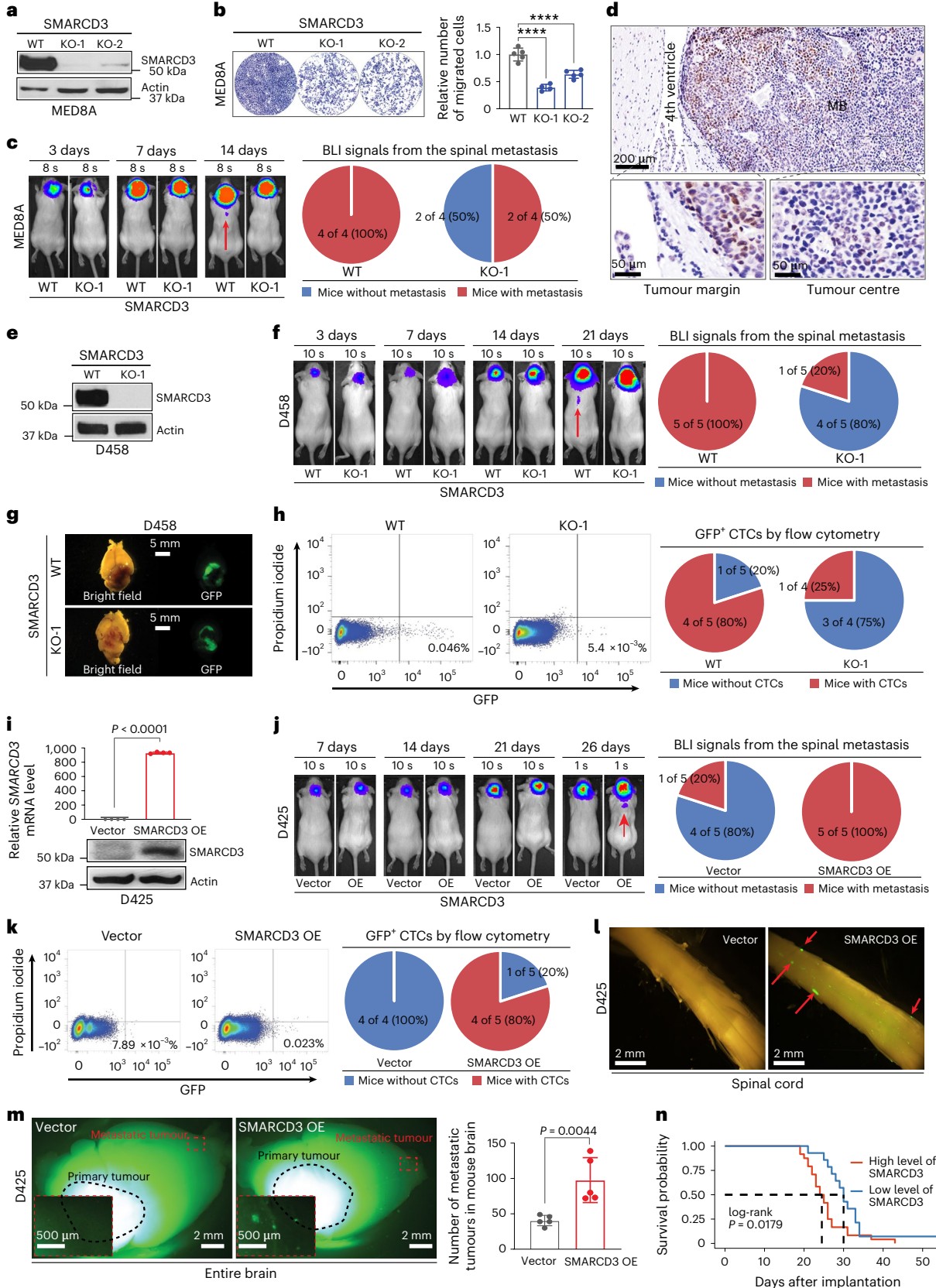

suggest that SMARCD3 regulation of DAB1-mediated Reelin signalling is specific to the cerebellum in physiological and pathological conditions.

Reelin signalling controls PC radial migration and cerebellar circuit function in brain development[13]. Thus, we asked whether SMARCD3 is positively correlated with Reelin signalling in the developmental trajectory of the cerebellum. We analysed scRNA-seq data from the developing mouse cerebellum[26] and found that *Smarcd3*, *Dab1*, *Vldlr* and *Lrp8* mRNA were highly expressed in PCs (Fig. 4a,b and Extended Data Fig. 5a). PCs emerge in the ventricular zone from embryonic day 10.5 (E10.5) to E13.5 in mice and from gestation week 7 (GW7) to GW13 in humans[27,28] (Extended Data Fig. 5b), then migrate towards the outer surface of the cerebellar cortex to subsequently form the PC layer from E12.5 to the early postnatal days in mice and during GW16–GW28 in humans[13,29,30]. Reelin secreted by glutamatergic neurons (granule cells (GCs)) acts on PCs and activates its downstream VLDLR–ApoER2–DAB1 signalling pathway to control PC migration[31,32]. We found low levels of *Smarcd3*, *Dab1*, *Vldlr* and *Lrp8* but high levels of *Reln* mRNA expression in GCs (Fig. 4b). Further analysis of spatiotemporal gene expression revealed a similar trajectory of *Smarcd3* expression and Reelin signalling, particularly *Dab1* expression in PCs (Fig. 4c and Extended Data Fig. 5c). Immunofluorescence staining of SMARCD3 with the PC-specific markers FOXP2 and calbindin 1 (CALB1) revealed increased SMARCD3 levels that colocalized with FOXP2 and CALB1 at E15.5 and postnatal day 0 (P0), respectively. Moreover, substantially decreased SMARCD3 levels after P0 that remained low or undetectable at P7, P28 and P84 in the mouse cerebellum were observed (Fig. 4d,e).

Analyses of single-nucleus RNA-seq data of 13 samples of human cerebella ranging in age from 9 to 21 post-conceptional weeks[33] revealed that SMARCD3 is highly expressed and associated with DAB1, VLDLR and LRP8 expression in PCs. Moreover, RELN was exclusively expressed in glutamatergic neurons, including precursor, cerebellar nuclei and GCs (Fig. 4f and Extended Data Fig. 5d,e). We further analysed normalized gene expression data of 291 samples of healthy cerebella across four age groups: foetal (year ≤ 0), infants (0 < years ≤ 3), children (3 < years < 18) and adults (≥ 18 years)[15]. *SMARCD3* mRNA expression was increased from around GW13 to GW28, then substantially decreased during 1 year postnatal and maintained at low levels in infant, children and adult age groups (Fig. 4g,h). These results suggest that spatiotemporal expression patterns of SMARCD3 are associated with Reelin signalling in the control of PC migration during cerebellar development. GO term and gene disease network (DisGeNET) analyses using the SMARCD3-positively related genes during human cerebellar development revealed enrichment for biological processes involved in cell projection assembly and organization, brain development, response to wounding and pathways in childhood and adult MB (Supplementary Table 3 and Extended Data Fig. 5f,g). Collectively, these results indicate that MB hijacks SMARCD3–Reelin–DAB1-mediated cell migration, a neurodevelopmental programme in the cerebellum, to promote tumour metastatic dissemination.

## Epigenomic regulation of DAB1 by SMARCD3 in cerebellum and MB

To determine the functions of SMARCD3 in the genome architecture that regulates the gene expression of components involved in cell migration and tumour metastasis, we performed assay for transposase-accessible chromatin using sequencing (ATAC-seq) to obtain nucleosome-free fragments (<100 base pairs) and mononucleosome fragments (180–247 base pairs)[34]. Global changes in chromatin accessibility in SMARCD3 KO cells were observed compared with WT MED8A cells (Fig. 5a and Extended Data Fig. 6a,b). Out of 144,432 total accessible regions identified, 20,578 ATAC-seq peaks had increased accessibility and 10,131 peaks had decreased accessibility in SMARCD3 KO cells compared with WT cells (Fig. 5a). Genes (n = 725) proximal to these less-accessible peaks (positive correlation with SMARCD3) were involved in cellular movement, assembly and organization according to IPA (Fig. 5b). These data suggest that SMARCD3 regulates chromatin remodelling to promote cell migration and tumour dissemination.

We next assigned these differentially accessible regions to the nearest genes that could be regulated by *cis*-regulatory elements (CREs). Notably, changes in chromatin accessibility of most genes (90.29%) corresponded to changes in gene expression according to the RNA-seq results (Fig. 5c). Specifically, decreased accessibility of *DAB1* in the absence of SMARCD3 was consistent with its decrease in mRNA expression levels (Figs. 3b and 5c). To identify specific CREs in the genome that control SMARCD3-mediated *DAB1* regulation, we defined the topologically associating domain regions that were enriched in the *DAB1* locus using available Hi-C data[35] (Extended Data Fig. 6c). Analyses of the ATAC-seq and cleavage under targets and release using nuclease (CUT&RUN) data[36,37] revealed that the four CREs (CRE1, CRE2, CRE3 and CRE4) were associated with the decreased chromatin accessibility and histone modifications in SMARCD3 KO cells compared with WT MED8A cells (Fig. 5d and Extended Data Fig. 6d). Notably, there were obvious changes in CRE2 for accessibility and H3K4me3 at the transcription start site of *DAB1* between SMARCD3 KO cells and WT cells (Fig. 5d). This result indicates that CRE2 has a key function in SMARCD3-mediated DAB1 transcriptional activity.

To validate that these CREs are involved in DAB1 regulation in cerebellar development and MB, we analysed a dataset of chromatin immunoprecipitation sequencing (ChIP-seq) chromatin modification profiles and RNA-seq-based transcriptomics from five human G3 samples[38]. We classified the five tumours into higher or lower levels of *SMARCD3* mRNA expression (Extended Data Fig. 6e). Then the ChIP-seq enrichment data from the four CREs proximal to the *DAB1* locus in each tumour were pooled into either the higher or lower group. We observed histone mark enrichment at these CREs, particularly CRE2, in the higher compared with the lower group (Fig. 5e). Analyses of ChIP-seq datasets from mouse cerebellum[39] showed increased H3K4me3 and H3K27ac signals from E12.5 to P0, but decreased H3K4me3 and H3K27ac signals at P56. The signals localized at these CREs of the *Dab1* locus, particularly CRE2, which corresponded to *Dab1* expression during mouse cerebellar

---

**Fig. 3 | SMARCD3 promotes MB metastasis through the Reelin–DAB1 signalling pathway. a**, IPA canonical pathway enrichment analysis of DEGs in MED8A cells with SMARCD3 KO or WT. **b**, Volcano plot illustrating the DEGs in MED8A cells with SMARCD3 KO or WT (adjusted *P* < 0.05; twofold change). **c**, RT–qPCR analysis of *DAB1* mRNA expression in MED8A ($n_{KO}$ = 4, $n_{WT}$ = 4) and D458 ($n_{KO}$ = 12, $n_{WT}$ = 8) cells with SMARCD3 KO or WT. **d**, RT–qPCR analysis of *DAB1* mRNA expression in MED8A ($n_{SMARCD3\ OE}$ = 4, $n_{vector}$ = 4), D425 ($n_{SMARCD3\ OE}$ = 8, $n_{vector}$ = 6) and D556 ($n_{SMARCD3\ OE}$ = 12, $n_{vector}$ = 12) cells with SMARCD3 OE or vectors. **e**, Violin plot showing *DAB1* mRNA expression in MB and healthy cerebellum. **f**, Boxplots showing expression levels of total DAB1 ($n_{G3}$ = 14, $n_{G4}$ = 13, $n_{SHH}$ = 15, $n_{WNT}$ = 3) and phospho-DAB1 (Y232) ($n_{G3}$ = 11, $n_{G4}$ = 9, $n_{SHH}$ = 11, $n_{WNT}$ = 3) protein in proteomics datasets. **g**, Scatterplot showing the correlation between *SMARCD3* and *DAB1* mRNA expression in 1,280 MB samples. **h**, Scatterplots showing the correlations between SMARCD3 and total or phospho-DAB1 protein expression

in 45 MB samples. **i**, RT–qPCR analysis of *DAB1* mRNA expression in MED8A cells with DAB1 KO (n = 8) (three independent sgRNAs) or WT (n = 8). **j**, Representative images (left) and quantification (right) of cell migration of MED8A cells with DAB1 KO ($n_{KO-1}$ = 5, $n_{KO-4}$ = 10, $n_{KO-5}$ = 5) or WT (n = 5) in Transwell assays. **k**, Bar diagrams showing the percentage of patients with MB with or without metastasis (0, no metastasis; 1+, metastasis at diagnosis) between high and low *DAB1* mRNA expression. **l**, Boxplot showing *DAB1* mRNA expression in patients with MB with metastasis compared with without metastasis. Each dot represents one patient bulk sample (**e**–**h**). *n* represents the number of biologically independent samples (**c**,**d**,**i**,**j**) or patient samples (**f**). Data are presented as the mean ± s.d. *P* values were calculated using right-tailed Fisher's exact test (**a**), one-tailed unpaired *t*-test (**c**,**d**), two-tailed Welch's *t*-test with FDR correction (**e**,**f**,**l**), two-tailed Spearman's rank correlation analysis (**g**,**h**) or one-way ANOVA with Dunnett's multiple comparison test (**i**,**j**). \*\*\*\**P* < 0.0001.

development (Fig. 5f and Extended Data Fig. 6f,g). These data suggest that SMARCD3 epigenetically regulates DAB1 transcriptional activity by controlling chromatin accessibility and histone modifications of CREs in the developing cerebellum and in MB.

## Chromatin dynamics of SMARCD3 expression in the cerebellum and in MB

To examine the epigenetic regulation of SMARCD3 in the development of MB and the cerebellum, we analysed ATAC-seq and CUT&RUN data of MED8A cells. We identified the seven accessible regions (CRE1–CRE7)

proximal to the *SMARCD3* locus, which were enriched with peaks of H3K4me1, H3K4me3 and/or H3K27ac as hallmarks of active or poised enhancers (Fig. 6a). To verify that these CREs are involved in SMARCD3 regulation, we analysed ChIP-seq and RNA-seq datasets of five samples from patients with MB[38]. H3K4me1, H3K4me3 and H3K27ac were enriched at these CREs in the higher compared with the lower group (Fig. 6b and Extended Data Fig. 6e). In particular, H3K27ac, a marker of active enhancers and transcription start sites, was significantly enriched at these CREs in G3 compared with the other MB subgroups. This result corresponded with SMARCD3 expression levels based on

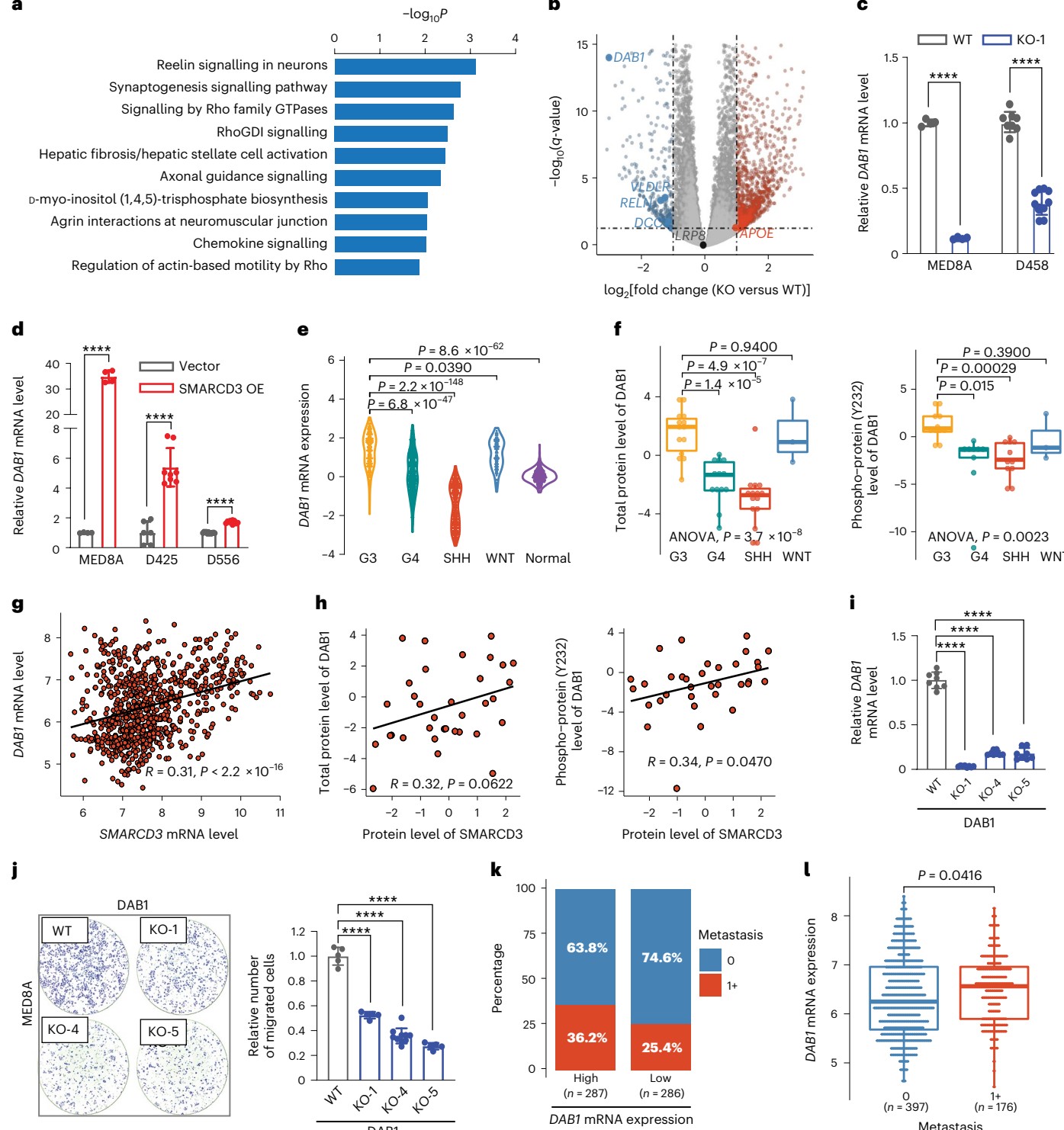

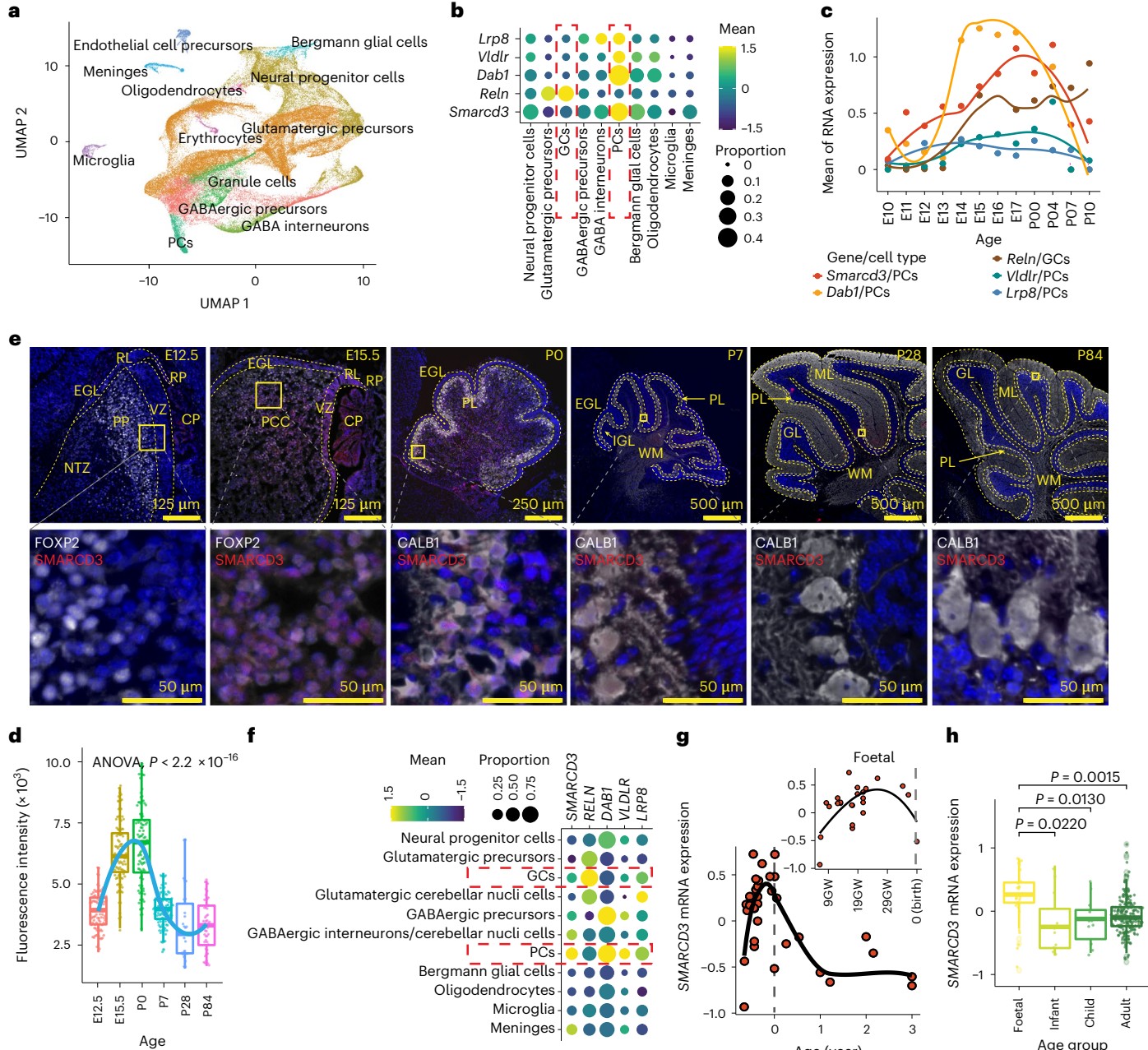

**Fig. 4 | SMARCD3 regulates Reelin–DAB1 signalling in the developing cerebellum. a**, UMAP visualization and marker-based annotation of cell types from developing mouse cerebellum. **b**, Dot plot showing gene expression in the indicated cell types from the developing mouse cerebellum. **c**, mRNA expression in mouse PCs and GCs across the timeline of cerebellar development. **d**, Boxplot showing fluorescence intensity of SMARCD3 expression in PCs at each time point ($n_{E12.5} = 100$, $n_{E15.5} = 100$, $n_{P0} = 100$, $n_{P7} = 100$, $n_{P28} = 26$, $n_{P84} = 43$). **e**, Representative images of SMARCD3 (red) and FOXP2 (white) or CALB1 (white) in mouse cerebellum at each time point. Dashed lines outline indicated cerebellar regions. CP, choroid plexus; EGL, external granule layer; GL, granular layer; IGL, internal granule layer; ML, molecular layer; NTZ, nuclear transitory zone; PCC, Purkinje cell plate; PL, Purkinje layer; RL, upper rhombic lip; RP, roof plate;

VZ, ventricular zone; WM, white matter. **f**, Dot plot showing gene expression in the indicated cell types from the developing human cerebellum. **g**, Scatterplots showing changes in *SMARCD3* mRNA expression of human cerebella across the developmental process. **h**, Boxplot showing *SMARCD3* mRNA expression levels in human cerebella from the indicated age groups. Each dot represents one cell (**a**,**d**) or a patient sample (**g**,**h**). Dot colour reflects the mean gene expression and dot size represents the percentage of cells expressing the gene (**b**,**f**). *n* represents the number of patient samples ($n_{Foetal} = 29$, $n_{Infant} = 11$, $n_{Child} = 12$, $n_{Adult} = 215$ for **g**,**h**). Representative images from four independent mice at each time point were repeated, with similar results obtained (**e**). Data are presented as the mean ± s.d. *P* values were calculated using one-way ANOVA (**d**) or two-tailed Welch's *t*-test with FDR correction (**h**).

an analysis of a previously published RNA-seq dataset[40] (Extended Data Fig. 7a–c). Analyses of the public enhancer datasets ENCODE and Roadmap further supported these newly identified CREs at the *SMARCD3* locus in human and mouse genomes (Extended Data Fig. 7d,e). To explore these chromatin dynamics in cerebellar development, we analysed Hi-C data to map the regulatory regions of the

mouse *Smarcd3* locus and then analysed the enrichment of histone modifications during cerebellar development using ENCODE datasets[39] (Fig. 6c). We observed higher enrichment of H3K4me3 and H3K27ac around these CREs at E16.5 and P0 compared with E12.5 and P56, which corresponded to the levels of *Smarcd3* mRNA expression at these time points (Fig. 6c,d). These results suggest that the CREs play a crucial

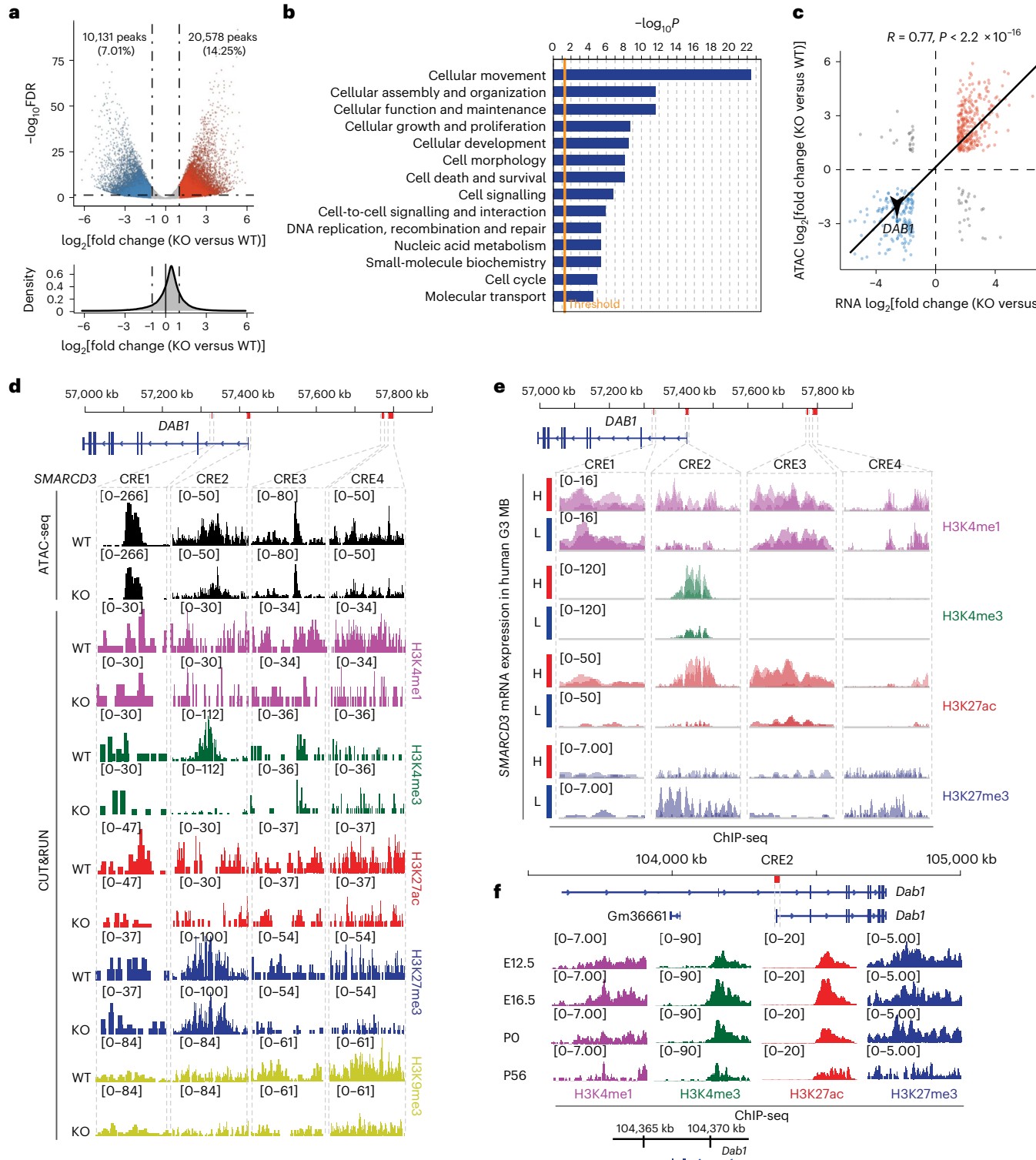

**Fig. 5 | SMARCD3 regulates DAB1 transcriptional activation through chromatin remodelling in MB and cerebellar development. a**, Volcano plots showing the differential accessibility (log$_2$(fold change) in reads per peak) against the FDR (−log$_{10}$) of MED8A cells with SMARCD3 KO or WT. Each dot represents one peak called by MACS3. **b**, The top ten molecular and cellular functions enriched according to IPA using the genes associated with reduced chromatin accessibility (FDR < 0.05; twofold change) in MED8A cells with SMARCD3 KO. **c**, Two-tailed Pearson's correlation analysis of peak accessibility in ATAC-seq

compared to DEGs in RNA-seq. **d**, ATAC-seq and histone-marker-binding signals from CUT&RUN in the *DAB1* locus using MED8A cells with SMARCD3 KO or WT. The four CREs are marked by red bars and dashed-line boxes in the schematic of the genome (top). **e**, Histone-modification signals at the four CREs based on analyses of ChIP-seq data from five samples from patients with G3. H, high; L, low. **f**, Histone-modification signals at CRE2 based on analyses of ChIP-seq data from mouse cerebellum at the indicated time points. *P* value was calculated using right-tailed Fisher's exact test (**b**).

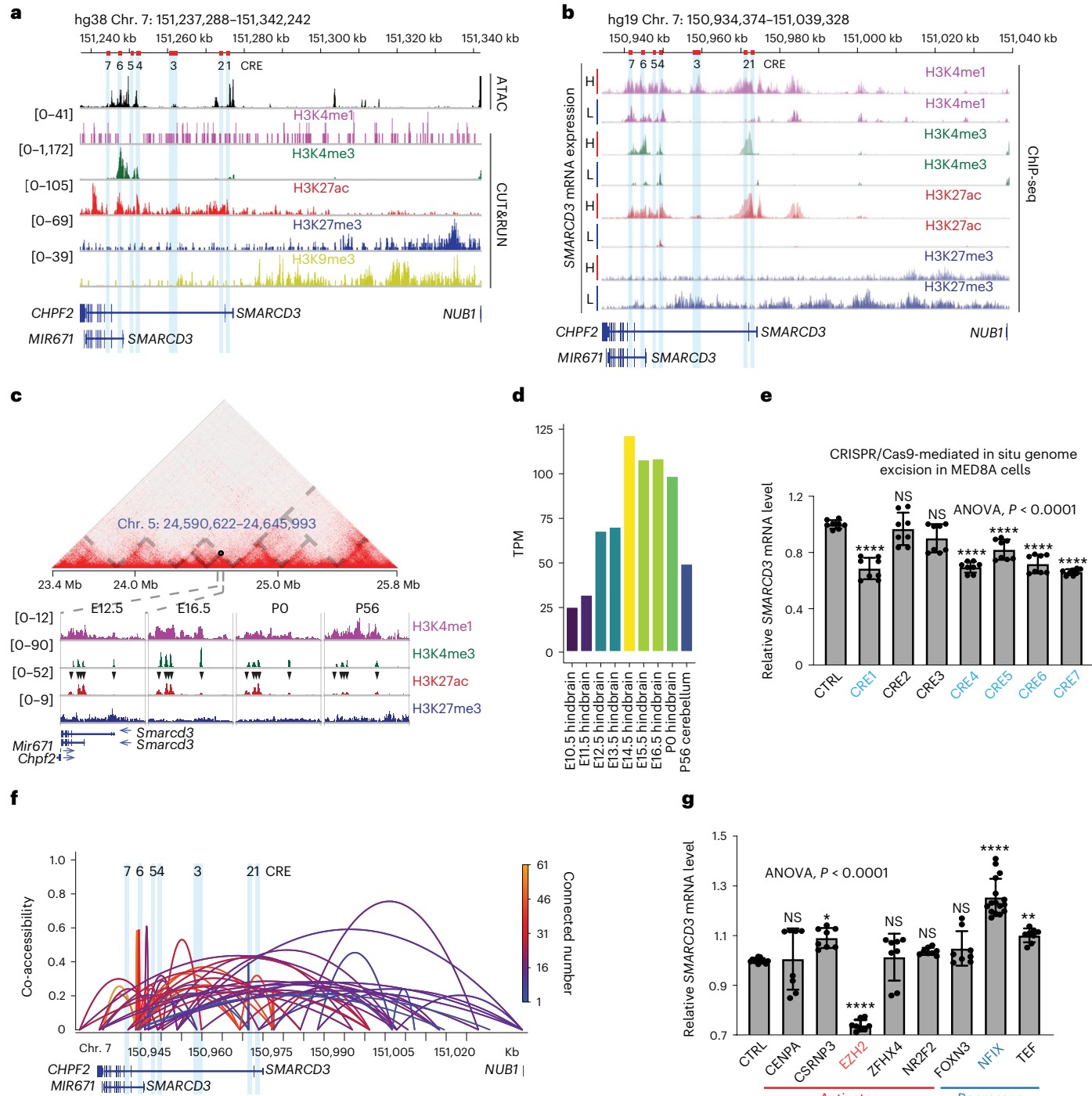

**Fig. 6 | TF-mediated chromatin hubs control SMARCD3 transcriptional activation in cerebellar development and MB. a**, ATAC-seq and histone-modification signals from CUT&RUN at the *SMARCD3* locus in MED8A cell. The CREs (1–7) are marked with red bars in the schematic of the genome (top) and in light blue. **b**, Histone-modification signals at the *SMARCD3* locus based on analyses of ChIP-seq data from five samples from patients with G3. **c**, Hi-C chromatin interaction map on a region centred in the *Smarcd3* locus in mouse cerebellum (P22). Grey dashed lines outline topologically associating domain borders. Histone-modification signals are based on analyses of ChIP-seq data of mouse cerebellum samples at the indicated time points. Black arrowheads denote the CREs that are homologous to the CREs in MED8A cells. **d**, Histogram of *Smarcd3* mRNA expression during mouse cerebellar development. TPM, transcripts per million. **e**, RT–qPCR analysis of *SMARCD3* mRNA expression in

MED8A cells after CRISPR–Cas9-mediated in situ CRE excision ($n = 8$ for each group). Excision of CREs in blue caused significant decreases in *SMARCD3* mRNA levels. **f**, Cicero co-accessibility links among SMARCD3 CREs in PCs using sci-ATAC-seq3 data from the human cerebellum. The height and colour of connections indicate the magnitude of the Cicero co-accessibility score and the number of connected peaks. **g**, RT–qPCR analysis of *SMARCD3* mRNA expression in MED8A cells after CRISPR–Cas9-mediated KO of the indicated TF ($n_{CTRL} = 12$, $n_{CENPA} = 8$, $n_{CSRNP3} = 8$, $n_{EZH2} = 8$, $n_{ZFHX4} = 8$, $n_{NR2F2} = 8$, $n_{FOXN3} = 8$, $n_{NFIX} = 16$, $n_{TEF} = 8$). $n$ represents the number of biologically independent samples from at least three independent experiments. Data are presented as the mean ± s.d. $P$ values were calculated using one-way ANOVA with Dunnett's multiple comparisons test (**e**,**g**). NS, not significant, $^{*}P = 0.0203$, $^{**}P = 0.0070$, $^{****}P < 0.0001$.

part in the regulation of SMARCD3 transcription through control of the chromatin architecture.

To functionally evaluate these CREs, we used CRISPR–Cas9-mediated in situ genome excision to remove CREs, which leads to transcriptional inactivation of targeted genes (Extended Data Fig. 8a). Quantitative PCR with reverse transcription (RT–qPCR) analysis revealed that site-specific excision of CRE1, CRE4, CRE5, CRE6 and CRE7, but not CRE2 or CRE3, resulted in decreased *SMARCD3* mRNA expression in MED8A cells (Fig. 6e). Notably, two isoforms of the *SMARCD3* gene shared CRE4–CRE7 but not CRE1, which indicated the occurrence of divergence in transcriptional regulation (Fig. 6a). In detail, we observed decreased *SMARCD3* mRNA expression after site-specific excision of CRE4–CRE7 but not CRE1 in D458 cells and increased enrichment in H3K4me3 and H3K27ac around CRE1 in MED8A cells but not in D458 cells (Fig. 6a and Extended Data Fig. 8b,c). We also found higher enrichment for H3K4me3 and H3K27ac around CRE4–CRE7 in metastatic tumour-derived D458 cells compared with the paired primary tumour-derived D425 cells (Extended Data Fig. 8c). These results implicate the involvement of these CREs in SMARCD3-mediated MB metastasis.

To define how these CREs regulate SMARCD3 transcription, we analysed datasets of the single-cell combinatorial indexing assay for profiling chromatin accessibility (sci-ATAC-seq3) in the human foetal cerebellum[41]. Higher levels of SMARCD3 expression in PCs were observed compared with astrocytes, GCs and inhibitory interneurons, which is concordant with a more open chromatin structure in PCs, an effect confirmed by higher gene activity scores calculated using Cicero, an algorithm that quantitatively measures how changes in chromatin accessibility relate to changes in the expression of nearby genes[42] (Extended Data Fig. 8d,e). Cicero links were heavily enriched around CRE4–CRE7 at the *SMARCD3* locus in PCs compared with astrocytes, GCs and inhibitory interneurons (Fig. 6f and Extended Data Fig. 8f). These data suggest that CRE1–CRE7, particularly CRE4–CRE7, can form chromatin hubs that physically and functionally control SMARCD3 transcriptional regulation.

Chromatin hubs are enriched for physical proximity, interactions with a common set of transcription factors (TFs) and orchestration of histone modifications in gene expression[42]. To identify TFs controlling the *SMARCD3* chromatin hubs, we generated a list of putative TFs that meet the following four criteria: (1) differentially expressed in the human foetal cerebellum compared with infants, children and adults (absolute log$_2$(fold change) > 0.5, $P$ < 0.05); (2) positively or negatively correlated with *SMARCD3* mRNA expression in human healthy cerebellum ($R$ > 0.25, $P$ < 0.05); (3) positively or negatively correlated with *SMARCD3* mRNA expression in G3 only or all MB subgroups ($R$ > 0.25, $P$ < 0.05); and (4) defined in the human TF database[43]. CENPA, CSRNP3, EZH2, FOXN3, NFIX, NR2F2, TEF and ZFHX4 satisfied the above criteria, and experimental validation showed that CRISPR–Cas9-mediated gene deletion of *EZH2* and *NFIX* in MED8A cells led to the most significant decrease and increase in *SMARCD3* mRNA expression, respectively (Fig. 6g). Conversely, overexpression of EZH2 significantly increased *SMARCD3* mRNA expression in MED8A and D458 cells (Extended Data Fig. 8g). An analysis of transcriptomics data from healthy human brain showed that SMARCD3 was positively correlated with EZH2 ($R$ = 0.38, $P$ = 3.1 × 10$^{-6}$) but negatively correlated with NFIX ($R$ = −0.33, $P$ = 0.0004) (Extended Data Fig. 8h). Moreover, the mRNA expression of *EZH2* and *NFIX* was oppositely changed during cerebellar development (Extended Data Fig. 8i–l). Together, these results provide a comprehensive map of a chromatin hub that orchestrates CREs, chromatin accessibility, TFs and histone modifications in the regulation of SMARCD3 transcription in the developing cerebellum and in MB metastasis (Extended Data Fig. 8m).

### Inhibiting Src activity reduces SMARCD3-induced metastasis

The Reelin–DAB1-activated Src family of tyrosine kinases (SFKs) are required for the phosphorylation of DAB1, which in turn potentiates SFK activation in a positive feedback manner. This process plays a central part in the activation of its downstream signalling cascades during cerebellar development[44,45]. We asked whether SMARCD3 levels are increased in metastatic tumours and in turn whether SFKs are activated. Consequently, we also investigated responses to SFK inhibitor treatment for clinical application (Extended Data Fig. 9a). To this end, we assessed the protein levels of SMARCD3 and phosphorylated Src (p-Src) in ten patient-matched primary and metastatic MB samples (Fig. 7a and Supplementary Table 4). IHC analysis revealed a positive correlation between SMARCD3 and p-Src (Y416), both of which were highly increased in metastatic tumours compared with the paired primary tumours (Fig. 7b–d). Furthermore, SMARCD3 deletion reduced p-Src levels in MED8A and D458 cells and in xenograft tumours derived from these cells (Fig. 7e,f and Extended Data Fig. 9b), suggesting that Src activation is induced by increased SMARCD3 expression. Similar to SMARCD3 tumour expression, we observed higher levels of p-Src in the tumour margin than in the centre (Figs. 2d and 7g).

To test our hypothesis that SFK inhibition can reduce metastatic dissemination, we examined in vitro attenuation of cell migration using a low concentration of dasatinib. A 50 nM concentration of dasatinib significantly decreased MED8A and D458 cell migration (Fig. 7h,i and Extended Data Fig. 9c,d). Next, dasatinib was administered orally once daily at the standard dose of 15 mg per kg or a low dose of 7.5 mg per kg to mice bearing D458-derived orthotopic xenograft MB. BLI and flow cytometry analyses showed that the treatments resulted in decreased spinal metastasis and a reduced percentage of mice carrying CTCs compared with placebo treatment (Fig. 7j and Extended Data Fig. 9e,f). However, assessment of tumour cell proliferation and apoptosis in these mice revealed that a low dose of dasatinib did not significantly affect the levels of Ki-67 and cleaved caspase-3 (Fig. 7k and Extended Data Fig. 9g). This result indicated that inhibition of SFK activity mainly influences cell migration rather than cell proliferation and apoptosis. Moreover, SFK inhibition may reduce tumour cell migration and metastatic dissemination at a low and safe dose in MB, which indicates a potential repurposing of this drug in clinical studies for the treatment of MB metastasis.

## Discussion

The most crucial challenge in the design of therapies for children with MB is to reduce tumour metastasis. In this study, we identified that MB cells hijack a neurodevelopmental epigenetic programme to promote metastatic dissemination, whereby abnormally increased SMARCD3 expression activates Reelin–DAB1–Src signalling-mediated cell migration. Our findings provide evidence from developmental neuroscience to translational perspectives across molecular, cellular and tissue or organ levels, in which SMARCD3 has a central role in cerebellar development and MB metastasis, and sheds light on antimetastatic therapy for patients with MB (Extended Data Fig. 9h).

SMARCD3, a subunit of the SWI/SNF chromatin remodelling complex, regulates gene expression programmes essentially for heart development and function[46,47]. Pathologically, SMARCD3 was reported to regulate epithelial–mesenchymal transition in breast cancer by inducing WNT5A signalling[48]. Our previous study[49] demonstrated that epigenetic upregulation of WNT5A contributes to glioblastoma invasiveness and recurrence. The results from this study together with the previous findings indicate a crucial role of SMARCD3 in MB metastasis, which was validated in G3, the most aggressive subgroup with strong metastatic potential compared with other MB subgroups. We further discovered that SMARCD3 epigenetically regulates Reelin–DAB1 signalling and that their positive correlation is evolutionarily conserved in the cerebellum and MB. These findings suggest that SMARCD3–Reelin–DAB1 signalling mediates PC migration and cancer cells, and hijack of this pathway for tumour metastasis could be specific to cerebellar development and MB aggressiveness, respectively.

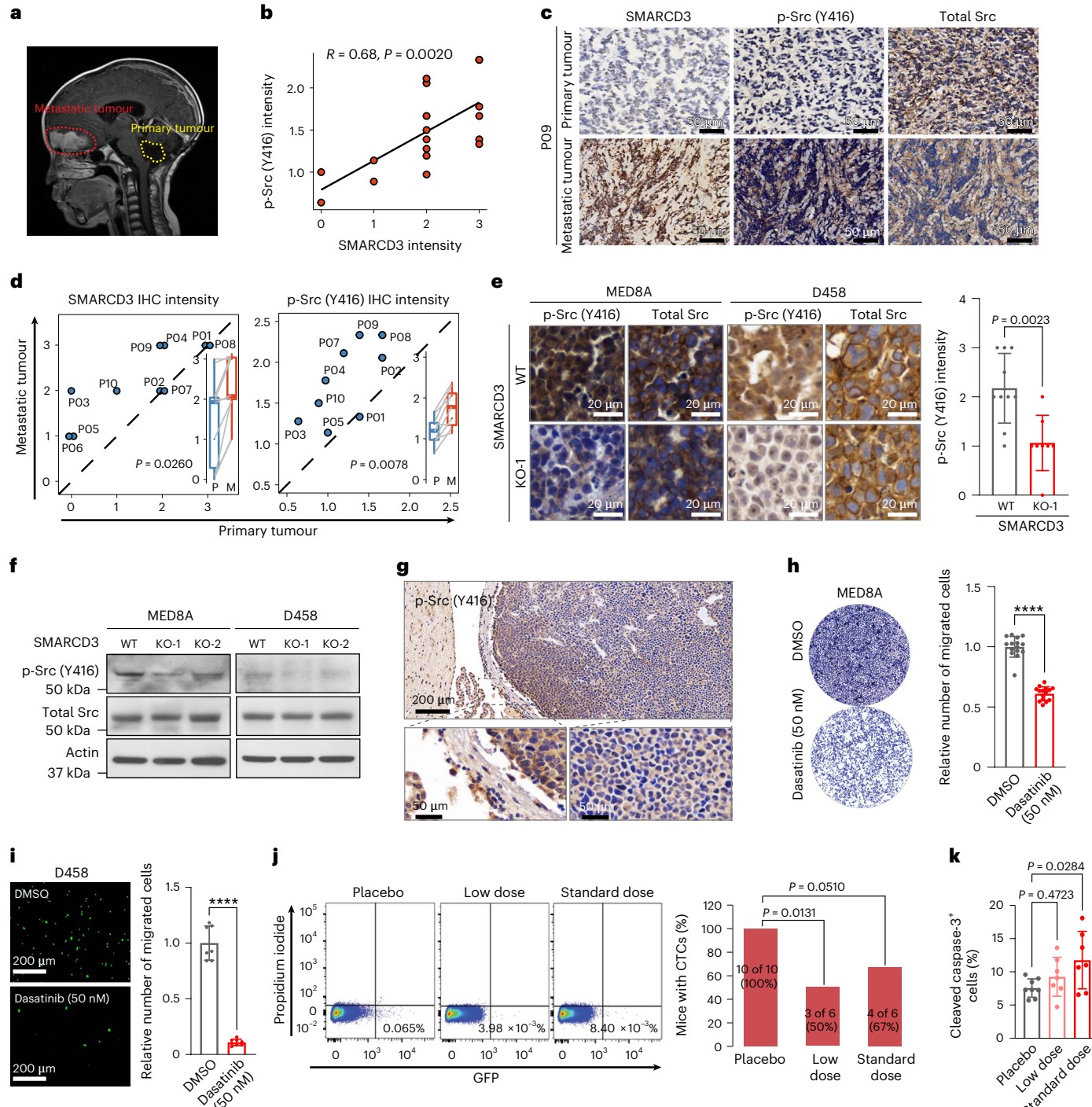

**Fig. 7 | Targeting SMARCD3–DAB1–Src activation attenuates MB metastatic dissemination. a**, Preoperative MRI sagittal image showing a patient with an enhancing metastatic tumour located at peritumoural brain oedema in the frontal lobe (red dashed line) and complete resection of the primary tumour in the cerebellum (yellow dashed line). **b**, Scatterplot showing the correlation between the IHC intensity of SMARCD3 and p-Src in MB tumours. **c**, Representative images of SMARCD3 and p-Src IHC staining in paired primary and metastatic MB samples from patient P09. **d**, Quantitative analyses of SMARCD3 and p-Src expression intensity in ten paired primary (P) and metastatic (M) MB samples. **e**, IHC (left) and quantitative analysis (right) of p-Src and total Src protein in tumours derived from mice bearing MED8A or D458 cells with SMARCD3 WT ($n = 10$) or KO ($n = 8$), respectively. **f**, IB for p-Src and total Src in MED8A and D458 cells with SMARCD3 WT or KO. **g**, Representative IHC images of p-Src in a MED8A-derived xenograft MB tumour. High-magnification images

show the tumour margin and core areas. **h,i**, Representative images (left) and quantification (right) showing cell migration of MED8A (**h**; $n_{DMSO} = 15$, $n_{Dasatinib} = 15$) and D458 (**i**; $n_{DMSO} = 7$, $n_{Dasatinib} = 8$) cells treated with DMSO or 50 nM dasatinib in Transwell assays. **j**, Flow cytometry analyses (left) and quantification (right) of GFP$^+$ CTCs from PBMCs of treated mice. **k**, IHC quantitative analysis of cleaved caspase-3 levels in tumours derived from the treated mice ($n_{Placebo} = 8$, $n_{Low dose} = 7$, $n_{Standard dose} = 7$). $n$ represents the number of biologically independent samples (**h,i**) or mouse tissues (**e,k**). Data are presented as the mean ± s.d. $P$ and $R$ values were calculated using two-tailed Spearman's rank correlation analysis (**b**), two-tailed paired $t$-test (**d**), one-tailed unpaired $t$-test (**e,h,i**), chi-square test (**j**) or one-way ANOVA with Dunnett's multiple comparison test (**k**). ****$P < 0.0001$. At least five replicates (**f**) or five mice (**g**) for each experiment were repeated independently, with similar results obtained.

In this study, SMARCD3 expression was substantially decreased at the late stage of PC development. At this time point, there is no migratory activity after birth in the human and mouse cerebellum, which is regulated by the Reelin–DAB1 signalling pathway[32,50]. These findings suggest that the SMARCD3–Reelin–DAB1 pathway acts as a modulator in the balance of 'go' and 'stop' signalling that orchestrates cerebellar development. This process is hijacked in MB metastasis, thereby implicating an important role of SMARCD3 in neurodevelopment and neurological disorders. We further defined that EZH2 and NFIX regulate SMARCD3 transcriptional activation in opposite ways through a chromatin hub. The roles of EZH2 in MB are controversial and its mechanisms of action are incompletely understood. Previous studies have reported that targeting EZH2 has significant antitumour effects in MB[51–54]. Paradoxically, inactivation of EZH2 accelerates MB development and progression by upregulating GFI1 and DAB2IP[55,56]. Besides its histone methyltransferase activity, EZH2 acts as a transcriptional co-activator in gene regulation processes involved in aggressive castration-resistant prostate cancer and in breast cancer[57–59]. NFIX, as a member of the nuclear factor I family (including NFIA and NFIB), plays a vital part in the regulation of granule precursor cell proliferation and differentiation within the postnatal cerebellum[60]. NFIB was reported to repress EZH2 expression within the neocortex and hippocampus[61], which indicates that there is negative regulation of these TFs in brain development. Our data showed that EZH2 and NFIX serve as a core set of TFs for binding to the CREs proximal to the *SMARCD3* locus to form a chromatin hub, which controls spatiotemporal gene expression in the cerebellum and MB metastasis. Our findings suggest that targeting EZH2 for MB therapy is complex and challenging, although multiple EZH2 inhibitors are currently being actively investigated in clinical trials.

This study also provided perspectives on the development of anti-metastatic therapy for MB by testing the inhibitory effects of dasatinib on tumour cell migration and metastatic dissemination. Although good tolerability of dasatinib was observed in a paediatric phase I trial for patients with leukaemia and other solid tumours[62], another phase I trial study reported that administration of dasatinib at 50 mg m$^{-2}$ twice daily resulted in poor tolerance with significant toxicities in combination with crizotinib (an oral c-Met inhibitor) in children with recurrent or progressive high-grade and diffuse intrinsic pontine glioma[63]. Failures in clinical trials for glioblastoma treatment were also observed after administering dasatinib in combination with other drugs, including erlotinib and bevacizumab[64–66]. These clinical studies indicate that targeting SFK activation may need more specific context-dependent mechanisms to exert optimal efficacy in brain tumour treatment. In this study, we identified the EZH2–NFIX–SMARCD3–Reelin–DAB1–SFK signalling pathway in the early, but not late stage, of cerebellar development. The finding that MB hijacks this cerebellum-specific developmental programme provides a strong rationale to target Src activation downstream that can selectively reduce tumour metastasis and treatment-related toxicity for children with this brain tumour.

## Online content

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

¹Xiangya School of Medicine, Central South University, Changsha, China. ²Department of Neurosurgery, Xiangya Hospital, Central South University, Changsha, China. ³Hunan International Scientific and Technological Cooperation Base of Brain Tumor Research, Changsha, China. ⁴Department of Neurological Surgery, University of Pittsburgh, Pittsburgh, PA, USA. ⁵John G. Rangos Sr Research Center, UPMC Children's Hospital of Pittsburgh, Pittsburgh, PA, USA. ⁶Department of Biological Sciences, University of Pittsburgh, Pittsburgh, PA, USA. ⁷Department of Pathology, Xiangya Hospital, Central South University, Changsha, China. ⁸Department of Radiology, Xiangya Hospital, Central South University, Changsha, China. ⁹Developmental and Stem Cell Biology Program, The Hospital for Sick Children, Toronto, Ontario, Canada. ¹⁰Division of Pediatric Neurosurgery, Ann and Robert H. Lurie Children's Hospital, Northwestern University Feinberg School of Medicine, Chicago, IL, USA. ¹¹Center for Data-Driven Discovery in Biomedicine, Division of Neurosurgery, Children's Hospital of Philadelphia, Philadelphia, PA, USA. ¹²Department of Developmental Neurobiology, St Jude Children's Research Hospital, Memphis, TN, USA. ¹³Division of Pediatric Oncology, Johns Hopkins University School of Medicine, Baltimore, MD, USA. ¹⁴Department of Pathology, Johns Hopkins University School of Medicine, Baltimore, MD, USA. ¹⁵Department of Neurology, University of Pittsburgh School of Medicine, Pittsburgh, PA, USA. ¹⁶Office of Research, University of Pittsburgh Health Sciences, Pittsburgh, PA, USA. ¹⁷UPMC Hillman Cancer Center, Pittsburgh, PA, USA. ¹⁸Department of Biomedical Informatics, University of Pittsburgh, Pittsburgh, PA, USA. ¹⁹Department of Neurobiology, University of Pittsburgh School of Medicine, Pittsburgh, PA, USA. ²⁰Department of Surgery, University of Pittsburgh School of Medicine, Pittsburgh, PA, USA. ²¹Department of Pediatrics, University of Pittsburgh School of Medicine, Pittsburgh, PA, USA. ✉e-mail: sarah.hainer@pitt.edu; mdt.cns@gmail.com; baolihu@pitt.edu

## Methods

### Cell lines and cell culture

MED8A (provided by M. D. Taylor, The Hospital for Sick Children, Toronto, Canada) and D556 (provided by D. D. Bigner, Duke University Medical Center, Durham, NC, USA) were cultured in DMEM supplemented with 20% FBS (Sigma-Aldrich, F2442); D425 and D458 (provided by S. Agnihotri, UPMC Children's Hospital of Pittsburgh, Pittsburgh, PA, USA) were cultured in IMEM supplemented with 20% FBS; and D341 (purchased from American Type Culture Collection (ATCC, HTB-187)) was cultured in EMEM supplemented with 20% FBS. The hcNSCs provided by E. H. Raabe (Johns Hopkins University School of Medicine, Baltimore, MD, USA) were cultured in NSC medium as previously described[22,49]. 293T packaging cells from ATCC (CRL-3216) were cultured in DMEM with 10% FBS.

### Mice and animal housing

Animal experiments were performed with the approval of the University of Pittsburgh Animal Care and Use Committee (protocol number 21049271). Female and male ICR SCID mice aged 4–6 weeks were purchased from Taconic Biosciences (model ICRS-F/ICRS-M). C57BL/B6 mice aged 4–6 weeks were purchased from The Jackson Laboratory (strain 000664) and were bred and maintained at the CHP Rangos Research Center under pathogen-free conditions. All mice were housed under a 12-h light–dark cycle, a temperature range of 21–23 °C and relative humidity of 55 ± 10%.

### Orthotopic MB mouse models

For orthotopic xenograft MB models, SCID mice were anaesthetized with an intraperitoneal injection of ketamine–xylazine solution (1.75 ml of 100 mg ml$^{-1}$ ketamine and 0.25 ml of 100 mg ml$^{-1}$ xylazine in 8 ml sterile water) at a dosage of 100 µl per 20 g body weight, then placed into a stereotactic apparatus equipped with a $z$-axis (Kopf). A small hole was bored in the skull 2.0 mm posterior and 2.0 mm lateral to the lambda using a dental drill. Cells ($1 \times 10^5$) infected with luciferase-ZsGreen (Addgene, 39196) lentivirus in 3 µl DPBS were injected into the right cerebellum 2.5 mm below the surface of the brain using a 10 µl Hamilton syringe with an unbevelled 30-gauge needle. For virus-induced spontaneous MB models, postnatal C57BL/B6 mice were used for the stereotactic injection of lentiviruses into the cerebellum as described above. Animals were monitored for tumour development by assessing neurological function and signs (for example, hunchback, seizure and posterior paralysis). For in vivo BLI, mice were given intraperitoneal injections of 150 µg per g D-luciferin (GoldBio, LUCK-100) and anaesthetized with 2.5% isoflurane in an induction chamber. At 10 min after injection, animals were imaged using Perkin Elmer lumina IVIS S5 systems. In vivo MRI brain imaging was carried out using a Bruker BioSpec 70/30 USR spectrometer operating at 7-Tesla field strength with the following parameters: field of view of 3.0 × 2.0 cm; acquisition matrix of 384 × 256; acquisition slice thickness of 0.60 mm; repetition time/echo time = 2,177 ms/14 ms. Mice with neurological deficits or moribund appearance were euthanized. Brains were removed after transcardial perfusion with 4% paraformaldehyde (PFA) and then fixed in 4% PFA for paraffin embedding or making OCT frozen tissue blocks.

### Treatment

Dasatinib (MedChemExpress, HY-10181) was dissolved in DMSO (75 mg ml$^{-1}$, 37.5 mg ml$^{-1}$ or 0 mg ml$^{-1}$) and 25-fold diluted with 50 mmol per litre sodium acetate buffer (pH 4.6; Sigma, S7899). Mice were treated with dasatinib at a dose of 5 µl per g body weight through oral gavage into the stomach using curved feeding needles (Kent Scientific, FNC-20-1.5-2).

### Ex vivo time-lapse imaging

Mouse brains were collected after perfusion with ice-cold HBSS and then embedded in 4% low-melting agarose diluted in HBSS. The 300 µm sagittal slices were obtained using a Leica VT1000S vibratome with a speed of 0.1 mm s$^{-1}$ and an amplitude of 1 mm. Slices were cultured on 0.4 µm culture inserts placed on MatTek glass-bottom dishes with the slice culture medium.

After 3 h in a cell culture incubator, the MatTek glass-bottom dishes were moved to a confocal microscope chamber with humidity and 5% $CO_2$. A Zeiss LSM 719 confocal microscope was used to obtain acquisitions every 6 min with a $z$-stack.

The MTrackJ plugin in ImageJ (Fiji 1.53C) was used for analysing the videos to obtain the position of one cell at a specific time point $p_n$, instantaneous distance travelled $d_n = d(p_n, p_{n+1})$, total distance travelled $d_{\text{total}} = \sum_{n=1}^{N-1} d_n$, net distance travelled $d_{\text{net}} = d(p_1, p_N)$, instantaneous trajectory time $\Delta t$ and total trajectory time $t_{\text{total}} = (N-1)\Delta t$. The instantaneous speed $s_n = d_n/\Delta t$, average speed $s_a = d_{\text{total}}/t_{\text{total}}$, velocity $v = d_{\text{net}}/t_{\text{total}}$ and directionality $D = d_{\text{net}}/d_{\text{total}}$ were calculated using the calculated variables in MTrackJ.

### Lentivirus production and transduction of target cells

The expression vectors were generated by cloning the respective open reading frame into a pLenti6.3 vector using the Gateway Cloning system. The lentiviral CRISPR–Cas9 vectors were generated by ligating the oligonucleotides of sgRNA sequences (Supplementary Table 5) into lentiCRISPRv2-Blast (Addgene, 83480) or lentiGuide-Puro (Addgene, 52963) and then validated by Sanger DNA sequencing. Gene expression was validated by RT–qPCR (primers listed in Supplementary Table 6) or immunoblotting in lentivirus-infected target cells. For enhancer deletion, genomic DNA was extracted (New England BioLabs, T3010S) and amplified by PCR (ApexBio, K1025), then gel purified (Qiagen, 28704) and sequencing validated. Genotyping PCR primers are listed in Supplementary Table 7. Lentiviruses were produced in 293T cells with a packaging system (pCMVR8.74, pMD2.G, pRSV-Rev) per the vendor's instruction.

### MTS assay and BrdU assays

For the MTS assay, 5,000 cells were seeded into a 96-well plate with 150 µl medium. Then, 30 µl of the combined MTS–PMS solution (Promega, G5430) was pipetted into each well. After 2 h of incubation, 100 µl medium out of the total 150 µl medium was pipetted into a new 96-well plate, and absorbance at 490 nm was measured using a BioTech Synergy HTX. For the bromodeoxyuridine (BrdU) assay, cells were incubated overnight at 4 °C with anti-BrdU antibody after being incubated in medium containing BrdU for 1 h, fixation, HCl incubation and blocking. A fluorescence-conjugated antibody was used to visualize the anti-BrdU-labelled cells.

### Scratch wound-healing assay and Transwell assay

Cells were seeded into a 12-well plate and allowed to reach 95% confluence. Wounds were made with pipette tips, and images were captured at specific time points and analysed using ImageJ. For time-lapse imaging, cells were seeded on MatTek dishes and wounds were made with pipette tips at 95% confluency. The MatTek dishes were moved to a confocal microscope chamber with humidity and 5% $CO_2$. A Zeiss LSM 719 confocal microscope was used for imaging acquisition as described for ex vivo brain slices.

Transwell assays were performed in Falcon 24-well insert systems (8.0 µm pore sizes). After 6 h of starvation, cells were seeded in Transwell inserts at $1 \times 10^6$ cells per well in medium without FBS or with dasatinib, and the inserts were transferred into medium containing FBS or dasatinib. After 24 h or 48 h of incubation, cells were fixed and stained using a Hema 3 stain set (Fisher Scientific, 22-122911) or directly stained with calcein AM (BD Biosciences, 564061).

### Immunoblotting (IB), IHC and immunofluorescence (IF)

For IB, cells were collected, washed with PBS, lysed in RIPA buffer (Millipore, 20-188) with protease and phosphatase inhibitor mini-tablet

(Thermo Fisher, A32961) and centrifuged at 10,000$g$ at 4 °C for 15 min. Protein lysates were subjected to SDS–PAGE on a 12% gradient poly-acrylamide gel, transferred onto polyvinylidene fluoride membranes, which were incubated with the indicated primary antibodies, washed and probed with horseradish peroxidase (HRP)-conjugated secondary antibodies. For IHC staining, brain sections were incubated with the indicated primary antibodies overnight at 4 °C after deparaffiniza-tion, rehydration, antigen retrieval (Vector Laboratories, H3300), quenching of endogenous peroxidase and blocking. The sections were incubated with HRP-conjugated horse anti-rabbit IgG polymer (Vector Laboratories, MP-7401) for 1 h, and then diaminobenzidine using DAB substrate (Vector Laboratories, SK-4105) for 1–15 min at room temperature, followed by haematoxylin staining. Images were acquired using a Nikon Eclipse E800 and scanned using DigiPath's digital pathology scanner. For IF staining, mouse brains were isolated and fixed in 4% PFA overnight and then processed for OCT frozen tissue blocks. OCT frozen brain sections were thawed at room temperature for 30 min, rinsed and rehydrated with PBS 3 times. After blocking with PBS buffer containing 10% FBS, 1% BSA and 0.3% Triton, the sections were incubated with the indicated primary antibodies overnight at 4 °C following species-appropriate secondary antibodies coupled to AlexaFluor dyes (594 or 647, Invitrogen) for 1 h at room temperature. Vectashield with DAPI (Vector Laboratories, H-1500) was used to mount coverslips. Images were acquired using a Leica DMI8 microscope and analysed using ImageJ. Information about antibodies used for these assays are described in Supplementary Table 8.

### Metastasis evaluation by direct fluorescence

The presence or absence of metastatic deposits was observed under a direct fluorescence stereoscope (Leica M165FC). Images were acquired with consistent exposure settings during the experiments. The spines of mice were defined as positive if a single metastatic deposit was observable.

### Flow cytometry and FACS

Mice showing neurological signs of late-stage brain tumours or deemed end point were killed, and blood was collected through cardiac exsan-guination under deep general anaesthesia. The collected blood (500–900 µl) was swiftly prepared for flow cytometry using RBC lysis buffer (Invitrogen, 00-4333), and cells were suspended in ice-cold PBS with 1% BSA and 2 mM EDTA. After incubation with propidium iodide (Thermo Fisher, P3566) in the dark, the stained cells were analysed using a BD Fortessa analyser. FACS was performed using a BD FACSAria cell sorter. Data were analysed using FlowJo software (v.10.6.1).

### RNA isolation, RT–qPCR and RNA-seq

RNA was isolated using a RNeasy Plus Mini kit (Qiagen, 74134) and then used for first-strand cDNA synthesis (Invitrogen, 28025-013). RT–qPCR was performed using PowerUp SYBR Green master mix (Applied Biosys-tems, A25742). The relative expression of genes was normalized using ribosomal protein L39 (*RPL39*) as a housekeeping gene.

For RNA-seq, sequencing libraries were generated using a NEBNext Ultra RNA Library Prep kit for Illumina following the manufacturer's recommendations, and index codes were added to attribute sequences to each sample. After cluster generation, the library preparations were sequenced using a NovaSeq 6000 platform, and paired-end reads were generated. Reads were aligned using Hisat2 (v.2.1.0) against the hg38 genome and transcriptome. After initial mapping, the aligned reads were filtered out if their best placements were only mapped to unique genomic coordinates. The statistical environment R was used to perform all the statistical analyses and graph plots.

### Analyses of scRNA-seq and sci-ATAC-seq3 data

For scRNA-seq analysis, genes not expressed in any cells had already been removed. Cells with fewer than 200 genes or more than 5,000 genes expressed or more than 10% mitochondrial genes expressed were removed using Seurat (v.3.2.3). Clusters gener-ated using UMAP were assigned to cell types using known marker genes. For sci-ATAC-seq3, processed data were directly downloaded. Co-accessibility scores and Cicero gene activity scores were calculated using Cicero (v.1.6.2). Data were visualized using Sushi. Cellranger (v.5.0.1) was also used to analyse the scRNA-seq and sci-ATAC-seq3 data.

### ATAC-seq

Approximately 100,000 nuclei were extracted from freshly collected MED8A cells with SMARCD3 WT or KO by incubating for 15 min on ice in lysis buffer (10 mM Tris-HCl, pH 7.5, 10 mM NaCl, 3 mM MgCl$_2$, 0.1% NP-40, 0.1% Tween-20 and 0.01% digitonin). Samples were washed in 1 ml wash buffer (10 mM Tris-HCl, pH 7.5, 10 mM NaCl, 3 mM MgCl$_2$ and 0.1% Tween-20) and centrifuged at 500$g$ for 10 min at 4 °C. The supernatant was removed, and nuclei pellets were flash-frozen in liquid nitrogen and stored at −80 °C.

Nuclei were incubated in transposition reaction mix using 50 µl 2× tagmentation buffer (Diagenode) and 5 µl preloaded tagmentase (Diagenode) for 30 min in a thermomixer set to 37 °C at 1,000 r.p.m. Following transposition, DNA was isolated using a Qiagen MinElute Reaction Cleanup kit. Samples were PCR amplified using NEBNext High-Fidelity 2× PCR master mix and 25 µM of Nextera 70* and 25 µM Nextera 50* primers. Samples were amplified using five cycles of PCR, then quantified by qPCR and amplified using three additional cycles. Samples were run on 1.5% agarose gel, and 150–500 bp bands from each sample lane were extracted. Libraries were run on a Fragment Analyzer according to the manufacturer's instructions (Agilent) to validate library quality. Libraries were pooled and sequenced on using an Illumina NextSeq500.

### CUT&RUN

The CUT&RUN protocol was performed as previously described[36,37,67] with the following modifications.

MED8A cells with SMARCD3 WT or KO were diluted to 1 million cells in PBS. Cells were centrifuged and resuspended in 1 ml cold nuclear extraction buffer (20 mM HEPES-KOH, pH 7.9, 10 mM KCl, 0.5 mM spermidine, 0.1% Triton X-100, 20% glycerol and freshly added pro-tease inhibitors) and incubated for 10 min on ice. Nuclei were then centrifuged, and pellets were resuspended in 600 µl of nuclear extrac-tion buffer. Concanavalin A beads (400 µl bead slurry per 1 million nuclei) were prepared in binding buffer (20 mM HEPES-KOH, pH 7.9, 10 mM KCl, 1 mM CaCl$_2$ and 1 mM MnCl$_2$) and washed twice in bind-ing buffer before adding the nuclei and incubating for 10 min at 4 °C with rotation.

Following nuclei binding to concanavalin A beads, samples were pre-blocked for 5 min at room temperature using 1 ml blocking buffer (20 mM HEPES, pH 7.5, 150 mM NaCl, 0.5 mM spermidine, 0.1% BSA, 2 mM EDTA and freshly added protease inhibitors). Bound nuclei were washed in 1 ml wash buffer (20 mM HEPES, pH 7.5, 150 mM NaCl, 0.5 mM spermidine, 0.1% BSA and freshly added protease inhibitors). Following this wash, nuclei were resuspended in 2 ml wash buffer and aliquoted in 250 µl volumes to eight 1.5 ml tubes for the individual antibody reactions (125,000 nuclei per antibody sample for each cell line). Nuclei were incubated for 1 h at room temperature with rotation with the primary antibody in wash buffer to a final concentration of 1:100. Negative controls were included for each cell line, in which no primary antibody was added. Following incubation, samples were washed twice in 1 ml wash buffer.

Samples were incubated with 2.4 µl in-house purified pA-MNase per sample in 250 µl wash buffer for 30 min at room temperature with rotation. Samples were pre-equilibrated to 0 °C in an ice water bath for 5 min before 3 mM CaCl$_2$ was added to initiate MNase digestion. Following a 30 min digestion in an ice water bath, the digestion reaction

was chelated using 2XRSTOP+ buffer (200 mM NaCl, 20 mM EDTA, 4 mM EGTA, 50 μg ml$^{-1}$ RNase A, 40 μg ml$^{-1}$ glycogen and 10 pg ml$^{-1}$ MNase-digested *Saccharomyces cerevisiae* mononucleosomes added for a spike-in control). Samples were incubated at 37 °C for 20 min and centrifuged to separate and release fragments. Protein was digested with ProK, and DNA was purified using PCI extraction and ethanol precipitation.

DNA libraries were prepared using end-repair, adenylation and NEBNext stem-loop adapter ligation. Fragments were then purified using AMPure XP beads (Beckman Coulter) and amplified using 15 cycles of high-fidelity PCR. A final AMPure clean-up step was performed to purify the DNA fragments before sequencing. Samples were run on 1.5% agarose gels to validate library quality before sequencing. Libraries were pooled and sequenced using an Illumina NextSeq500.

### Defining accessible sites
Reads in ATAC-seq and CUT&RUN were mapped to the hg38 reference genome using bowtie2 (v.2.3.5.1) with the options "−very-sensitive -X 2000" and "−very-sensitive -X 2000 −dovetail", respectively. PCR duplicates were removed using sambamba.

For ATAC-seq, using MACS3 (v.3.0.0a6) call peak with the options "-f BEDPE -B -q 0.01", reads with a fragment size between 1 and 100 or between 180 and 247 were used to define peaks of accessibility across all sites. ChIPseeker was used to annotate the peaks.

For CUT&RUN, using MACS3 call peak with the options "−broad −broad-cutoff 0.1", reads with a fragment size between 150 and 500 were used to define peaks of histone-marker-binding sites. Inputs were used as a control for peak calling.

Peaks and alignments were visualized using IGV (V.2.6.3).

### Pathology analysis of patient tumour samples
The tissue microarray MB slides (formalin-fixed paraffin-embedded) for IHC were provided by C. G. Eberhart (Johns Hopkins University School of Medicine, Baltimore, MD, USA), approved by the institutional review board (protocol number NA_00015113). The ten paired primary and metastatic MB MRI images and slides (formalin-fixed paraffin-embedded) for IHC from Xiangya Hospital were used and analysed with approval by the institutional review board (number 202110207). Informed consent was obtained for the biorepositories that provided the above study materials. The pathology analysis of MB samples was conducted by at least two experienced neuropathologists. The study was compliant with all ethical regulations.

### Statistics and reproducibility
All the boxplots show the interquartile range (IQR), whiskers denote quartile 3 + 1.5× the IQR or quartile 1 − 1.5× the IQR. Data points that were more or less than the whiskers were considered outliers. Column bar plots show the mean with standard deviation. Statistical parameters, including the exact value of $n$, the definition of centre, dispersion, precision measures, statistical test and statistical significance, are reported in the figures and figure legends. Data were judged to be significant when $P < 0.05$. No statistical methods were used to predetermine the sample sizes, but our sample sizes were similar to those reported in the previous publications[49,68]. In the experiments with dasatinib treatment for mice bearing MB, tumour sizes were assessed, and then mice were grouped to minimize variations in tumour size among the groups. No randomization was performed for other experiments. The investigators were blinded to assess protein expression in IHC and IF experiments; other data collection and analyses were not performed blind to the conditions of the experiments. No animals or data were excluded except for low-quality cells during scRNA-seq analysis. GSVA (v.1.36.3) and IPA (v.01-16) were used to calculate the meta-proliferating cell nuclear antigen scores and pathway analysis scores, respectively. R (v.3.5.1) and GraphPad Prism (v.9.1.0) were used for statistical analyses.

### Reporting summary
Further information on research design is available in the Nature Portfolio Reporting Summary linked to this article.

### Data availability
The RNA-seq, ATAC-seq and CUT&RUN data that support the findings of this study have been deposited into the Gene Expression Omnibus (GEO) under accession code GSE194217. Previously published data that were re-analysed here are available from the following sources: transcriptomics of 1,350 MB samples and 291 healthy cerebellum samples (GEO: GSE124814); scRNA-seq data of 25 MB samples (GEO: GSE119926); expression profiles and clinical data of 763 MB samples (GEO: GSE85217); Hi-C data of mouse cerebellum (GSE138822); scRNA-seq data of developing mouse cerebellum (European Nucleotide Archive: PRJEB23051); ChIP-seq data of 5 MB samples (GEO: GSE92585); sci-ATAC-seq3 data of foetal cerebellum (GEO: GSE149683); ChIP-seq data of D458 and D425 (GEO: GSE129521); proteomics data of 45 MB samples from a previous publication[19]; 167 MB RNA-seq data from R2 (https://r2.amc.nl); processed The Cancer Genome Atlas pan-cancer RNA-seq data from Xena[69]; gene profiling of healthy human tissues from GTEx (https://www.gtexportal.org/home/); human cerebellum scRNA-seq data from the Human Cell Atlas (https://www.covid19cel-latlas.org/aldinger20); ChIP-seq data of mouse cerebellum from the ENCODE portal (https://www.encodeproject.org/); and H3K27ac ChIP-seq data of 4 MB subgroups from St Jude Cloud Visualization Community (https://viz.stjude.cloud/). All other data supporting the findings of this study are available from the corresponding authors on reasonable request. Source data are provided with this paper.

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

### Acknowledgements

We thank R. A. DePinho and X. Wu for critically evaluating the manuscript; E. Jane, P. Daniel and D. Yimlamai for their assistance with reagents; J. Dai, X. Lin, X. Lin, T. Wu, M. Wu, J. Hu, K. Peng, Y. Li, Y. Zhang, J. Wang and D. Xing from Central South University, and X. Zheng from UPMC Children's Hospital of Pittsburgh for their technical support; J. J. Michel, M. L. Mulkeen, M. Airik, K. Prasadan, Y. Wu, A. C. Poholek, W. A. MacDonald and R. Elbakri from the core facilities at the Rangos Research Center for their assistance with flow cytometry, microscopy, mouse imaging and sequencing analyses. We gratefully acknowledge funding support from the Matthew Larson Foundation (to B.H.), the Connor's Cure Fund from the V Foundation (to B.H.), the Andrew McDonough B+ Foundation (to B.H.), the Scientific Program Fund from the Children's Hospital of Pittsburgh (to B.H.), NIH/NINDS 1R21NS125218-01 (to B.H.) and NIGMS R35GM133732 (to S.J.H.). This research was supported in part by the University of Pittsburgh Center for Research Computing through the resources provided. H.Z. is a University of Pittsburgh-affiliated visiting research scholar supported by CSC and Central South University.

### Author contributions

Conceptualization: H.Z., B.P. and B.H.; Methodology: H.Z., E.E.B., A.C., M.L., S.J.H. and B.H.; Investigation: H.Z., B.P., E.E.B., J.Q., B.X., E.A., V.R., V.S., Y.G., Z.L. and W.F.-M.; Data curation and analysis: H.Z., J.Q., D.A.,

S.X., F.L., O.S. and Z.H.; Resources: S.W., G.X., T.T., A.C.R., S.C.M., E.H.R., C.G.E., D.S., S.A., G.K., S.L., J.N.R., G.K.G., R.M.F. and M.D.T.; Writing original draft: H.Z. and B.H.; Writing, review and editing: B.E.S., K.H., A.B., I.F.P. and S.J.H.; Supervision: X.L., I.F.P., R.M.F., S.J.H., M.D.T. and B.H.

## Competing interests

The authors declare no competing interests.

## Additional information

**Extended data** is available for this paper at

**Correspondence and requests for materials** should be addressed to Sarah J. Hainer, Michael D. Taylor or Baoli Hu.

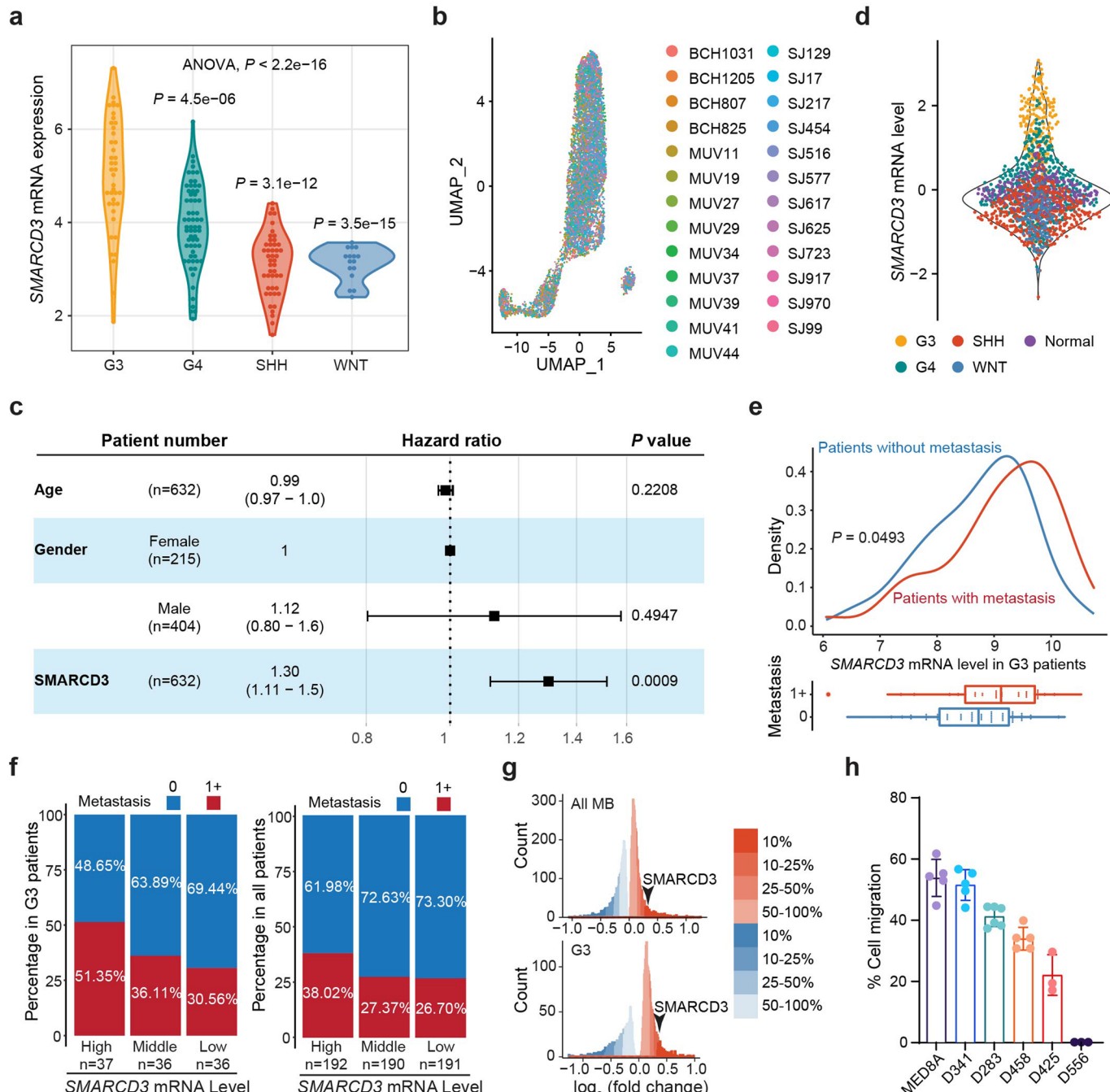

**Extended Data Fig. 1 | SMARCD3 expression and association with MB metastasis. a**, Violin plot showing expression levels of *SMARCD3* mRNA in four MB subgroups in the RNAseq dataset ($n_{G3} = 41$, $n_{G4} = 64$, $n_{SHH} = 46$, $n_{WNT} = 16$). **b**, UMAP visualization of SMARCD3 expression in each cell in the scRNAseq data. **c**, Multivariable forest plot showing the significance of age, gender, and *SMARCD3* mRNA expression level for overall survival in a multivariable model. **d**, Violin plot showing expression levels of *SMARCD3* mRNA in four MB subgroups and normal cerebellum. **e**, Density plot and boxplot showing the association between metastasis status (0, no metastasis; 1 +, metastasis at diagnosis) and expression levels of *SMARCD3* mRNA in primary G3 MB samples ($n_0 = 43$, $n_{1+} = 66$) only. **f**, Bar diagrams showing the percentage of primary tumors without

metastasis (0) or with metastasis (1 + ) in three groups with different expression levels of *SMARCD3* mRNA (high, middle, low) in G3 MBs only and all subgroups of MBs, respectively. **g**, Histograms showing the number of differentially expressed genes between patients with metastasis and without metastasis by the log$_2$(fold change). The arrows denote where SMARCD3 is located. **h**, Assessment of cell migration for 6 MB cell lines by Transwell assay ($n_{MED8A} = 5$, $n_{D341} = 5$, $n_{D283} = 6$, $n_{D458} = 5$, $n_{D425} = 3$, $n_{D556} = 3$). One dot represents one cell (**b**), or one patient sample (**a**, **d**, **e**). *n* represents the number of human patients (**a**, **e**, **f**) or biologically independent samples (**h**). Data are presented as the mean ± s.d. (**c**, **h**). *P* values were calculated by two-tailed Welch's *t*-test with FDR correction (**a**, **e**) or the two-tailed Wald test (**c**).

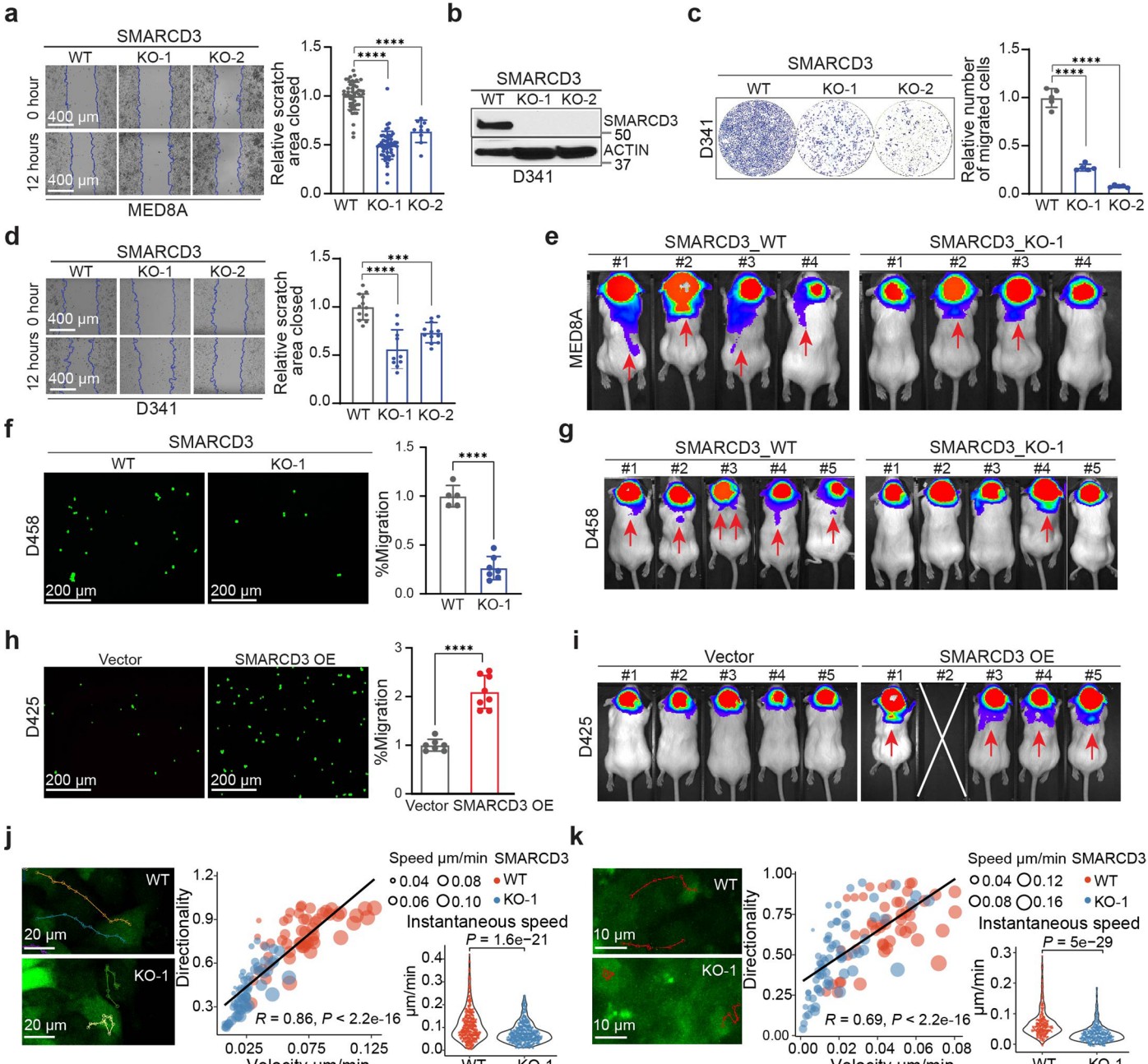

**Extended Data Fig. 2 | Altering SMARCD3 expression influences MB cell migration and metastatic dissemination.** Time-lapse images and quantification of cell migration of MED8A (**a**) with SMARCD3 WT (*n* = 52), KO-1 (*n* = 52), and KO-2 (*n* = 10); and D341 (**d**) with SMARCD3 WT (*n* = 12), KO-1 (*n* = 10), and KO-2 (*n* = 12) by scratch-wound healing assays. **b**, IB for SMARCD3 in D341 cells with control (WT) and SMARCD3 KO-1 and KO-2. Transwell images and quantification of cell recruitment in D341 (**c**) with SMARCD3 WT, KO-1, and KO-2 (*n* = 5 for each group); D458 (**f**) with SMARCD3 WT (*n* = 5) and KO-1 (*n* = 7); and D425 (**h**) with vector (*n* = 7) and SMARCD3 OE (*n* = 8). Bioluminescence images of mice bearing MED8A (**e**) and D458 (**g**) cells with SMARCD3 WT *vs* KO-1 at day 18 and endpoint after implantation respectively; and D425 cells (**i**) with vector and SMARCD3 OE at day 26 after implantation (one mouse died before imaging scan).

Time-lapse imaging for MED8A cell migration with SMARCD3 WT *vs* KO-1 by scratch-wound healing (**j**) or *ex vivo* brain slice assays (**k**); the color lines show cell tracks; scatterplot shows the correlation between cell migration velocity and directionality; violin plot shows the instantaneous motility speed (one dot shows one cell at a one-time point). Each dot represents one cell and the dot size represents the average motility speed of one cell (scatterplots in **j**, **k**). *n* represents the number of the biologically independent samples from at least 2 independent experiments; data are presented as the mean ± s.d. (**a**, **c**, **d**, **f**, **h**). *P* and *R* values were calculated using one-way ANOVA with Dunnett's multiple comparison test (**a**, **c**, **d**), one-tailed unpaired *t*-test (**f**, **h**), or two-tailed Welch's *t*-test with FDR correction and two-tailed Pearson's correlation analysis (**j**, **k**). ∗∗∗*P* = 0.0002, ∗∗∗∗*P* < 0.0001.

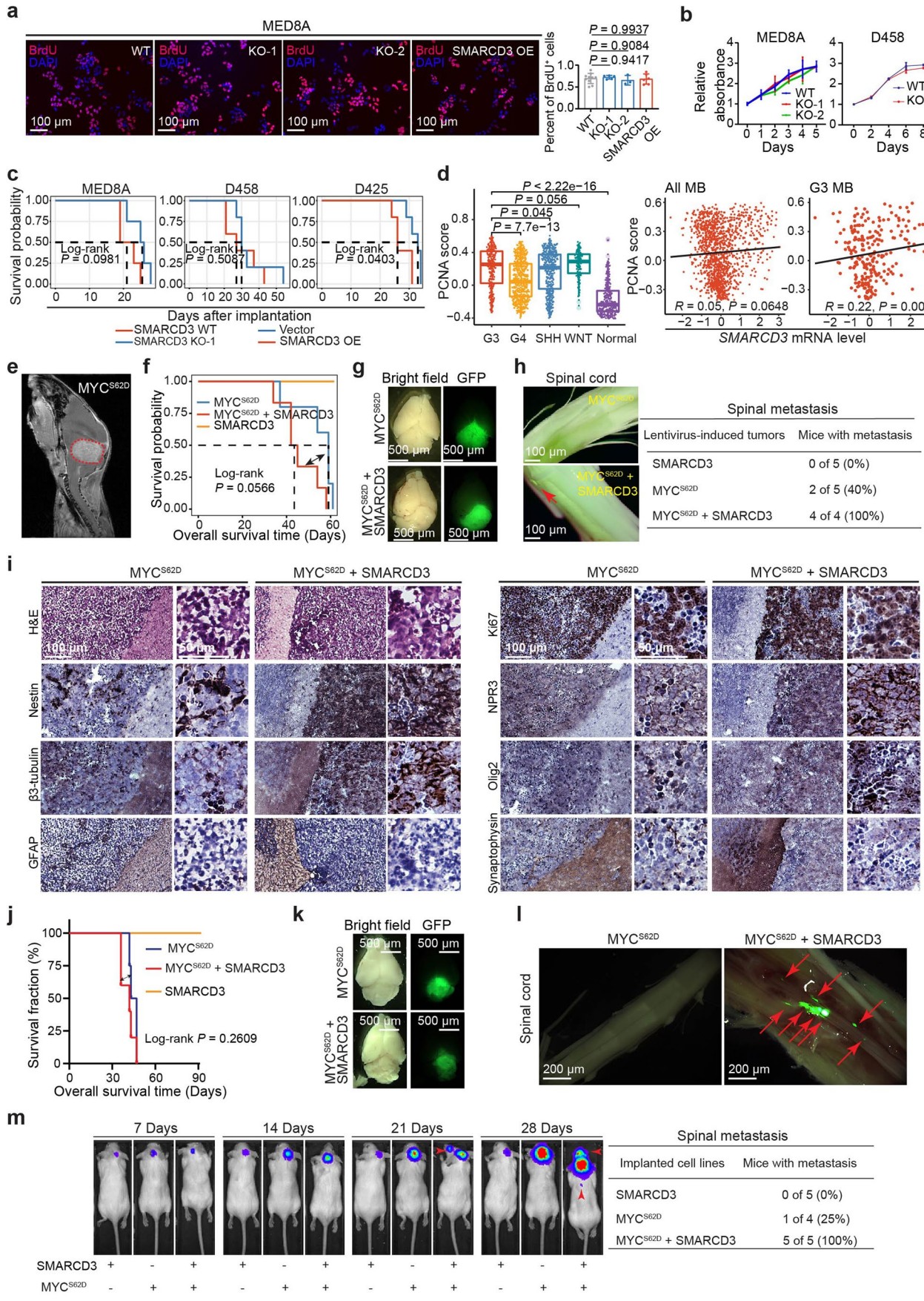

**Extended Data Fig. 3 | See next page for caption.**

**Extended Data Fig. 3 | SMARCD3 influences tumor metastasis but not tumor development and growth. a**, IF images and quantification of bromodeoxyuridine (BrdU) in MED8A cells with SMARCD3 WT ($n = 10$), KO ($n_{KO-1} = 5$, $n_{KO-2} = 3$), and OE ($n = 6$). **b**, Cell proliferation (MTS assay) of MED8A ($n = 9$) and D458 ($n = 3$) with SMARCD3 WT vs KO. **c**, Kaplan-Meier survival of SCID mice bearing indicated tumor cells. **d**, PCNA signature score levels in G3, G4, SHH, WNT, and normal bulk samples (boxplot); and correlations with *SMARCD3* mRNA expression (scatterplots) in all MBs ($n = 1,280$) and G3 ($n = 233$). **e**, A MRI image of the cerebellar tumor induced by lentivirus carrying MYC^S62D in the C57BL/6 J mouse. **f**, Kaplan-Meier survival of C57BL/6 J mice intracranially infected by lentivirus-mediated indicated gene expression. **g**, Representative bright-field and fluorescence images of the mouse brains bearing indicated tumors. **h**, The fluorescence images and pie charts showing the spinal cords from mice bearing indicated tumors. **i**, H&E and IHC images showing histopathological assessment of MYC^S62D- and MYC^S62D + SMARCD3-induced tumors for the indicated marker expression. **j**, Kaplan-Meier survival of SCID mice intracranially implanted by hcNSCs expressing indicated genes. **k**, Representative bright-field and fluorescence images of the SCID mouse brains bearing indicated tumors. Representative fluorescence images of the spinal cords (**l**) or luminescence images/pie charts showing spinal metastasis (**m**) from SCID mice bearing indicated tumors. The n number represents the biologically independent samples (**a**, **b**) or patient samples (**d**), and data are presented as the mean ± s.d. (**a**, **b**). *P* and *R* values were calculated using one-way ANOVA with Dunnett's multiple comparison test (**a**), or two-tailed Welch's t-test with FDR correction (boxplot in **d**), two-tailed Spearman's rank correlation analysis (scatterplots in **d**), and log-rank test (**f**, **j**). The representative images from 4 independent mice were repeated with similar results (**g**, **h**, **i**, **k**, **l**).

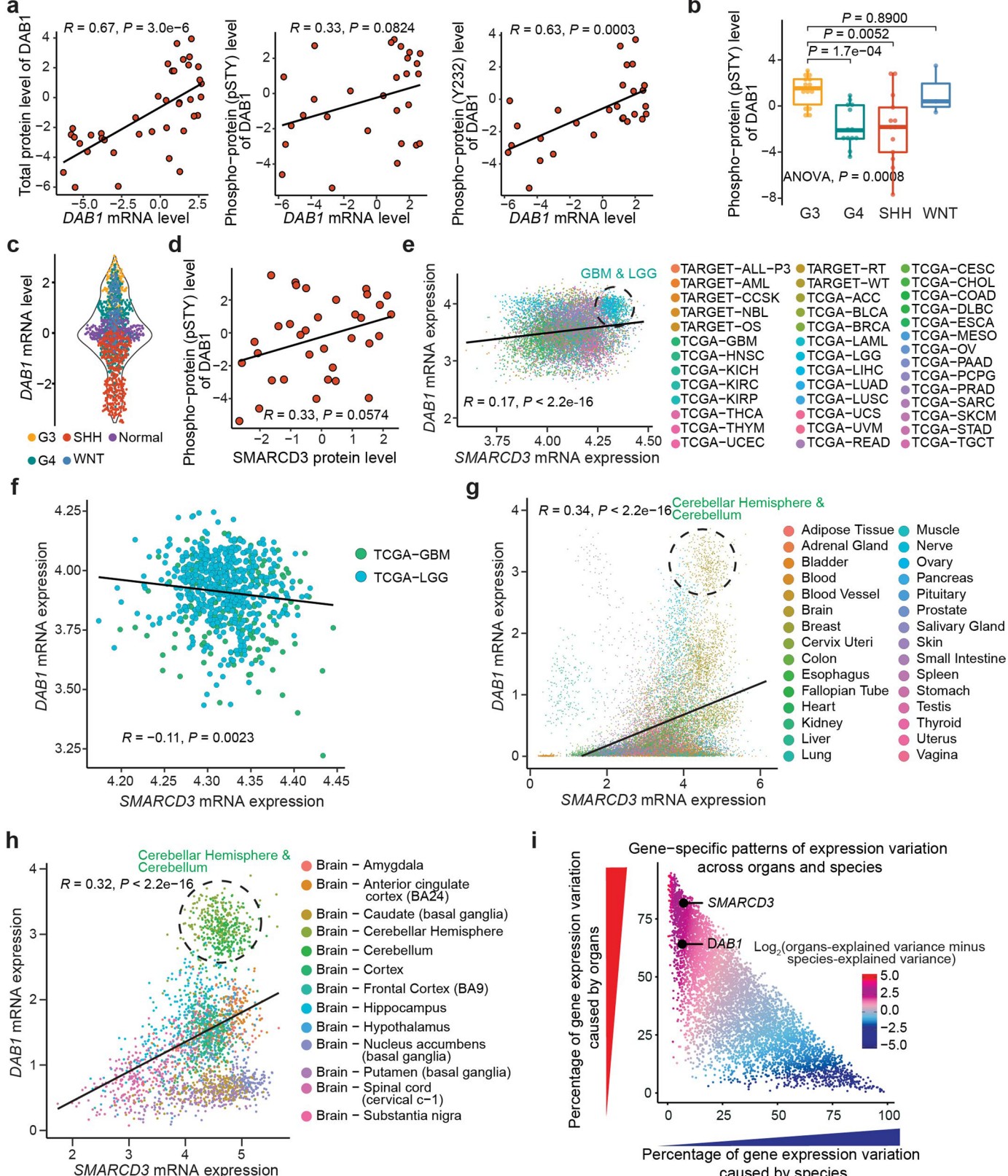

**Extended Data Fig. 4 | See next page for caption.**

**Extended Data Fig. 4 | The association between SMARCD3 and DAB1 is evolutionarily conserved in the cerebellum. a**, Scatterplots showing the correlation of expression levels between *DAB1* mRNA and the total DAB1 protein, phospho-DAB1 (pSTY), or phospho-DAB1 (Y232) in all MB tumors ($n = 45$). **b**, Boxplot showing the levels of phospho-DAB1 (pSTY) protein expression in the MB proteomics dataset ($n_{G3} = 14$, $n_{G4} = 13$, $n_{SHH} = 13$, and $n_{WNT} = 3$). **c**, Violin plot showing *DAB1* mRNA expression in four MB subgroups and normal cerebellum. **d**, Scatterplot showing the correlation between SMARCD3 protein expression levels and the phospho-DAB1 (pSTY) protein levels in all MBs. **e, f**, Scatterplot showing the correlation between *SMARCD3* and *DAB1* mRNA expression levels in all cancer types and gliomas using the datasets from the TARGET and TCGA projects. The dashed line outlines the glioblastomas and low-grade gliomas. **g**, Scatterplot showing the correlations between *SMARCD3* mRNA expression

levels and *DAB1* mRNA expression levels in normal tissues using the datasets from the GTEx projects. **h**, Scatterplot showing the correlations between *SMARCD3* and *DAB1* mRNA expression levels in brain tissues only. Dashed line circles highlight the tissues from the cerebellum and cerebellar hemisphere (**g, h**). **i**, Percentage of gene expression variance explained by species and by organs. Data from the reference[25] were used for analysis, including six organs (brain, cerebellum, heart, kidney, liver, and testis) in seven vertebrate species (chicken, chimpanzee, human, mouse, opossum, platypus, and rhesus). Each color dot represents one tumor sample (**a-f**), a normal tissue sample (**g-h**), or a gene (**i**). The red dot is explained mostly by organ differences, while the blue dot is by species differences (**i**). *P* and *R* values were calculated using two-tailed Welch's *t*-test with FDR correction (**b**) and/or two-tailed Spearman's rank correlation analysis (**a, d, e, f, g, h**).

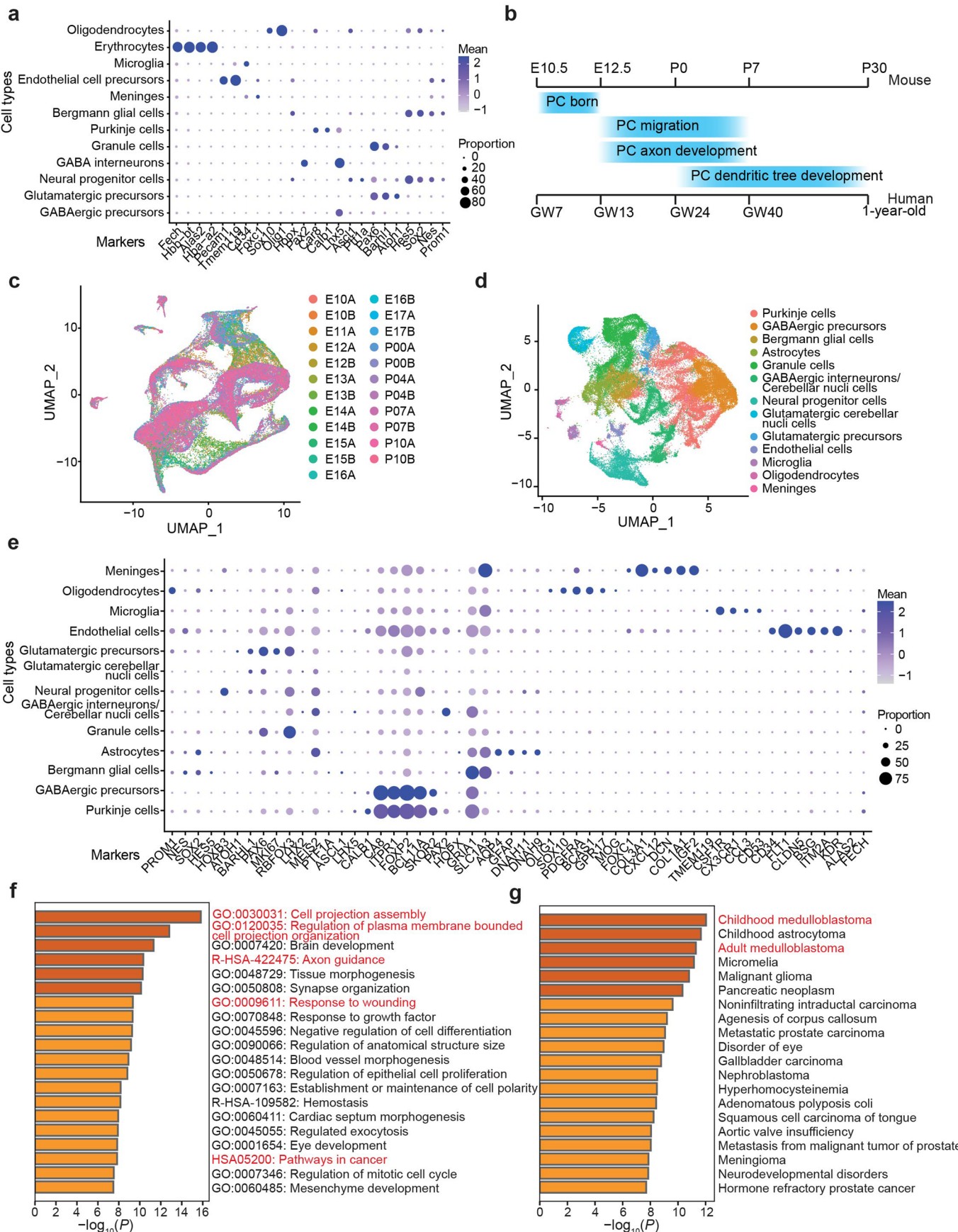

**Extended Data Fig. 5 | See next page for caption.**

**Extended Data Fig. 5 | Characterization of SMARCD3 expression during human and mouse cerebellar development using scRNAseq datasets.**
**a**, Dotplot showing the expression of each selected marker gene per cell type in mouse cerebellum. **b**, The schematic diagram shows the timeline of Purkinje cell development in both humans and mice. **c**, UMAP visualization of the developing mouse cerebellum. **d**, UMAP visualization and marker-based annotation of cell types from the developing human cerebellum. **e**, Dotplot showing the expression of each selected marker gene per cell type in the human cerebellum. **f**, The top 20 biological pathways are enriched by Gene-ontology analysis of the SMARCD3-positively correlated genes in the human cerebellum. **g**, The top 20 diseases are enriched in DisGeNET by analyzing the SMARCD3-positively correlated genes in the human cerebellum. Each dot represents one cell (**c**, **d**). Dots with the same color come from the same mouse cerebellum (**c**) or the same cell type (**e**). *P* value from the multiple Fisher test is corrected using two-tailed Benjamini-Hochberg method (**f**, **g**).

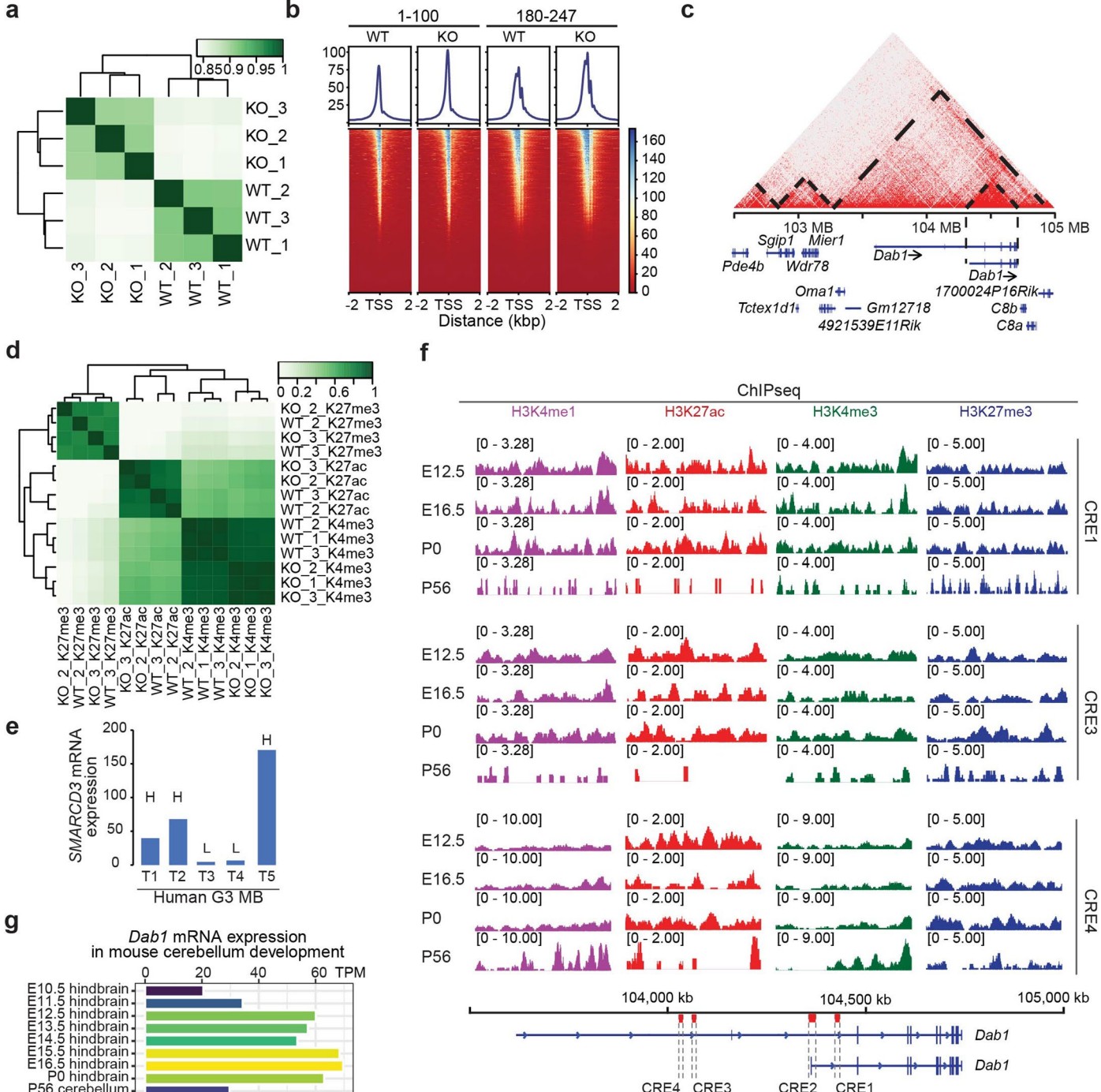

**Extended Data Fig. 6 | SMARCD3 modulates chromatin architecture for gene regulation. a**, Pearson correlation analysis of ATACseq replicates of MED8A cells with SMARCD3 KO *vs* WT. **b**, Mean accessibility and enrichment of ATACseq data with different insert lengths around transcription start site (TSS). **c**, Hi-C chromatin interaction map on a region centered in the *Dab1* gene of mouse cerebellum (P22). The dashed lines outline TAD borders. **d**, Pearson correlation analysis of histone marker binding signals from CUT&RUN replicates of MED8A cells with SMARCD3 KO *vs* WT. **e**, Histogram of *SMARCD3* mRNA expression

(FPKM) in 5 G3 patient samples. Each tumor (T) is marked with higher (H), or lower (L) levels of SMARCD3 expression. **f**, Histone modification signals at CRE1, CRE3, and CRE4 at the locus of the *Dab1* gene based on analyzing ChIPseq data of mouse hindbrain or cerebellum samples. These CREs in the mouse are homologous to the human CREs of the *DAB1* gene. **g**, Histogram of *Dab1* mRNA expression (TPM) during mouse cerebellar development at indicated time points.

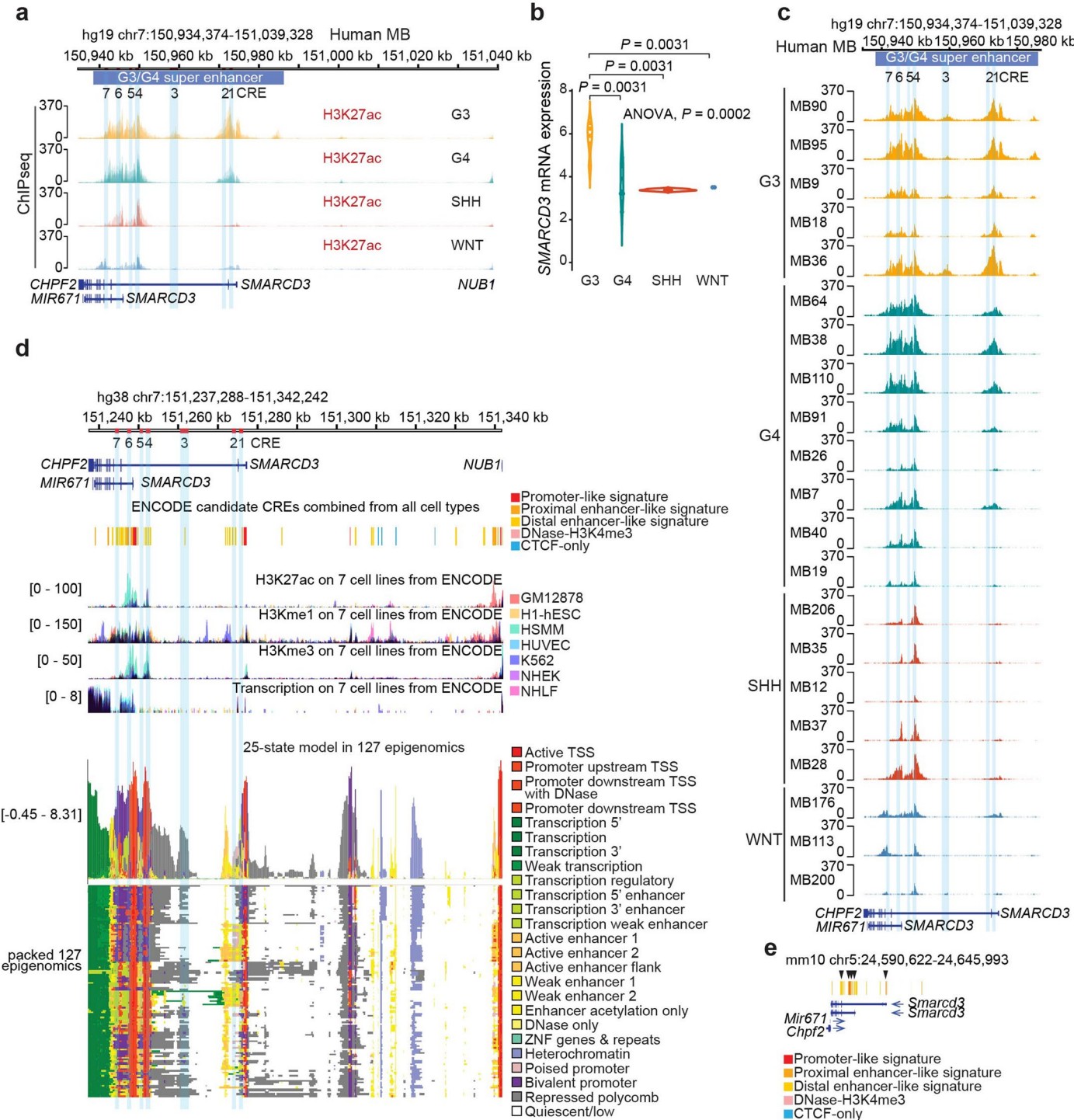

**Extended Data Fig. 7 | The newly-identified CREs at the *SMARCD3* gene locus in the primary human MB and combined cell and tissue types from the public enhancer databases. a**, H3K27ac binding signals at the *SMARCD3* gene locus based on analyzing ChIPseq data of 4 subgroup MB from patients. **b**, Violin plot showing *SMARCD3* mRNA levels of each subgroup of tumors. **c**, H3K27ac binding signals at the *SMARCD3* gene locus based on analyzing ChIPseq data of each MB. **d**, CREs and histone modifications at the human *SMARCD3* gene locus identified by ENCODE Data Analysis Center and Roadmap Epigenomics Project. **e**, CREs at the mouse *Smarcd3* gene locus identified by ENCODE Data Analysis Center. Each dot represents one bulk sample (**b**). The color of the peak represents the subgroup (**a**, **c**). The CREs (1-7) in the genome are marked in light blue (**a**, **c**, **d**). The homologous CREs in the mouse genome are denoted with black arrowheads (**e**). *P* values were calculated using two-tailed Welch's *t*-test with FDR correction (**b**).

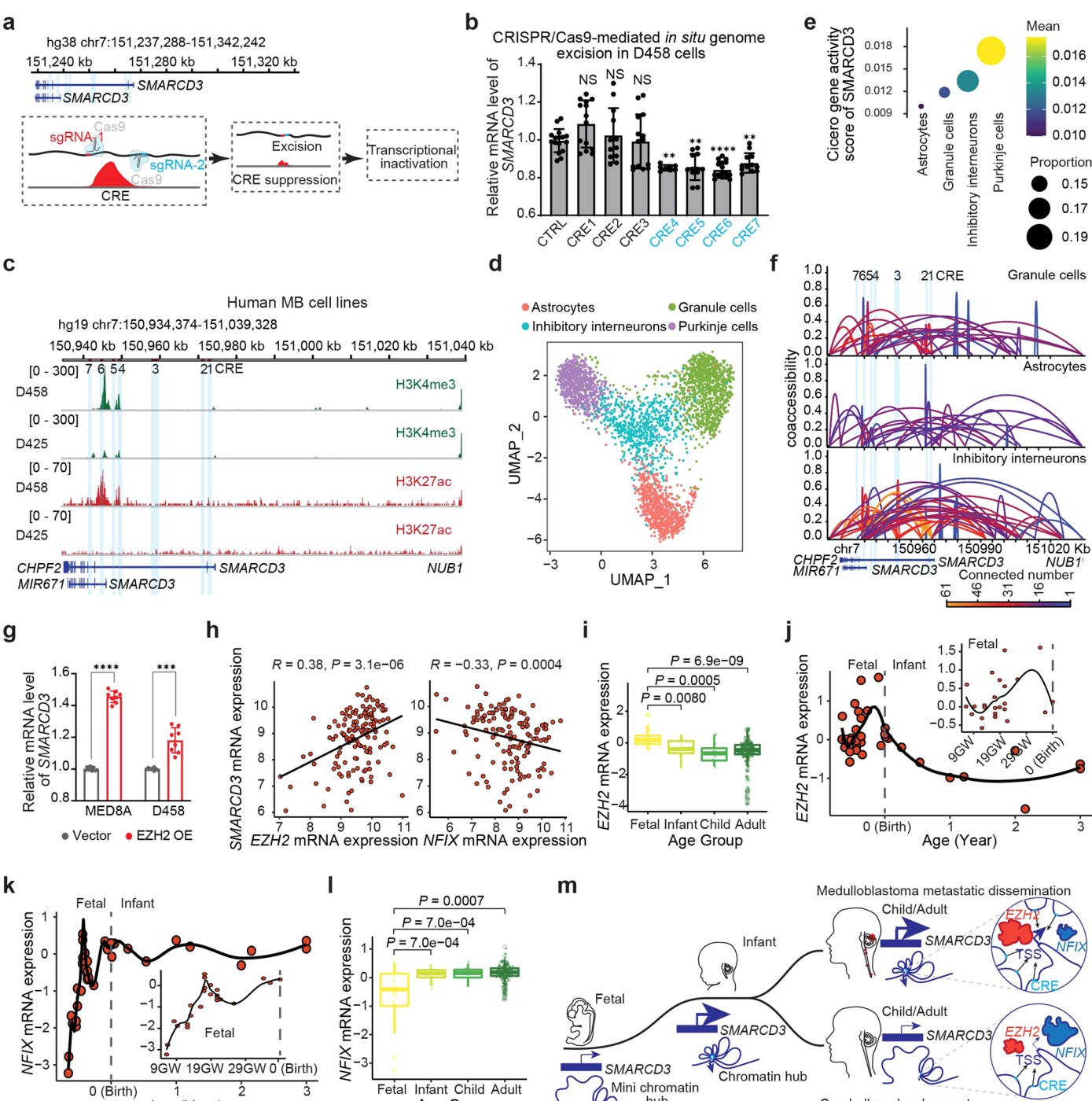

**Extended Data Fig. 8 | Chromatin remodeling of SMARCD3 transcriptional regulation in MB and normal human cerebellum. a**, The schematic showing CRISPR/Cas9-mediated *in situ* genome exclusion. **b**, qRT-PCR for *SMARCD3* mRNA expression in D458 cells with CRE excision ($n_{CTRL}=16, n_{CRE1}=14, n_{CRE2}=12, n_{CRE3}=14, n_{CRE4}=8, n_{CRE5}=12, n_{CRE6}=16, n_{CRE7}=12$). **c**, Histone marker signals at the *SMARCD3* locus. The CREs are marked in light blue. **d**, UMAP visualization of sci-ATACseq3 data from human cerebellum ($n_{Astrocytes}=790, n_{Granule neurons}=1080, n_{Inhibitory interneurons}=669, n_{Purkinje neurons}=774$). **e**, Dotplot showing Cicero gene activity score calculated using sci-ATACseq3 data. **f**, Cicero coaccessibility links among *SMARCD3* CREs. **g**, qRT-PCR for *SMARCD3* mRNA expression in the cells with EZH2 OE *vs* Vector (n = 8 for each group). **h**, Association between *SMARCD3* and *EZH2* or *NFIX* mRNA expression in patient MB samples. Boxplot showing *EZH2* (**i**) or *NFIX* (**l**) mRNA expression of human cerebella. Scatterplots showing *EZH2* (**j**) and *NFIX*

(**k**) mRNA expression of human cerebella changing along with the developmental process. **m**, The schematic diagram shows SMARCD3 transcriptional regulation mediated by chromatin hubs in cerebellar development and MB metastatic dissemination. Each dot represents one bulk sample (**h**-**l**), one cell (**d**), or the average Cicero gene activity score of SMARCD3 within a cell type (**e**). Dot size represents the percentage of nuclei within a cell type in which the Cicero gene activity score is not zero (**e**). *n* represents the number of the biologically independent samples from at least 2 independent experiments (**b**, **g**); data are presented as the mean ± s.d. *P* and *R* values were calculated using one-way ANOVA with Dunnett's multiple comparison test (**b**), one-tailed unpaired *t*-test (**g**), two-tailed Welch's *t*-test with FDR correction (**i**, **l**), and/or two-tailed Spearman's rank correlation analysis (**h**). NS, not significant, ∗∗*P* = 0.0045 (CRE4)/0.0014 (CRE5)/0.0099 (CRE7), ∗∗∗∗*P* < 0.0001.

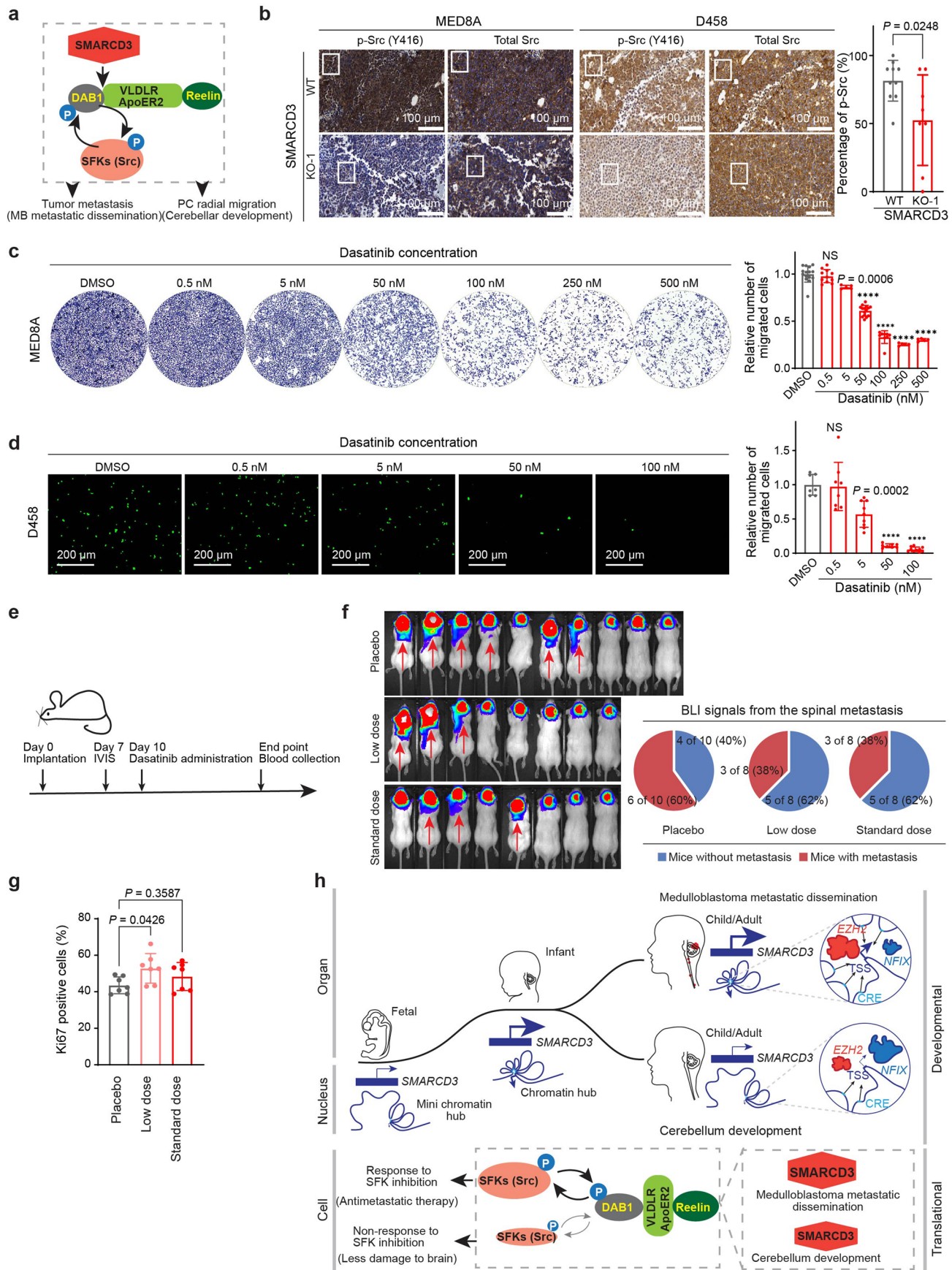

**Extended Data Fig. 9 | See next page for caption.**

**Extended Data Fig. 9 | Dasatinib treatment inhibits cell migration and tumor metastasis. a**, The schematic diagram shows that SMARCD3 induces PC radial migration and MB metastasis mediated by the Reelin/DAB1-activated SFK loop. **b**, IHC and quantitative analysis of expression levels of p-Src (416) and total Src in the tumors derived from mice bearing MED8A and D458 with SMARCD3 KO ($n = 8$) *vs* WT ($n = 10$), respectively. Boxed regions correspond to the regions shown in Fig. 7e. **c, d**, Representative images showing cell migration of MED8A ($n \geq 5$) and D458 ($n \geq 7$) cells treated with DMSO or indicated concentrations of dasatinib by Transwell assay. **e**, Scheme of experiment in which mice bearing MB were gavaged with placebo, low dose, and standard dose dasatinib. **f**, Bioluminescence images and pie charts showing mice bearing D458 cells with dasatinib treatment at day 21 after intracranial implantation. **g**, IHC

quantitative analysis of Ki67 levels in the treated mice ($n = 7$ for each group). **h**, The schematic diagram shows that SMARCD3 plays a central role in cerebellar development and MB metastatic dissemination by regulating the Reelin/DAB1/Src signaling at the molecular, cellular, and tissue/organ levels. SMARCD3 transcription regulation is mediated by chromatin hubs during cerebellar development and MB aggressiveness. Targeting SMARCD3/Reelin/DAB1/Src signaling provides a potential novel antimetastatic therapy for patients with MB. $n$ represents the number of the biologically independent samples from at least 3 independent experiments (**c, d**) or mouse samples (**b, g**); data are presented as the mean ± s.d. *P* value was calculated using two-tailed unpaired *t*-test (**b**), or one-way ANOVA with Dunnett's multiple comparison test (**c, d, g**). NS, not significant, ∗∗∗∗*P* < 0.0001.

# Reporting Summary

## Statistics

For all statistical analyses, confirm that the following items are present in the figure legend, table legend, main text, or Methods section.

| n/a | Confirmed | |
|---|---|---|
| ☐ | ☒ | The exact sample size (*n*) for each experimental group/condition, given as a discrete number and unit of measurement |
| ☐ | ☒ | A statement on whether measurements were taken from distinct samples or whether the same sample was measured repeatedly |
| ☐ | ☒ | The statistical test(s) used AND whether they are one- or two-sided<br>*Only common tests should be described solely by name; describe more complex techniques in the Methods section.* |
| ☐ | ☒ | A description of all covariates tested |
| ☐ | ☒ | A description of any assumptions or corrections, such as tests of normality and adjustment for multiple comparisons |
| ☐ | ☒ | A full description of the statistical parameters including central tendency (e.g. means) or other basic estimates (e.g. regression coefficient) AND variation (e.g. standard deviation) or associated estimates of uncertainty (e.g. confidence intervals) |
| ☐ | ☒ | For null hypothesis testing, the test statistic (e.g. *F*, *t*, *r*) with confidence intervals, effect sizes, degrees of freedom and *P* value noted<br>*Give P values as exact values whenever suitable.* |
| ☒ | ☐ | For Bayesian analysis, information on the choice of priors and Markov chain Monte Carlo settings |
| ☒ | ☐ | For hierarchical and complex designs, identification of the appropriate level for tests and full reporting of outcomes |
| ☐ | ☒ | Estimates of effect sizes (e.g. Cohen's *d*, Pearson's *r*), indicating how they were calculated |

*Our web collection on statistics for biologists contains articles on many of the points above.*

## Software and code

Policy information about availability of computer code

| Data collection | The RNAseq data were generated in this study by using Illumina NovaSeq 6000 platform; and the ATACseq and CUT&RUN were generated by using Illumina NextSeq 500 platform. The detailed information was available in Methods. |
|---|---|
| Data analysis | Hisat2 (v.2.1.0) was used to map reads from RNAseq data.<br>Seurat (v.3.2.3) was used to analyze the scRNAseq data.<br>Cicero (v.1.6.2) was used to analyze the sci-ATAC-seq3 data.<br>bowtie2 (v.2.3.5.1) was used to map reads from ATACseq and CUT&RUN data.<br>macs3 (v.3.0.0a6) was used to define accessible sites of ATACseq and CUT&RUN data.<br>R (v.3.5.1) was used to perform other statistical analysis and graph plots.<br>GSVA (v.1.36.3) was used to calculate the meta-PCNA score.<br>Cellranger (v.5.0.1) was used to analyze the scRNAseq and sci-ATAC-seq3 data.<br>IPA (v.01-16) was used to analyze pathway.<br>ImageJ (Fiji 1.53C) was used to analyze images taking by microscopes.<br>IGV (v.2.6.3) was used to visualize the peaks and alignments.<br>FlowJo (v.10.6.1) was used to analyze flow cytometry data.<br>GraphPad Prism (v.9.1.0) was used to analyze data of wound-healing assay, transwell assay, extent of metastasis, qRT-PCR, and statistical analysis.<br>Detailed information can be found in Methods. |

For manuscripts utilizing custom algorithms or software that are central to the research but not yet described in published literature, software must be made available to editors and reviewers. We strongly encourage code deposition in a community repository (e.g. GitHub). See the Nature Portfolio guidelines for submitting code & software for further information.

# Data

Policy information about <u>availability of data</u>

All manuscripts must include a <u>data availability statement</u>. This statement should provide the following information, where applicable:

- Accession codes, unique identifiers, or web links for publicly available datasets
- A description of any restrictions on data availability
- For clinical datasets or third party data, please ensure that the statement adheres to our <u>policy</u>

The availability of data is provided in the paper, which includes:
1) The primary datasets: the RNAseq, ATACseq, and CUT&RUN data that were generated in this study were deposited in the Gene Expression Omnibus (GEO) with accession number GSE194217. At this time, the editors and the reviewers can use this token (wnyvqusifjknfyd) to access the data using this link:https://www.ncbi.nlm.nih.gov/geo/query/acc.cgi?acc=GSE194217.
2) The referenced datasets: previously published data that were re-analyzed in the study include: transcriptomics of 1350 MBs and 291 normal cerebellum samples (GSE124814), scRNAseq data of 25 MBs (GSE119926), expression profiles and clinical data of 763 MBs (GSE85217), Hi-C data of mouse cerebellum (GSE138822), scRNAseq data of developing mouse cerebellum (European Nucleotide Archive: PRJEB23051), ChIPseq data of 5 MBs (GSE92585), sci-ATAC-seq3 data of fetal cerebellum (GSE149683), ChIPseq data of D458 and D425 (GSE129521), proteomic data of 45 MBs (Supplemental Table), 167 MB RNAseq data from R2 (https://r2.amc.nl), processed TCGA pan-cancer RNAseq data from Xena (https://xena.ucsc.edu/), gene profiling of normal human tissues from GTEx (https://www.gtexportal.org/home/), human cerebellum scRNAseq data were obtained from the Human Cell Atlas (https://www.covid19cellatlas.org/aldinger20), ChIPseq data of mouse cerebellum from ENCODE portal (https://www.encodeproject.org/), H3K27ac ChIPseq data of 4 MB subgroups from St. Jude Cloud Visualization Community (https://viz.stjude.cloud/).

# Human research participants

Policy information about <u>studies involving human research participants and Sex and Gender in Research.</u>

| Reporting on sex and gender | We analyzed the clinical data/samples of 10 medulloblastoma patients including 6 male and 4 female subjects from the bio-repositories in the Xiangya Hospital, Central South University. The MB tissue microarray FFPE slides were obtained from the bio-repositories at Johns Hopkins University. The sex and gender of these patients were provided by the bio-repositories, which were determined based on self-reporting and clinical routine physical examination. Our analysis in this study did not find significant differences in sex and gender. This study does not focus on sex and gender differences either. |
|---|---|
| Population characteristics | All 10 patients (age from 1 to 38 years old) included in this analysis were diagnosed with medulloblastoma based on histological, radiological, and clinical properties. The IMR/CT images and FFPE slides of 10 patients as well as the MB tissue microarray FFPE slides, which are de-identified and de-linked to any subject privacy information, were provided by the bio-repositories. |
| Recruitment | No patients recruited occurred for this study. Study materials and de-identified medical record information of the 10 medulloblastoma patients and the MB tissue microarray FFPE slides in this study were obtained from the established bio-repositories. |
| Ethics oversight | Given that the specimens or data were not collected specifically for this study and the subject identifiers linked to these specimens or data were not requested for this study, this study is not considered human subject research. The data and FFPE tissue slides for this study were provided by the bio-repositories that have the IRB-approval protocols, #202110207 (approved by the Clinical Ethics Committee of Xiangya Hospital, Central South University) and #NA_00015113 (approved by the Johns Hopkins University Institutional Review Board). Informed consent was obtained for these bio-repositories. The study is compliant with all ethical regulations. |

Note that full information on the approval of the study protocol must also be provided in the manuscript.

# Field-specific reporting

Please select the one below that is the best fit for your research. If you are not sure, read the appropriate sections before making your selection.

☒ Life sciences ☐ Behavioural & social sciences ☐ Ecological, evolutionary & environmental sciences

For a reference copy of the document with all sections, see <u>nature.com/documents/nr-reporting-summary-flat.pdf</u>

# Life sciences study design

All studies must disclose on these points even when the disclosure is negative.

| Sample size | Sample size was chosen on the basis of our previously published studies (PMID: 27863244; PMID: 34228644) and chosen empirically as per the standard custom followed in the field. No statistical method was used to pre-determine sample size. |
|---|---|
| Data exclusions | We excluded low quality cells during single-cell RNAseq analysis using the criteria as described in the Methods. No other data were excluded. |

| Replication | For sequencing replication, three replicates were performed for the RNAseq, ATACseq and H3K4me3 in CUT&RUN of SMARCD3 WT and KO in MED8A cells; and two replicates were performed for H3K4me1, H3K27ac and H3K27me3 in CUT&RUN of SMARCD3 WT and KO in MED8A cells.<br>Number of biological replicates of other experiments were described in figure legends. All attempts of replication were successful. |
|---|---|
| Randomization | Mice for implanting tumor cells were randomly grouped.<br>Mice for dasatinib treatment (Figures7j, 7k, and extended data Figures 9f, 9g) were measured for tumor size with IVIS at 7 days after implanting tumor cells, and then were divided into big, middle, or small tumor groups. The mice in each group were randomly grouped into dasatinib standard dose, low dose, or placebo group.<br>No randomization was performed for other experiments because control groups and treated groups (such as sgRNA knockout, overexpression, and drug treatment what do you mean drug treatment) in these experiments were defined. |
| Blinding | The investigators were blinded for experimental group during data collection and analysis, IHC or IF analysis for the assessment of protein staining intensity in Figures 1g; 2g, l, m; 4e, 7b-e, and extended data Figures 3a, b, e, g, h, i, k, l; 9b. No blinding was applied for other experiments because the investigators had to know the groups for assessment and analyses. |

# Reporting for specific materials, systems and methods

We require information from authors about some types of materials, experimental systems and methods used in many studies. Here, indicate whether each material, system or method listed is relevant to your study. If you are not sure if a list item applies to your research, read the appropriate section before selecting a response.

## Materials & experimental systems

| n/a | Involved in the study |
|---|---|
| ☐ | ☒ Antibodies |
| ☐ | ☒ Eukaryotic cell lines |
| ☒ | ☐ Palaeontology and archaeology |
| ☐ | ☒ Animals and other organisms |
| ☒ | ☐ Clinical data |
| ☒ | ☐ Dual use research of concern |

## Methods

| n/a | Involved in the study |
|---|---|
| ☒ | ☐ ChIP-seq |
| ☐ | ☒ Flow cytometry |
| ☐ | ☒ MRI-based neuroimaging |

# Antibodies

| Antibodies used | The detailed information about all antibodies used in this study was provided below and in Supplementary Table 8.<br>Rabbit monoclonal anti-SMARCD3/BAF60C (D6F1S), Cat# 62265, RRID:AB_2799624, Cell Signaling Technology<br>Rabbit polyclonal anti-SMARCD3, Cat# PA5-41093, RRID:AB_2607216, Thermo Fisher Scientific<br>Goat polyclonal anti-FOXP2 (C terminus), Cat# EB05226, RRID:AB_2107112, Everest<br>Chicken polyclonal anti-CALB1, Cat# CH22118, RRID:AB_2737107,Neuromics<br>Rabbit monoclonal anti-SRC (36D10),Cat# 2109, RRID:AB_2106059,Cell Signaling Technology<br>Rabbit polyclonal anti-Phospho-Src Family (Tyr416),Cat# 2101, RRID:AB_331697,Cell Signaling Technology<br>Rabbit polyclonal anti-Phospho-Src (Y419),Cat# AF2685, RRID:AB_442167,R&D<br>Mouse monoclonal anti-b-Actin (Clone AC-74),Cat# A2228, RRID:AB_476697,Sigma-Aldrich<br>Rabbit recombinant polyclonal anti-H3K4me1,Cat# 710795, RRID: AB_2532764,Thermo Fisher Scientific<br>Rabbit monoclonal anti-H3K4me3 (clone 15-10C-E4),Cat# 05-745R, RRID: AB_1587134, Millipore<br>Rabbit polyclonal anti-H3K9me3, Cat# Ab8898, RRID: AB_306848, Abcam<br>Rabbit polyclonal anti-H3K27ac, Cat# Ab4729, RRID: AB_2118291, Abcam<br>Rabbit polyclonal anti-H3K27me3,Cat# 07-449, RRID: AB_310624, Millipore<br>Mouse monoclonal anti-Nestin (Clone rat-401), Cat# MAB353, RRID:AB_94911, Millipore<br>Rabbit polyclonal anti-GFAP, Cat# Z0334, RRID:AB_10013382, Agilent<br>Rabbit polyclonal anti-Olig-2, Cat# AB9610, RRID:AB_570666, Millipore<br>Mouse monoclonal anti-NPR-C (clone E-5), Cat #515449, Santa Cruz Biotechnology<br>Mouse monoclonal anti-nestin (10c2), Cat# sc-23927, RRID:AB_627994, Santa Cruz Biotechnology<br>Mouse monoclonal anti-TUBB3 (clone TUJ1), Cat# 801201, RRID:AB_2313773,BioLegend<br>Mouse monoclonal anti-Synaptophysin (clone SP17), Cat# 837103, RRID:AB_2783410, BioLegend<br>Rat monoclonal anti-BrdU (clone BU1/75 (ICR1)), Cat# ab6326, RRID:AB_305426, Abcam<br>Rabbit monoclonal anti-Ki67 (VP-RM04), Cat# VP-RM04 RRID:AB_2336545, Vector Laboratories<br>Rabbit polyclonal anti-Cleaved Caspase-3 (Asp175) (clone D175), Cat# 9661, RRID:AB_2341188, Cell Signaling Technology<br>Goat anti-chicken IgY (H+L) secondary antibody, Alexa Fluor™ 647, Cat# A-21449, RRID:AB_2535866, Thermo Fisher Scientific<br>Donkey anti-rabbit IgG (H+L) highly cross-adsorbed, Alexa Fluor™ 594, Cat# A-21207, RRID:AB_141637, Thermo Fisher Scientific<br>Labeled polymer-HRP anti-mouse, Cat# K4006, Dako<br>HRP Horse anti-rabbit IgG polymer reagent, Cat# MP-7401, RRID:AB_2336529, Vector Laboratories |
|---|---|
| Validation | Regarding SMARCD3 antibody (Cat# 62265,CST), the validation was provided by the the manufacturer's website and also performed by using the MED8A cell line with CRISPR/CAS9 mediated SMARCD3 deletion vs wildtype for IHC and WB assay.<br><br>Rabbit polyclonal anti-H3K27me3,  Rabbit polyclonal anti-Phospho-Src Family (Tyr416), and Labeled polymer-HRP anti-mouse antibodies have been validated  in the following publications: PMID: 32313005 and PMID: 33958790 for Rabbit polyclonal anti- |

H3K27me3, PMID: 31263101 and PMID: 29533785 for Rabbit polyclonal anti-Phospho-Src Family (Tyr416); PMID: 27863244 and PMID: 34228644 for Labeled polymer-HRP anti-mouse antibody.

The validations for other antibodies were provided by the manufacturers' websites:
Rabbit polyclonal anti-SMARCD3 https://www.thermofisher.com/antibody/product/BAF60C-Antibody-Polyclonal/PA5-41093
Goat polyclonal anti-FOXP2 (C terminus) https://everestbiotech.com/product/goat-anti-foxp2-c-terminus-antibody/
Chicken polyclonal anti-CALB1 https://www.neuromics.com/CH22118
Rabbit monoclonal anti-SRC (36D10) https://www.cellsignal.com/products/primary-antibodies/src-36d10-rabbit-mab/2109
Rabbit polyclonal anti-Phospho-Src (Y419) https://www.rndsystems.com/products/human-phospho-src-y419-antibody_af2685
Mouse monoclonal anti-b-Actin (Clone AC-74) https://www.sigmaaldrich.com/US/en/product/sigma/a2228
Rabbit recombinant polyclonal anti-H3K4me1 https://www.thermofisher.com/antibody/product/H3K4me1-Antibody-Recombinant-Polyclonal/710795
Rabbit monoclonal anti-H3K4me3 https://www.emdmillipore.com/US/en/product/Anti-trimethyl-Histone-H3-Lys4-Antibody-clone-15-10C-E4-rabbit-monoclonal,MM_NF-05-745R?ReferrerURL=https%3A%2F%2Fwww.google.com%2F
Rabbit polyclonal anti-H3K9me3 https://www.abcam.com/histone-h3-tri-methyl-k9-antibody-chip-grade-ab8898.html
Rabbit polyclonal anti-H3K27ac https://www.abcam.com/histone-h3-acetyl-k27-antibody-chip-grade-ab4729.html

Mouse monoclonal anti-Nestin (Clone rat-401) https://www.emdmillipore.com/US/en/product/Anti-Nestin-Antibody-clone-rat-401,MM_NF-MAB353?ReferrerURL=https%3A%2F%2Fwww.google.com%2F
Rabbit polyclonal anti-GFAP https://www.agilent.com/en/product/immunohistochemistry/antibodies-controls/primary-antibodies/glial-fibrillary-acidic-protein-(concentrate)-76683
Rabbit polyclonal anti-Olig-2 https://www.sigmaaldrich.com/US/en/product/mm/ab9610
Mouse monoclonal anti-NPR-C https://www.scbt.com/p/npr-c-antibody-e-5
Mouse monoclonal anti-nestin (10c2) https://www.scbt.com/p/nestin-antibody-10c2
Mouse monoclonal anti-TUBB3 https://www.biolegend.com/ja-jp/products/purified-anti-tubulin-beta-3-tubb3-antibody-11580
Mouse monoclonal anti-Synaptophysin https://www.biolegend.com/it-it/products/purified-anti-synaptophysin-antibody-16778
Rat monoclonal anti-BrdU https://www.abcam.com/brdu-antibody-bu175-icr1-proliferation-marker-ab6326.html
Rabbit monoclonal anti-Ki67 https://www.labome.com/product/Vector-Laboratories/VP-RM04.html check
Rabbit polyclonal anti-Cleaved Caspase-3 (Asp175) https://www.cellsignal.com/products/primary-antibodies/cleaved-caspase-3-asp175-antibody/9661
Goat anti-chicken IgY (H+L) secondary antibody, Alexa Fluor™ 647 https://www.thermofisher.com/antibody/product/Goat-anti-Chicken-IgY-H-L-Secondary-Antibody-Polyclonal/A-21449
Donkey anti-rabbit IgG (H+L) highly cross-adsorbed, Alexa Fluor™ 594 https://www.thermofisher.com/antibody/product/Donkey-anti-Rabbit-IgG-H-L-Highly-Cross-Adsorbed-Secondary-Antibody-Polyclonal/A-21207
HRP Horse anti-rabbit IgG polymer reagent https://vectorlabs.com/products/enzyme-polymer/immpress-hrp-horse-anti-rabbit-igg

# Eukaryotic cell lines

Policy information about cell lines and Sex and Gender in Research

| | |
|---|---|
| Cell line source(s) | MED8A was provided by Dr. Michael D.Taylor, The Hospital for Sick Children, Toronto, Canada.<br>D556 was provided by Dr. Darell D. Bigner, Duke University Medical Center, Durham, NC.<br>D425 and D458 were provided by Dr. Sameer Agnihotri, UPMC Children's Hospital of Pittsburgh, Pittsburgh, PA.<br>D341 was purchased from ATCC (# HTB-187).<br>The human cerebellar neural stem cells (hcNSCs) was provided by Dr. Eric H. Raabe, Johns Hopkins University School of Medicine, Baltimore, MD.<br>The 293T packaging cells was purchased from ATCC. |
| Authentication | The MB cell lines used in this study were obtained from the brain tumor labs.<br>The human cerebellar neural stem cells (hcNSCs) was obtained and used in the publication (PMID: 27012813).<br>D341 and 293T cells were purchased from ATCC with the vendor's authentication.<br>These cell lines were not authenticated (such as shot tandem repeat assay) by us in the lab. |
| Mycoplasma contamination | All cell lines were tested to be negative for mycoplasma using MycoAlert PLUS Mycoplasma Detection Kit (Lonza). |
| Commonly misidentified lines<br>(See ICLAC register) | No commonly misidentified cell lines from the ICLAC Register were used in the study. |

# Animals and other research organisms

Policy information about studies involving animals; ARRIVE guidelines recommended for reporting animal research, and Sex and Gender in Research

| | |
|---|---|
| Laboratory animals | Female and male ICR SCID mice at 4-6 weeks of age were purchased from Taconic Biosciences (Model # ICRS-F/ICRS-M). C57BL/B6 mice 4-6 weeks of age purchased from The Jackson Laboratory (Strain # 000664) were bred and maintained at CHP Rangos Research Center under pathogen-free conditions. All animal experiments were performed with the approval of the University of Pittsburgh Animal Care and Use Committee (IACUC) with #21049271. |
| Wild animals | The study did not involve wild animals. |
| Reporting on sex | The study doesn't focus on sex differences. The equal number of female and male SCID mice were used for dasatinib treatment experiments. Female SCID mice were mostly used for tumor xenograft experiments. Both male and female embryos were used to examine SMARCD3 expression experiments (Fig. 4d, e). The difference between female and male mice in the study is not significant. |

Mouse sex was determined through genital area and nipples.

Field-collected samples | The study did not involve samples collected from the field.

Ethics oversight | All animal experiments were performed with the approval of University of Pittsburgh Animal Care and Use Committee (IACUC).

Note that full information on the approval of the study protocol must also be provided in the manuscript.

# Flow Cytometry

## Plots

Confirm that:

☒ The axis labels state the marker and fluorochrome used (e.g. CD4-FITC).

☒ The axis scales are clearly visible. Include numbers along axes only for bottom left plot of group (a 'group' is an analysis of identical markers).

☒ All plots are contour plots with outliers or pseudocolor plots.

☒ A numerical value for number of cells or percentage (with statistics) is provided.

## Methodology

Sample preparation | See Methods, in the section "Flow cytometry and FACS sorting".

Instrument | BD Fortessa and BD FACSAria cell sorter.

Software | Data were analyzed using FlowJo (v.10.6.1).

Cell population abundance | Sorted cells were resorted using the same gating strategy and purity was above 90%.

Gating strategy | For analyzing circulating tumor cells (CTCs), the PBMCs isolated from the mice without tumor cell implantation were used as the negative control, and the GFP-labeled tumor cells were used as the positive control to check the background signal and set the gate for the GFP channel. Tumor cells added hydrogen peroxide or not were used as a positive control or negative control to check the background signal and set the gate for the Propidium iodide (PI) channel.
For sorting GFP positive cells, the tumor cells without GFP labeled were used as the negative control to check the background signal and set the gate for cell sorting.

☒ Tick this box to confirm that a figure exemplifying the gating strategy is provided in the Supplementary Information.

# Magnetic resonance imaging

## Experimental design

Design type | Standard of small animal imaging and standard of care clinical imaging.

Design specifications | Standard design for assessing tumor development and growth in mouse brain. Routine of clinical imaging for human. No specific design was applied.

Behavioral performance measures | Behavioral performance measures were not applicable in the study.

## Acquisition

Imaging type(s) | Anatomical

Field strength | 7 Tesla for mouse and 3 Tesla for human.

Sequence & imaging parameters | For mouse brain MRI (Extended Data Fig. 3e): T1-weighted with contrast images were obtained with the typical imaging parameters: FOV 3.0 cm × 2.0 cm, acquisition matrix 384 × 256, acquisition slice thickness 0.60 mm, TR/TE = 2177/14 ms.
For human brain MRI (Fig. 7a): 2D sagittal T1-weighted FLAIR with contrast images were obtained with imaging parameters: acquisition matrix 320 × 256, acquisition slice thickness 5 mm, TR/TE = 1961/27 ms.

Area of acquisition | Whole brain imaging.

Diffusion MRI ☐ Used ☒ Not used

## Preprocessing

Preprocessing software | Preprocessing was not applied in this study.

| Normalization | Normalization was not applied in this study. |
| Normalization template | Normalization template was not applied in this study. |
| Noise and artifact removal | Noise and artifact removal was not applied in this study. |
| Volume censoring | Volume censoring was not applied in this study. |

## Statistical modeling & inference

| Model type and settings | No statistical modeling and inference was used in this study. |
| Effect(s) tested | No statistical modeling and inference was used in this study. |

Specify type of analysis: ☒ Whole brain ☐ ROI-based ☐ Both

| Statistic type for inference (See Eklund et al. 2016) | No statistical modeling and inference was used in this study. |
| Correction | No statistical modeling and inference was used in this study. |

## Models & analysis

| n/a | Involved in the study |
| ☒ ☐ | Functional and/or effective connectivity |
| ☒ ☐ | Graph analysis |
| ☒ ☐ | Multivariate modeling or predictive analysis |

