## [Peer Review File · Nature Cell Biology]

Peer Review Information

Journal: Nature Cell Biology

Manuscript Title: A neurodevelopmental epigenetic program mediated by SMARCD3-DAB1-Reelin signaling is hijacked to promote metastatic dissemination of medulloblastoma

Corresponding author name(s): Sarah J. Hainer, Michael D. Taylor, Baoli Hu

Editorial Notes:

**Redactions –
unpublished data**

Reviewer Comments & Decisions:

Decision Letter, initial version:

Date: 17th March 22 18:22:58

Last Sent: 17th March 22 18:22:58

Triggered By: Zhe Wang

From: zhe.wang@nature.com

To: baoli.hu@pitt.edu

CC: edmund.irwin@nature.com

Subject: Decision on Nature Cell Biology submission NCB-A47467

Message: *Please delete the link to your author homepage if you wish to forward this email to co-authors.

Dear Dr Hu,

Please first accept our apology for the delay getting back to you due to difficulties in retrieving reviewers comments

Your manuscript, "Hijacking a neurodevelopmental epigenomic program in metastatic dissemination of medulloblastoma", has now been seen by 2 referees, who are experts in medulloblastoma, epigenetics (referee 2); and brain tumors, cancer-neuro crosstalk (referee 3). As you will see from their comments (attached below) they find this work of potential interest, but have raised substantial concerns, which in our view would need to be addressed with considerable revisions before we can consider

publication in Nature Cell Biology.

We would like to clarify that although we have engaged a third referee (Reviewer 1) with expertise on SWI/SNF in cancer on the referee panel, the expertise/comments by the other two referees were sufficient for us to form a decision in the absence of this expert's feedback, and we felt a further delay would be counterproductive for the authors. We will send you the third report if/when we receive it.

Nature Cell Biology editors discuss the referee reports in detail within the editorial team, including the chief editor, to identify key referee points that should be addressed with priority, and requests that are overruled as being beyond the scope of the current study. To guide the scope of the revisions, I have listed these points below. We are committed to providing a fair and constructive peer-review process, so please feel free to contact me if you would like to discuss any of the referee comments further.

In particular, it would be essential to:

A) Strengthen the *in vivo/ex vivo* data as requested by Reviewer 3:

"A weaker part of the manuscript is the cell biology part, e.g. the definite proof that tumor cell migration/invasion is influenced by the proposed mechanism, and how exactly with respect to the known different modes of tumor cell movement. Time-lapse videos of cellular invasion/migration, ideally in an *in vivo* or at least proper *ex vivo* setting (tumor/brain slices), would be important."

"Moreover, it would also improve the story if genetic medulloblastoma models are included, ideally with dysregulated SMARCD3 signaling, to gain a better understanding how the dysregulation of this pathway influences medulloblastoma development, growth and metastasis - "from the start"."

B) Clarify the cohort information as questioned by both reviewers:

Reviewer 2

"...However, it remains unclear, whether this was a Group 3 cohort or an overall MB cohort across molecular groups, in which case the survival analysis would probably be of limited value. The same holds true for the IHC results on a TMA. There was also no information provided on a potential correlation with metastatic stage (a trend for which was provided in Figure 1i)..."

"...It is not clear, whether this was a group 3 MB cohort or across all molecular groups."

Reviewer 3

"...the authors report that medulloblastoma cells of at least the G3 subtype (this subtype issue needs more clarification) use neurodevelopmental epigenetic programs to promote metastatic dissemination, ..."

C) All other referee concerns pertaining to strengthening existing data, providing controls, methodological details, clarifications and textual changes, as applicable should also be addressed.

D) Finally please pay close attention to our guidelines on statistical and methodological reporting (listed below) as failure to do so may delay the reconsideration of the revised manuscript. In particular please provide:

- a Supplementary Figure including unprocessed images of all gels/blots in the form of a multi-page pdf file. Please ensure that blots/gels are labeled and the sections presented in the figures are clearly indicated.
- a Supplementary Table including all numerical source data in Excel format, with data for different figures provided as different sheets within a single Excel file. The file should include source data giving rise to graphical representations and statistical descriptions in the paper and for all instances where the figures present representative experiments of multiple independent repeats, the source data of all repeats should be provided.

We would be happy to consider a revised manuscript that would satisfactorily address these points, unless a similar paper is published elsewhere, or is accepted for publication in Nature Cell Biology in the meantime.

- ensure that it conforms to our format instructions and publication policies (see below and www.nature.com/nature/authors/).
- provide a point-by-point rebuttal to the full referee reports verbatim, as provided at the end of this letter.
- provide the completed Editorial Policy Checklist (found here <https://www.nature.com/authors/policies/Policy.pdf>), and Reporting Summary (found here <https://www.nature.com/authors/policies/ReportingSummary.pdf>). This is essential for reconsideration of the manuscript and these documents will be available to editors and referees in the event of peer review. For more information see <http://www.nature.com/authors/policies/availability.html> or contact me.

Nature Cell Biology is committed to improving transparency in authorship. As part of our efforts in this direction, we are now requesting that all authors identified as 'corresponding author' on published papers create and link their Open Researcher and Contributor Identifier (ORCID) with their account on the Manuscript Tracking System (MTS), prior to acceptance. ORCID helps the scientific community achieve unambiguous attribution of all scholarly contributions. You can create and link your ORCID from the home page of the MTS by clicking on 'Modify my Springer Nature account'. For more information please visit please visit www.springernature.com/orcid.

[REDACTED]

*This url links to your confidential home page and associated information about

manuscripts you may have submitted or be reviewing for us. If you wish to forward this email to co-authors, please delete the link to your homepage.

We would like to receive a revised submission within six months. We would be happy to consider a revision even after this timeframe, however if the resubmission deadline is missed and the paper is eventually published, the submission date will be the date when the revised manuscript was received.

We hope that you will find our referees' comments, and editorial guidance helpful. Please do not hesitate to contact me if there is anything you would like to discuss.

Best wishes,

Zhe Wang

Zhe Wang, PhD
Senior Editor
Nature Cell Biology

Tel: +44 (0) 207 843 4924
email: zhe.wang@nature.com

Reviewers' Comments:

Reviewer #1:
None

Reviewer #2:

Remarks to the Author:

The manuscript entitled "Hijacking a neurodevelopmental epigenomic program in metastatic dissemination of medulloblastoma" by Han Zou and coworkers aims at identifying the underpinnings of metastatic dissemination in Group 3 medulloblastoma.

The authors start by identifying SMARCD3 as the only epigenetic modulator amongst 77 Group 3 specific DEGs, which in turn shows (modestly) elevated mRNA and protein expression in Group 3 MB. They also show survival data for SMARCD3 high and low expressing MBs. However, it remains unclear, whether this was a Group 3 cohort or an overall MB cohort across molecular groups, in which case the survival analysis would probably be of limited value. The same holds true for the IHC results on a TMA. There was also no information provided on a potential correlation with metastatic stage (a trend for which was provided in Figure 1i). The authors went on to demonstrate that higher SMARCD3 levels in Group 3 cell lines (n=6) were associated with a higher migratory potential in transwell assays as well as in xenograft models. Knockout of SMARCD3 in two cell lines was associated with less migration in vitro and in vivo (the latter with an n=4 each).

This was much further substantiated in gain- and loss-of-function assays in a matched primary-metastatic cell line pair including strong differences in the number of CTCs. Survival differences in these experiments were rather modest. Knockout experiments in two cell lines revealed a downregulation of RELN pathway

genes. Of these, DAB1 was also associated with modest overexpression in G3 and correlation with SMARCD3 levels (both mRNA and protein). There also seems to be a slight difference in DAB1 expression between metastatic and non-metastatic tumors. It is not clear, whether this was a group 3 MB cohort or across all molecular groups. Next the authors investigated the role of RELN pathway members in normal embryonal murine Purkinje cells and their interaction with granule cells. They showed that DAB1 (and others) were highly expressed in developing PCs, whereas RELN was highly expressed in GCs. The conclusions from figure 4 seem quite strong given that they are solely based on correlations.

ATAC-Seq in one of the knockout cell lines revealed decreased chromatin accessibility at the DAB1 locus upon knockout of SMARCD3 indicating a direct regulatory role. This was also associated with other repressive chromatin marks in tumor samples. In a cell line model and a small patient cohort the authors also showed that SMARCD3 was associated with increased enhancer activity in group 3 MB. It is not clear why the author didn't use published enhancer datasets on primary samples to substantiate this finding.

Furthermore, the authors demonstrated that specific CREs were responsible for the regulation of SMARCD3.

Using knockout experiments, the authors narrowed down candidate transcription factors to regulate SMARCD3 in Gr 3 MB cell lines to EZH2 and NFIX.

Finally, the authors provide evidence that SMARCD3 expression and src phosphorylation were correlated in 10 paired patient samples and increased upon metastatic dissemination. Dasatinib reduced the migratory propensity in vitro and in vivo, but did not influence cell survival.

Overall, this is a well conducted project with some limitations as indicated, which identifies one pro-metastatic mechanism in group 3 medulloblastoma based on hijacking a developmental program.

Minor:

LGG is the most common brain tumor in children, and even for malignant brain tumors, HGGs are more common than MB.

Reviewer #3:

Remarks to the Author:

In this manuscript, the authors report that medulloblastoma cells of at least the G3 subtype (this subtype issue needs more clarification) use neurodevelopmental epigenetic programs to promote metastatic dissemination, with SMARCD3 activating Reelin/DAB1/SRC signaling. Tumor biology is mapped to normal neuro/cerebellar development, back-and-forth in a quite compelling way.

While the manuscript is often not an easy read for the non-molecular biologist, the findings are of considerable interest and value for the field. The question how (brain) tumor hijack neurodevelopmental programs to thrive is certainly a very relevant one, and one that bears the promise to gain much deeper understanding of these challenging diseases. The authors might want to stress this point even more, and put their findings into a broader (neuro-)oncology context, not restricted to medulloblastoma.

The experiments are well conducted, and the conclusions are solid. There are a few points that reduce the enthusiasm a little bit at this point of time and which should be addressed:

A weaker part of the manuscript is the cell biology part, e.g. the definite proof that tumor cell migration/invasion is influenced by the proposed mechanism, and how exactly with respect to the known different modes of tumor cell movement. Time-lapse videos of cellular invasion/migration, ideally in an in vivo or at least proper ex vivo setting (tumor/brain slices), would be important.

Moreover, it would also improve the story if genetic medulloblastoma models are included, ideally with dysregulated SMARCD3 signaling, to gain a better understanding how the dysregulation of this pathway influences medulloblastoma development, growth and metastasis - "from the start".

Fig. 1f and 1g: what about survival differences in Subgroup G3 only when comparing high vs low expressers? Otherwise, this analysis does not add a lot to Fig. 1e.

Fig. 1i: the difference of SMARCD3 expression between metastatic and non-metastatic patients is there and statistically significant, but not extremely strong. What about other genes: how many genes differ between two groups- how many show a stronger difference, how many a smaller difference? Where does SMARCD3 fit into this picture?

It would be important to know whether SMARCD3 KO/lower expressing cells have a general problem with cell viability (e.g., decreased proliferation rate), or whether the effect of low/no expression is specific for cell invasion/migration. The latter would be more compelling when speaking of SMARCD3 as a factor for metastasis.

Minor: the first sentence of the abstract ("how dysregulation...remains elusive") reads a little bold, when considering the high number of neurodevelopmental genes that have been investigated in medulloblastoma. Nevertheless, the authors are certainly correct that a study covering both aspects in depths (and providing new findings for both, tumorigenesis and neurodevelopment at the same time) is very rare in the field. Maybe the authors want to consider to stress this point more.

Minor 2: A better graphical summary/schematic drawing could make the complex story better digestible for most readers.

READABILITY OF MANUSCRIPTS – Nature Cell Biology is read by cell biologists from

diverse backgrounds, many of whom are not native English speakers. Authors should aim to communicate their findings clearly, explaining technical jargon that might be unfamiliar to non-specialists, and avoiding non-standard abbreviations. Titles and abstracts should concisely communicate the main findings of the study, and the background, rationale, results and conclusions should be clearly explained in the manuscript in a manner accessible to a broad cell biology audience. Nature Cell Biology uses British spelling.

REFERENCES – are limited to a total of 70 for Articles, Resources, Technical Reports; and 40 for Letters. This includes references in the main text and Methods combined. References must be numbered sequentially as they appear in the main text, tables and figure legends and Methods and must follow the precise style of Nature Cell Biology references. References only cited in the Methods should be numbered

consecutively following the last reference cited in the main text. References only associated with Supplementary Information (e.g. in supplementary legends) do not count toward the total reference limit and do not need to be cited in numerical continuity with references in the main text. Only published papers can be cited, and each publication cited should be included in the numbered reference list, which should include the manuscript titles. Footnotes are not permitted.

Methods should be written concisely, but should contain all elements necessary to allow interpretation and replication of the results. As a guideline, Methods sections typically do not exceed 3,000 words. The Methods should be divided into subsections listing reagents and techniques. When citing previous methods, accurate references should be provided and any alterations should be noted. Information must be provided about: antibody dilutions, company names, catalogue numbers and clone numbers for monoclonal antibodies; sequences of RNAi and cDNA probes/primers or company names and catalogue numbers if reagents are commercial; cell line names, sources and information on cell line identity and authentication. Animal studies and experiments involving human subjects must be reported in detail, identifying the committees approving the protocols. For studies involving human subjects/samples, a statement must be included confirming that informed consent was obtained. Statistical analyses and information on the reproducibility of experimental results should be provided in a section titled "Statistics and Reproducibility".

All Nature Cell Biology manuscripts submitted on or after March 21 2016 must include a Data availability statement at the end of the Methods section. For Springer Nature policies on data availability see <http://www.nature.com/authors/policies/availability.html>; for more information on this particular policy see <http://www.nature.com/authors/policies/data/data-availability-statements-data-citations.pdf>. The Data availability statement should include:

- Accession codes for primary datasets (generated during the study under consideration and designated as "primary accessions") and secondary datasets (published datasets reanalysed during the study under consideration, designated as "referenced accessions"). For primary accessions data should be made public to coincide with publication of the manuscript. A list of data types for which submission to community-endorsed public repositories is mandated (including sequence, structure, microarray, deep sequencing data) can be found here <http://www.nature.com/authors/policies/availability.html#data>.
- Unique identifiers (accession codes, DOIs or other unique persistent identifier) and hyperlinks for datasets deposited in an approved repository, but for which data deposition is not mandated (see here for details <http://www.nature.com/sdata/data-policies/repositories>).
- At a minimum, please include a statement confirming that all relevant data are available from the authors, and/or are included with the manuscript (e.g. as source

data or supplementary information), listing which data are included (e.g. by figure panels and data types) and mentioning any restrictions on availability.

- If a dataset has a Digital Object Identifier (DOI) as its unique identifier, we strongly encourage including this in the Reference list and citing the dataset in the Methods.

We recommend that you upload the step-by-step protocols used in this manuscript to the Protocol Exchange. More details can found at www.nature.com/protocolexchange/about.

All imaging data should be accompanied by scale bars, which should be defined in the legend.

Cropped images of gels/blots are acceptable, but need to be accompanied by size markers, and to retain visible background signal within the linear range (i.e. should not be saturated). The boundaries of panels with low background have to be demarked with black lines. Splicing of panels should only be considered if unavoidable, and must be clearly marked on the figure, and noted in the legend with a statement on whether the samples were obtained and processed simultaneously. Quantitative comparisons between samples on different gels/blots are discouraged; if this is unavoidable, it should only be performed for samples derived from the same experiment with gels/blots were processed in parallel, which needs to be stated in the legend.

- For line art, graphs, charts and schematics we prefer Adobe Illustrator (.AI), Encapsulated PostScript (.EPS) or Portable Document Format (.PDF). Files should be

saved or exported as such directly from the application in which they were made, to allow us to restyle them according to our journal house style.

SUPPLEMENTARY INFORMATION – Supplementary information is material directly relevant to the conclusion of a paper, but which cannot be included in the printed version in order to keep the manuscript concise and accessible to the general reader. Supplementary information is an integral part of a Nature Cell Biology publication and should be prepared and presented with as much care as the main display item, but it must not include non-essential data or text, which may be removed at the editor's discretion. All supplementary material is fully peer-reviewed and published online as part of the HTML version of the manuscript. Supplementary Figures and Supplementary Notes are appended at the end of the main PDF of the published

manuscript.

The total number of Supplementary Figures (not including the “unprocessed scans” Supplementary Figure) should not exceed the number of main display items (figures and/or tables (see our Guide to Authors and March 2012 editorial <http://www.nature.com/ncb/authors/submit/index.html#suppinfo>; <http://www.nature.com/ncb/journal/v14/n3/index.html#ed>). No restrictions apply to Supplementary Tables or Videos, but we advise authors to be selective in including supplemental data.

GUIDELINES FOR EXPERIMENTAL AND STATISTICAL REPORTING

REPORTING REQUIREMENTS – To improve the quality of methods and statistics reporting in our papers we have recently revised the reporting checklist we introduced in 2013. We are now asking all life sciences authors to complete two items: an Editorial Policy Checklist (found here <https://www.nature.com/authors/policies/Policy.pdf>) that verifies compliance with all required editorial policies and a reporting summary (found here <https://www.nature.com/authors/policies/ReportingSummary.pdf>) that collects information on experimental design and reagents. These documents are available to referees to aid the evaluation of the manuscript. Please note that these forms are dynamic ‘smart pdfs’ and must therefore be downloaded and completed in Adobe Reader. We will then flatten them for ease of use by the reviewers. If you would like to reference the guidance text as you complete the template, please access these flattened versions at <http://www.nature.com/authors/policies/availability.html>.

STATISTICS – Wherever statistics have been derived the legend needs to provide the n number (i.e. the sample size used to derive statistics) as a precise value (not a

range), and define what this value represents. Error bars need to be defined in the legends (e.g. SD, SEM) together with a measure of centre (e.g. mean, median). Box plots need to be defined in terms of minima, maxima, centre, and percentiles. Ranges are more appropriate than standard errors for small data sets. Wherever statistical significance has been derived, precise p values need to be provided and the statistical test used needs to be stated in the legend. Statistics such as error bars must not be derived from $n < 3$. For sample sizes of $n < 5$ please plot the individual data points rather than providing bar graphs. Deriving statistics from technical replicate samples, rather than biological replicates is strongly discouraged. Wherever statistical significance has been derived, precise p values need to be provided and the statistical test stated in the legend.

Author Rebuttal to Initial comments

The authors thank the reviewers for their comments on our manuscript (# NCB-A47467) and have addressed these comments as follows:

Reviewer #1:

None

Reviewer #2:

Remarks to the Author:

The manuscript entitled "Hijacking a neurodevelopmental epigenomic program in metastatic dissemination of medulloblastoma" by Han Zou and coworkers aims at identifying the underpinnings of metastatic dissemination in Group 3 medulloblastoma.

The authors start by identifying SMARCD3 as the only epigenetic modulator amongst 77 Group 3 specific DEGs, which in turn shows (modestly) elevated mRNA and protein expression in Group 3 MB. They also show survival data for SMARCD3 high and low expressing MBs. However, it remains unclear, whether this was a Group 3 cohort or an overall MB cohort across molecular groups, in which case the

survival analysis would probably be of limited value. The same holds true for the IHC results on a TMA. There was also no information provided on a potential correlation with metastatic stage (a trend for which was provided in Figure 1i).

Response: We appreciate the reviewer for highlighting this issue and we apologize for the point being unclear in the manuscript. First, we clarified that the survival analyses in **Fig. 1f** and **Fig. 1g** were based upon all four molecular subgroups of medulloblastoma (MB). We did the overall survival in Group 3 MB only and the result showed a trend of higher levels of SMARCD3 mRNA expression associated with a worse prognosis although this is not statistically significant (**Rebuttal Fig. 1a**). A similar observation was obtained in Group 3 MB by immunohistochemistry (IHC) analysis of SMARCD3 protein expression on the tissue microarray (TMA) (**Rebuttal Fig. 1b**). The evidence provided in the manuscript together with these additional data suggest that the elevated level of SMARCD3 is a hallmark feature of Group 3 MB that is the most aggressive subgroup compared to other MB subgroups¹. Therefore, we believe this is why the significant correlation between SMARCD3 and overall survival in Group 3 only was not easily observed.

Rebuttal Fig. 1: Patient survival analysis in Group 3 MB only. (a) Kaplan-Meier survival curve of patients in Group 3 MBs based on SMARCD3 mRNA expression levels used the Cavalli dataset. (b) Kaplan-Meier survival curve of patients in Group 3 MBs based on SMARCD3 protein expression levels by IHC analysis used the inhouse TMA.

Accordingly, we clarified these points in the revised manuscript (**Page 5; Line 105-110**).

The authors went on to demonstrate that higher SMARCD3 levels in Group 3 cell lines (n=6) were associated with a higher migratory potential in transwell assays as well as in xenograft models. Knockout of SMARCD3 in two cell lines was associated with less migration in vitro and in vivo (the latter with an n=4 each). This was much further substantiated in gain- and loss-of-function assays in a matched primary-metastatic cell line pair including strong differences in the number of CTCs. Survival differences in these experiments were rather modest.

Knockout experiments in two cell lines revealed a downregulation of RELN pathway genes. Of these, DAB1 was also associated with modest overexpression in G3 and correlation with SMARCD3 levels (both mRNA and protein). There also seems to be a slight difference in DAB1 expression between metastatic

and non-metastatic tumors. It is not clear, whether this was a group 3 MB cohort or across all molecular groups.

Response: We thank the reviewer for this question. The correlations between DAB1 mRNA expression and tumor metastasis were analyzed using MB patients across all the subgroups. We clarified this point in the revised manuscript (**Page 8, Line 194**).

Next the authors investigated the role of RELN pathway members in normal embryonal murine Purkinje cells and their interaction with granule cells. They showed that DAB1 (and others) were highly expressed in developing PCs, whereas RELN was highly expressed in GCs. The conclusions from figure 4 seem quite strong given that they are solely based on correlations. ATAC-Seq in one of the knockout cell lines revealed decreased chromatin accessibility at the DAB1 locus upon knockout of SMARCD3 indicating a direct regulatory role. This was also associated with other repressive chromatin marks in tumor samples. In a cell line model and a small patient cohort the authors also showed that SMARCD3 was associated with increased enhancer activity in group 3 MB. It is not clear why the author didn't use published enhancer datasets on primary samples to substantiate this finding.

Response: We thank the reviewer for asking for this clarification. We analyzed the published datasets, and the results were described in **the Extended Data Fig. 6a**. The datasets were used from the publication².

Furthermore, the authors demonstrated that specific CREs were responsible for the regulation of SMARCD3.

Using knockout experiments, the authors narrowed down candidate transcription factors to regulate SMARCD3 in Gr 3 MB cell lines to EZH2 and NFIX.

Finally, the authors provide evidence that SMARCD3 expression and src phosphorylation were correlated in 10 paired patient samples and increased upon metastatic dissemination. Dasatinib reduced the migratory propensity in vitro and in vivo, but did not influence cell survival.

Overall, this is a well conducted project with some limitations as indicated, which identifies one pro-metastatic mechanism in group 3 medulloblastoma based on hijacking a developmental program.

Response: We truly appreciate the evaluation made by the reviewer and the comments that are very helpful in improving the manuscript.

Minor:

LGG is the most common brain tumor in children, and even for malignant brain tumors, HGGs are more common than MB.

Response: We agree. We also thank the reviewer for pointing out this mistake. We edited the sentences in the revised manuscript (**Page 3, lines 48-49; page 3, Line 63-64**).

Reviewer #3:

Remarks to the Author:

In this manuscript, the authors report that medulloblastoma cells of at least the G3 subtype (this

subtype issue needs more clarification) use neurodevelopmental epigenetic programs to promote metastatic dissemination, with SMARCD3 activating Reelin/DAB1/SRC signaling. Tumor biology is mapped to normal neuro/cerebellar development, back-and-forth in a quite compelling way.

While the manuscript is often not an easy read for the non-molecular biologist, the findings are of considerable interest and value for the field. The question how (brain) tumor hijack neurodevelopmental programs to thrive is certainly a very relevant one, and one that bears the promise to gain much deeper understanding of these challenging diseases. The authors might want to stress this point even more, and put their findings into a broader (neuro-)oncology context, not restricted to medulloblastoma.

Response: We appreciate the reviewer making these important suggestions. MB is the most common type of embryonal tumor that arises in the cerebellum, indicating a relationship between tumor development and embryonal cell maturation. We are interested in understanding how abnormal brain development influences the tumor biology of medulloblastoma. We believe that SMARCD3-mediated tumor hijacking neurodevelopmental programs could happen in other subgroups of medulloblastoma. Interestingly, we found the enrichment of medulloblastoma in the SMARCD3-associated normal developmental genes based on gene-disease network analysis (DisGeNET) (**Extended Data Fig.4f**). These results suggest that SMARCD3-regulated Reelin/DAB1 signaling in cell migration and tumor cells hijacking SMARCD3-Reelin/DAB1 signaling for promoting metastasis could be specific to cerebellar development and medulloblastoma progression, respectively. Given activation of the Reelin/DAB1 signaling pathway in other cell types located in different regions of the brain, such as Cajal-Retzius cells in the neocortex and hippocampus^{3,4}, we also believe the regulatory network of SMARCD3-Reelin/DAB1 signaling could be happening in other cell types and brain areas under physiological and pathological conditions. We agree with the reviewer's comments and believe that tumor cells hijacking normal neurodevelopmental processes and mechanisms may broadly exist in other brain cancers.

We thank the reviewer again for this interesting discussion and we will focus on experimental validations on these points in our future research. We emphasized this point in the section of Discussion in the revised manuscript (**Page 17-18, Line 418-420, and Line 433-435**).

The experiments are well conducted, and the conclusions are solid. There are a few points that reduce the enthusiasm a little bit at this point of time and which should be addressed:

A weaker part of the manuscript is the cell biology part, e.g. the definite proof that tumor cell migration/invasion is influenced by the proposed mechanism, and how exactly with respect to the known different modes of tumor cell movement. Time-lapse videos of cellular invasion/migration, ideally in an in vivo or at least proper ex vivo setting (tumor/brain slices), would be important.

Response: We thank the reviewer for this question. We also thank the suggested experiments by the reviewer. To this end, time-lapse image acquisition was first performed in the cell culture condition of the medulloblastoma (MB) Group 3 cell line MED8A with SMARCD3 wildtype (WT) or SMARCD3 knockout (KO) in the scratch-wound healing assays. (**Supplementary Video 1**). Strikingly, SMARCD3 deletion significantly

decreased directional cell migration velocity and non-directional cell motility speed in MED8A cells (Revised Extended Data Fig.2j).

Revised Extended Data Fig. 2j: The effect of SMARCD3 on cell movement *in vitro*. Representative images of time-lapse imaging for the migration of MED8A cells with SMARCD3 WT vs KO-1 during scratch wound-healing assays. Cell tracks were drawn in the images. Scatterplot showing the correlation between the cell migration velocity and directionality for MED8A cells with SMARCD3 WT vs KO-1 in the wound healing assays. Each dot represents one cell. The size of the dot represents the average motility speed of one cell. Violin plot showing the instantaneous motility speed in the scratch-wound healing assays. Each dot represents the instantaneous motility speed of one cell at one time point. R and P values in the scatterplot were calculated by Pearson's correlation analysis. P value in the violin plot was calculated by FDR corrected Welch's t -test.

To further examine the effect of SMARCD3 on cell movement *in vivo*, we set up an *ex vivo* brain slice model by transplanting GFP-labeled MED8A-WT and MED8A-SMARCD3 KO cells into the cerebellum of SCID mice. A week following cell implantation, a time-lapse confocal microscope was used to examine cell movement under the brain slice culture condition (**Supplementary Video 2**). Consistent with *in vitro* results, the GFP+ tumor cell movement in directionality, velocity, and speed was significantly reduced in the cerebellar tissues implanted with MED8A-SMARCD3 KO compared with MED8A-WT cells (**Revised Extended Data Fig. 2k**).

Revised Extended Data Fig. 2k: The effect of SMARCD3 on cell movement in an *ex vivo* brain slice model. Representative images of time-lapse imaging for the migration of MED8A cells with SMARCD3 WT vs KO-1 in the brain slice culture. Cell tracks were drawn in the images. Scatterplot showing the correlation between the cell migration velocity and directionality for MED8A cells with SMARCD3 WT vs KO-1 in the brain tissues. Each dot represents one cell. The size of the dot represents the average motility speed of one cell. Violin plot showing the instantaneous motility speed in the brain tissues. Each dot represents the instantaneous motility speed of one cell at one time point. R and P values in the scatterplot were calculated by Pearson's correlation analysis. P value in the violin plot was calculated by FDR corrected Welch's t -test.

We added these extended data in the revised manuscript (**Page 7, Line 150-154**).

In this study, we found that mechanistically SMARCD3 upregulated DAB1 expression and activated its downstream signaling. It is known that DAB1 is a crucial cellular adaptor in Reelin signaling-mediated Purkinje cell migration and positioning during the early stages of the development of the cerebellum⁵. Moreover, the activated DAB1 signaling regulates cell skeleton, Cadherin, and integrin functions during neuronal positioning and migration^{4,6}. Three modes of tumor cell movement were described previously, including mesenchymal, amoeboid, and collective modes; and cancer cells can adapt their migration strategies to the different tumor microenvironments by switching among these migration modes^{7,8}. Based on our findings aligning with these time-lapse videos of cell movement, we believe that SMARCD3-driven MB cell movement is mediated by Reelin/DAB1 signaling-mediated neuronal migration, which increases the directionality in both MB cell movement and normal cerebellar development, together with mesenchymal and/or collective modes during tumor progression.

Moreover, it would also improve the story if genetic medulloblastoma models are included, ideally with dysregulated SMARCD3 signaling, to gain a better understanding how the dysregulation of this pathway influences medulloblastoma development, growth and metastasis - "from the start".

Response: Response: We thank the reviewer for making this point. To understand whether SMARCD3 also influences MB development and growth besides tumor metastasis, we examined the role of

SMARCD3 in the malignant transformation of neural stem cells that were considered as the cells of origin of Group 3 MB⁹⁻¹¹.

We employed virus-induced spontaneous tumor formation in postnatal C57BL/6J mice. [REDACTED] The results revealed that overexpression of MYC^{S62D} alone and MYC^{S62D} + SMARCD3 significantly induced tumor formation, however, overexpression of SMARCD3 alone did not induce tumor formation (**Rebuttal Fig. 2a, b**). While a statistically significant difference between the two groups was not obtained, there is a reproducible trend of shorter survival in mice bearing SMARCD3+ MYC^{S62D}-induced tumors compared with MYC^{S62D}-induced tumors (**Rebuttal Fig. 2b**). Furthermore, we observed no obvious differences in tumor sizes between MYC^{S62D}-induced tumors and SMARCD3+ MYC^{S62D}-induced tumors based on GFP fluorescence analysis (**Rebuttal Fig. 2c**). Consistently, SMARCD3 overexpression promoted tumor spinal

Rebuttal Fig. 2: Lentivirus-mediated genetic manipulations induce spontaneous MB formation in C57BL/6J mice. (a) A representative MRI image of the induced tumor. (b) Kaplan-Meier survival curve of C57BL/6J mice intracranially infected by lentivirus vector-mediated expression of SMARCD3, MYC^{S62D}, and MYC^{S62D} + SMARCD3. Log-rank test was used for survival fraction comparison between MYC^{S62D} and MYC^{S62D} + SMARCD3 groups. (c) Representative bright-field and fluorescence microscopy images of the mouse brain bearing MYC^{S62D}- and MYC^{S62D} + SMARCD3-induced tumors. (d) Representative bright-field and fluorescence microscopy images of the spinal cords from mice bearing MYC^{S62D}- and MYC^{S62D} + SMARCD3 induced tumors. The percentages of mice with spinal metastasis in the indicated groups are summarized. (e) Representative images of histopathological assessment of MYC^{S62D}- and MYC^{S62D} + SMARCD3-induced tumors for the indicated marker expression by H&E and IHC analyses.

metastasis in MYC^{S62D}-induced tumors (4 out of 4, 100% in SMARCD3 + MYC^{S62D} vs 2 out of 5, 40% in MYC^{S62D} alone) (**Rebuttal Fig. 2d**). Histopathological analysis revealed that both MYC^{S62D}- and SMARCD3+ MYC^{S62D}-induced tumors showed the typical features of Group 3 MB, including large cell/anaplastic (LCA) patterns with nuclear molding and wrapping, focally prominent nucleoli, abundant mitotic cells by H&E staining, high proliferation (Ki67), lack of glial marker glial fibrillary acidic protein (GFAP) expression, and differentiated neuronal lineage marker synaptophysin, but positive staining for neuronal progenitor markers, such as Nestin, β 3-tubulin, and Oligo2 (**Rebuttal Fig. 2e**). Importantly, both MYC^{S62D}- and SMARCD3 + MYC^{S62D}-induced tumors displayed Group 3 specific marker NPR3 expression by IHC staining (**Rebuttal Fig. 2e**). Of note, no differences in cell proliferation index (Ki67) were observed between these two group tumors (**Rebuttal Fig. 2e**). Collectively, these results suggest that SMARCD3 plays a pivotal role in tumor metastasis, rather than participating in MB initiation and growth.

We previously reported an approach of *in vivo* malignant transformation of human neural stem cells to generate a glioblastoma model¹². We employed this strategy to establish a new MB model by using a human cerebellar neural stem cell line. A previous study showed that this human cerebellar neural stem cell line expressing wildtype MYC alone (hereafter hcNSCs) displayed a low malignant potential for MB formation¹³. To examine if SMARCD3 increases the malignant transformation of hcNSCs, cells were infected with lentivirus carrying SMARCD3 (resulting in overexpression) and subsequently orthotopically implanted into the cerebellum of SCID mice. Notably, we did not observe SMARCD3-induced tumor formation in these SCID mice for up to 90 days, however, overexpression of constitutively active MYC^{S62D} in hcNSCs dramatically increased tumor formation in orthotopic SCID mouse models (**Rebuttal Fig. 3a**). While we did not observe significant differences in mouse survival and tumor sizes between MYC^{S62D} alone and SMARCD3 + MYC^{S62D} induced tumors (**Rebuttal Fig. 3a, b**), overexpression of SMARCD3 promoted tumor spinal metastasis in MYC^{S62D} induced tumors based on bioluminescence imaging (BLI) analysis and visualization of the spinal cord by assessing GFP-labeled tumor xenograft mice (**Rebuttal Fig. 3c, d**). The

data using this human-in-mouse MB model further support our conclusion that SMARCD3 significantly influences tumor metastasis rather than tumor development and growth.

The new data align with the results in the manuscript and together strongly indicate that SMARCD3 plays

Rebuttal Fig. 3: Malignant transformation of human cerebellar neural stem cells by MYC^{S62D} and/or SMARCD3.

(a) Kaplan-Meier survival curve of SCID mice intracranially implanted by hcNSCs expressing SMARCD3, MYC^{S62D}, and MYC^{S62D} + SMARCD3, respectively. Log-rank test was used for survival fraction comparison between MYC^{S62D} and MYC^{S62D} + SMARCD3 groups. (b) Representative bright-field and fluorescence microscopy images of the SCID mouse brain bearing MYC^{S62D}- and MYC^{S62D} + SMARCD3-induced tumors. (c) Representative luminescence images of SCID mice bearing hcNSCs expressing SMARCD3, MYC^{S62D}, and MYC^{S62D} + SMARCD3 following the days after cell implantation. The percentages of SCID mice with spinal metastasis in the indicated groups were summarized. (d) Representative bright-field and fluorescence microscopy images of the spinal cords from SCID mice bearing MYC^{S62D}- and MYC^{S62D} + SMARCD3 induced tumors.

a critical role in promoting cell migration and tumor metastasis, rather than tumor initiation and growth, during MB progression.

Fig. 1f and 1g: what about survival differences in Subgroup G3 only when comparing high vs low expressers? Otherwise, this analysis does not add a lot to Fig. 1e.

Response: We thank the reviewer for this concern. We did observe a trend of higher levels of SMARCD3 expression associated with a worse prognosis in Group 3 MB only, but not statistically different. Our data in this manuscript indicate a major role of SMARCD3 in driving cell migration and tumor metastasis, which is the hallmark of Group 3 MB, [REDACTED]

Fig. 1i: the difference of SMARCD3 expression between metastatic and non-metastatic patients is there and statistically significant, but not extremely strong. What about other genes: how many genes differ between two groups- how many show a stronger difference, how many a smaller difference? Where does SMARCD3 fit into this picture?

Response: We appreciate the reviewer's concerns and questions. We analyzed gene distributions between tumors with metastasis and tumors without metastasis. Based on gene transcriptomics in the Group 3 subgroup of human medulloblastoma, 1,937 genes are statistically significantly higher ($P < 0.05$, $\log_2(\text{fold change}) > 0$) in metastatic tumors and 1,126 genes are statistically significantly lower ($P < 0.05$, $\log_2(\text{fold change}) < 0$) in metastatic tumors. SMARCD3 is in the top 7.331% of the 1,937 genes according to $\log_2(\text{fold change})$ (**Rebuttal Fig. 4**). Based on gene transcriptomics in all subgroups of human medulloblastoma, 3,984 genes are statistically significantly higher ($P < 0.05$, $\log_2(\text{fold change}) > 0$) in metastatic tumors and 3,110 genes are statistically significantly lower ($P < 0.05$, $\log_2(\text{fold change}) < 0$) in metastatic tumors. SMARCD3 is on the top 8.584% of the 3,984 genes according to the $\log_2(\text{fold change})$ (**Rebuttal Fig. 4**).

Rebuttal Fig 4: The distributions of SMARCD3 gene expression in patient MBs with/without metastasis. Histograms showing the number of differentially expressed genes between patients with metastasis and without metastasis by the $\log_2(\text{fold change})$. The arrows denote where SMARCD3 is located. The data from Group 3 only (right) and all subgroups of human MBs (left) were analyzed.

It would be important to know whether SMARCD3 KO/lower expressing cells have a general problem with cell viability (e.g., decreased proliferation rate), or whether the effect of low/no expression is specific for cell invasion/migration. The latter would be more compelling when speaking of SMARCD3 as a factor for metastasis.

Response: We thank the reviewer for these points. To examine whether SMARCD3 influences cell proliferation or viability, we first performed a BrdU assay in MB cell line MED8A with SMARCD3 deletion or overexpression. We found no significant differences in cell proliferation when SMARCD3 was deleted by CRISPR/Cas9-mediated gene knockout or overexpressed SMARCD3 in MED8A cells (**Rebuttal Fig. 5a**). Second, we performed an MTS assay to examine cell viability in two MB cell lines. The results showed no significant changes in cell viability after SMARCD3 deletion in MED8A and D458 cells during cell growth (**Rebuttal Fig. 5b, c**). These additional data suggest that SMARCD3 plays a critical role in cell migration and tumor metastasis, rather than cell proliferation.

Rebuttal Fig 5: No significance of SMARCD3 effect on MB cell proliferation and viability. (a) Representative

immunofluorescence images and quantification of BrdU (red) of MED8A cells with SMARCD3 WT, KO, and overexpression (OE) in the BrdU assays. P values were calculated using one-way ANOVA followed by Dunnett's multiple comparison test. (b, c) Quantification by the MTS assay of the proliferation of MED8A and D458 cells with SMARCD3 WT vs KO. Data are presented as mean \pm SD.

Minor: the first sentence of the abstract ("how dysregulation...remains elusive") reads a little bold, when considering the high number of neurodevelopmental genes that have been investigated in medulloblastoma. Nevertheless, the authors are certainly correct that a study covering both aspects in depths (and providing new findings for both, tumorigenesis and neurodevelopment at the same time) is very rare in the field. Maybe the authors want to consider to stress this point more.

Response: We thank the reviewer's thoughtful concern and suggestion. The sentence of the abstract was altered, and we tried to stress the point of how abnormal neurodevelopment relates to tumor aggressiveness in medulloblastoma (**Page 3, Line 48-49**).

Minor 2: A better graphical summary/schematic drawing could make the complex story better digestible for most readers.

Response: We agree and thank the reviewer for this suggestion. A graphical summary was included in the revised manuscript (**Revised Extended Data Fig. 7e; Page 17, Line 415-418**).

Revised Extended Data Fig. 7e: SMARCD3-mediated epigenomic programs in neurodevelopment and medulloblastoma metastasis. The schematic diagram shows that SMARCD3 plays a central role in cerebellar development and medulloblastoma metastatic dissemination by regulating the Reelin/DAB1/Src signaling at the molecular, cellular, and tissue/organ levels. SMARCD3 transcription regulation is mediated by chromatin hubs during cerebellar development and medulloblastoma aggressiveness. Targeting SMARCD3/Reelin/DAB1/Src signaling provides a potential novel antimetastatic therapy for patients with medulloblastoma.

References:

- 1 Northcott, P. A. *et al.* Medulloblastomics: the end of the beginning. *Nat Rev Cancer* **12**, 818-834, doi:10.1038/nrc3410 (2012).
- 2 Lin, C. Y. *et al.* Active medulloblastoma enhancers reveal subgroup-specific cellular origins. *Nature* **530**, 57-62, doi:10.1038/nature16546 (2016).
- 3 Del Rio, J. A. *et al.* A role for Cajal-Retzius cells and reelin in the development of hippocampal connections. *Nature* **385**, 70-74, doi:10.1038/385070a0 (1997).

- 4 Franco, S. J., Martinez-Garay, I., Gil-Sanz, C., Harkins-Perry, S. R. & Muller, U. Reelin regulates cadherin function via Dab1/Rap1 to control neuronal migration and lamination in the neocortex. *Neuron* **69**, 482-497, doi:10.1016/j.neuron.2011.01.003 (2011).
- 5 Herz, J. & Chen, Y. Reelin, lipoprotein receptors and synaptic plasticity. *Nat Rev Neurosci* **7**, 850-859, doi:10.1038/nrn2009 (2006).
- 6 Sanada, K., Gupta, A. & Tsai, L. H. Disabled-1-regulated adhesion of migrating neurons to radial glial fiber contributes to neuronal positioning during early corticogenesis. *Neuron* **42**, 197-211, doi:10.1016/s0896-6273(04)00222-3 (2004).
- 7 Wu, J. S. *et al.* Plasticity of cancer cell invasion: Patterns and mechanisms. *Transl Oncol* **14**, 100899, doi:10.1016/j.tranon.2020.100899 (2021).
- 8 Krakhmal, N. V., Zavyalova, M. V., Denisov, E. V., Vtorushin, S. V. & Perelmuter, V. M. Cancer Invasion: Patterns and Mechanisms. *Acta Naturae* **7**, 17-28 (2015).
- 9 Tao, R. *et al.* MYC Drives Group 3 Medulloblastoma through Transformation of Sox2(+) Astrocyte Progenitor Cells. *Cancer Res* **79**, 1967-1980, doi:10.1158/0008-5472.CAN-18-1787 (2019).
- 10 Hovestadt, V. *et al.* Resolving medulloblastoma cellular architecture by single-cell genomics. *Nature* **572**, 74-79, doi:10.1038/s41586-019-1434-6 (2019).
- 11 Vladoiu, M. C. *et al.* Childhood cerebellar tumours mirror conserved fetal transcriptional programs. *Nature* **572**, 67-73, doi:10.1038/s41586-019-1158-7 (2019).
- 12 Hu, B. *et al.* Epigenetic Activation of WNT5A Drives Glioblastoma Stem Cell Differentiation and Invasive Growth. *Cell* **167**, 1281-1295 e1218, doi:10.1016/j.cell.2016.10.039 (2016).
- 13 Hanaford, A. R. *et al.* DiSCoVERing Innovative Therapies for Rare Tumors: Combining Genetically Accurate Disease Models with In Silico Analysis to Identify Novel Therapeutic Targets. *Clin Cancer Res* **22**, 3903-3914, doi:10.1158/1078-0432.CCR-15-3011 (2016).

Decision Letter, first revision:**Date:** 26th July 22 18:42:14**Last Sent:** 26th July 22 18:42:14**Triggered By:** Zhe Wang**From:** zhe.wang@nature.com**To:** baoli.hu@pitt.edu**CC:** edmund.irwin@nature.com**Subject:** Decision on Nature Cell Biology submission NCB-A47467A**Message:** *Please delete the link to your author homepage if you wish to forward this email to co-authors.

Dear Dr Hu,

Your manuscript, "Hijacking a neurodevelopmental epigenomic program in metastatic

dissemination of medulloblastoma", has now been seen by our original referees. As you will see from their comments (attached below) they find this work of interest, but have raised some important points. Although we are also very interested in this study, we believe that their concerns should be addressed before we can consider publication in Nature Cell Biology.

Nature Cell Biology editors discuss the referee reports in detail within the editorial team, including the chief editor, to identify key referee points that should be addressed with priority, and requests that are overruled as being beyond the scope of the current study. To guide the scope of the revisions, I have listed these points below. We are committed to providing a fair and constructive peer-review process, so please feel free to contact me if you would like to discuss any of the referee comments further.

In particular, it would be essential to:

A) Address the remaining concerns from Reviewer 2;

B) Finally please pay close attention to our guidelines on statistical and methodological reporting (listed below) as failure to do so may delay the reconsideration of the revised manuscript. In particular please provide:

We therefore invite you to take these points into account when revising the manuscript. In addition, when preparing the revision please:

- ensure that it conforms to our format instructions and publication policies (see below and www.nature.com/nature/authors/).

- provide a point-by-point rebuttal to the full referee reports verbatim, as provided at the end of this letter.

- provide the completed Editorial Policy Checklist (found here <https://www.nature.com/authors/policies/Policy.pdf>), and Reporting Summary (found here <https://www.nature.com/authors/policies/ReportingSummary.pdf>). This is essential for reconsideration of the manuscript and these documents will be available to editors and referees in the event of peer review. For more information see <http://www.nature.com/authors/policies/availability.html> or contact me.

Nature Cell Biology is committed to improving transparency in authorship. As part of our efforts in this direction, we are now requesting that all authors identified as

'corresponding author' on published papers create and link their Open Researcher and Contributor Identifier (ORCID) with their account on the Manuscript Tracking System (MTS), prior to acceptance. ORCID helps the scientific community achieve unambiguous attribution of all scholarly contributions. You can create and link your ORCID from the home page of the MTS by clicking on 'Modify my Springer Nature account'. For more information please visit please visit www.springernature.com/orcid.

[REDACTED]

We would like to receive the revision within four weeks. If submitted within this time period, reconsideration of the revised manuscript will not be affected by related studies published elsewhere, or accepted for publication in Nature Cell Biology in the meantime. We would be happy to consider a revision even after this timeframe, but in that case we will consider the published literature at the time of resubmission when assessing the file.

We hope that you will find our referees' comments, and editorial guidance helpful. Please do not hesitate to contact me if there is anything you would like to discuss.

Best wishes,
Zhe Wang

Zhe Wang, PhD
Senior Editor
Nature Cell Biology

Tel: +44 (0) 207 843 4924
email: zhe.wang@nature.com

Reviewers' Comments:

Reviewer #2:

Remarks to the Author:

The revision provides only partially satisfactory improvements over the first submission. However, the survival analyses re. SMARCD3 RNA expression and IHC should either be completely removed or the group3 specific analyses should be moved into the main manuscript in order to retransparently provide important information needed to fully interpret the data. Same holds true for the DAB1 mRNA expression correlations. A conclusion on the public enhancer datasets (and how supportive this is (or not)) was not provided in the revised manuscript.

Reviewer #3:

Remarks to the Author:

The authors have responded well and convincingly to my questions and criticisms.

I would strongly suggest to include the additional and important data that were generated in response to my points into the manuscript - and not to keep it as reviewer-only figures. This applies to Rebuttal Fig. 2-5. All this data should be integrated (as Extended Data Figure panels) into the manuscript, and briefly reported and discussed there. It is not clear why this has not happened. There should be three additional Ext Data Figures possible (current number: 7).

GUIDELINES FOR SUBMISSION OF NATURE CELL BIOLOGY ARTICLES

ARTICLE FORMAT

ABSTRACT – should not exceed 150 words and should be unreferenced. This paragraph is the most visible part of the paper and should briefly outline the background and rationale for the work, and accurately summarize the main results and conclusions. Key genes, proteins and organisms should be specified to ensure discoverability of the paper in online searches.

TEXT – the main text consists of the Introduction, Results, and Discussion sections

and must not exceed 3500 words including the abstract. The Introduction should expand on the background relating to the work. The Results should be divided in subsections with subheadings, and should provide a concise and accurate description of the experimental findings. The Discussion should expand on the findings and their implications. All relevant primary literature should be cited, in particular when discussing the background and specific findings.

REFERENCES – are limited to a total of 70 in the main text and Methods combined,. They must be numbered sequentially as they appear in the main text, tables and figure legends and Methods and must follow the precise style of Nature Cell Biology references. References only cited in the Methods should be numbered consecutively following the last reference cited in the main text. References only associated with Supplementary Information (e.g. in supplementary legends) do not count toward the total reference limit and do not need to be cited in numerical continuity with references in the main text. Only published papers can be cited, and each publication cited should be included in the numbered reference list, which should include the manuscript titles. Footnotes are not permitted.

Methods should be written concisely, but should contain all elements necessary to allow interpretation and replication of the results. As a guideline, Methods sections typically do not exceed 3,000 words. The Methods should be divided into subsections listing reagents and techniques. When citing previous methods, accurate references should be provided and any alterations should be noted. Information must be provided about: antibody dilutions, company names, catalogue numbers and clone numbers for monoclonal antibodies; sequences of RNAi and cDNA probes/primers or company names and catalogue numbers if reagents are commercial; cell line names, sources and information on cell line identity and authentication. Animal studies and experiments involving human subjects must be reported in detail, identifying the committees approving the protocols. For studies involving human subjects/samples, a statement must be included confirming that informed consent was obtained.

Statistical analyses and information on the reproducibility of experimental results should be provided in a section titled "Statistics and Reproducibility".

All Nature Cell Biology manuscripts submitted on or after March 21 2016, must include a Data availability statement as a separate section after Methods but before references, under the heading "Data Availability". For Springer Nature policies on data availability see <http://www.nature.com/authors/policies/availability.html>; for more information on this particular policy see <http://www.nature.com/authors/policies/data/data-availability-statements-data-citations.pdf>. The Data availability statement should include:

- Accession codes for primary datasets (generated during the study under consideration and designated as "primary accessions") and secondary datasets (published datasets reanalysed during the study under consideration, designated as "referenced accessions"). For primary accessions data should be made public to coincide with publication of the manuscript. A list of data types for which submission to community-endorsed public repositories is mandated (including sequence, structure, microarray, deep sequencing data) can be found here <http://www.nature.com/authors/policies/availability.html#data>.
- Unique identifiers (accession codes, DOIs or other unique persistent identifier) and hyperlinks for datasets deposited in an approved repository, but for which data deposition is not mandated (see here for details <http://www.nature.com/sdata/data-policies/repositories>).
- At a minimum, please include a statement confirming that all relevant data are available from the authors, and/or are included with the manuscript (e.g. as source data or supplementary information), listing which data are included (e.g. by figure panels and data types) and mentioning any restrictions on availability.
- If a dataset has a Digital Object Identifier (DOI) as its unique identifier, we strongly encourage including this in the Reference list and citing the dataset in the Methods.

We recommend that you upload the step-by-step protocols used in this manuscript to the Protocol Exchange. More details can found at www.nature.com/protocolexchange/about.

DISPLAY ITEMS – main display items are limited to 6-8 main figures and/or main tables. For Supplementary Information see below.

FIGURES – Colour figure publication costs \$395 per colour figure. All panels of a multi-panel figure must be logically connected and arranged as they would appear in the final version. Unnecessary figures and figure panels should be avoided (e.g. data presented in small tables could be stated briefly in the text instead).

All imaging data should be accompanied by scale bars, which should be defined in the legend.

Cropped images of gels/blots are acceptable, but need to be accompanied by size markers, and to retain visible background signal within the linear range (i.e. should not be saturated). The boundaries of panels with low background have to be

demarked with black lines. Splicing of panels should only be considered if unavoidable, and must be clearly marked on the figure, and noted in the legend with a statement on whether the samples were obtained and processed simultaneously. Quantitative comparisons between samples on different gels/blots are discouraged; if this is unavoidable, it has to be performed for samples derived from the same experiment with gels/blots were processed in parallel, which needs to be stated in the legend.

- For line art, graphs, charts and schematics we prefer Adobe Illustrator (.AI), Encapsulated PostScript (.EPS) or Portable Document Format (.PDF). Files should be saved or exported as such directly from the application in which they were made, to allow us to restyle them according to our journal house style.
- We accept PowerPoint (.PPT) files if they are fully editable. However, please refrain from adding PowerPoint graphical effects to objects, as this results in them outputting poor quality raster art. Text used for PowerPoint figures should be Helvetica (preferred) or Arial.
- We do not recommend using Adobe Photoshop for designing figures, but we can accept Photoshop generated (.PSD or .TIFF) files only if each element included in the figure (text, labels, pictures, graphs, arrows and scale bars) are on separate layers. All text should be editable in 'type layers' and line-art such as graphs and other simple schematics should be preserved and embedded within 'vector smart objects' - not flattened raster/bitmap graphics.
- Some programs can generate Postscript by 'printing to file' (found in the Print dialogue). If using an application not listed above, save the file in PostScript format or email our Art Editor, Allen Beattie for advice (a.beattie@nature.com).

Regardless of format, all figures must be vector graphic compatible files, not supplied in a flattened raster/bitmap graphics format, but should be fully editable, allowing us to highlight/copy/paste all text and move individual parts of the figures (i.e. arrows, lines, x and y axes, graphs, tick marks, scale bars etc). The only parts of the figure that should be in pixel raster/bitmap format are photographic images or 3D rendered graphics/complex technical illustrations.

Unprocessed scans of all key data generated through electrophoretic separation techniques need to be presented in a supplementary figure that should be labeled and numbered as the final supplementary figure, and should be mentioned in every relevant figure legend. This figure does not count towards the total number of figures and is the only figure that can be displayed over multiple pages, but should be provided as a single file, in PDF or TIFF format. Data in this figure can be displayed in a relatively informal style, but size markers and the figures panels corresponding to the presented data must be indicated.

The total number of Supplementary Figures (not including the “unprocessed scans” Supplementary Figure) should not exceed the number of main display items (figures and/or tables (see our Guide to Authors and March 2012 editorial <http://www.nature.com/ncb/authors/submit/index.html#suppinfo>; <http://www.nature.com/ncb/journal/v14/n3/index.html#ed>). No restrictions apply to Supplementary Tables or Videos, but we advise authors to be selective in including supplemental data.

Each Supplementary Figure should be provided as a single page and as an individual

file in one of our accepted figure formats and should be presented according to our figure guidelines (see above). Supplementary Tables should be provided as individual Excel files. Supplementary Videos should be provided as .avi or .mov files up to 50 MB in size. Supplementary Figures, Tables and Videos must be accompanied by a separate Word document including titles and legends.

GUIDELINES FOR EXPERIMENTAL AND STATISTICAL REPORTING

REPORTING REQUIREMENTS – To improve the quality of methods and statistics reporting in our papers we have recently revised the reporting checklist we introduced in 2013. We are now asking all life sciences authors to complete two items: an Editorial Policy Checklist (found here <https://www.nature.com/authors/policies/Policy.pdf>) that verifies compliance with all required editorial policies and a Reporting Summary (found here <https://www.nature.com/authors/policies/ReportingSummary.pdf>) that collects information on experimental design and reagents. These documents are available to referees to aid the evaluation of the manuscript. Please note that these forms are dynamic 'smart pdfs' and must therefore be downloaded and completed in Adobe Reader. We will then flatten them for ease of use by the reviewers. If you would like to reference the guidance text as you complete the template, please access these flattened versions at <http://www.nature.com/authors/policies/availability.html>.

Author Rebuttal, first revision:

The authors thank the reviewers for their comments on the revision of our manuscript (# NCB-A47467) and have addressed these comments as follows:

Reviewer #2:

The revision provides only partially satisfactory improvements over the first submission. However, the survival analyses re. SMARCD3 RNA expression and IHC should either be completely removed or the group3 specific analyses should be moved into the main manuscript in order to retransparently provide important information needed to fully interpret the data. Same holds true for the DAB1 mRNA expression correlations.

Response: We appreciate the reviewer for the evaluation of our revision and for making helpful comments to improve the manuscript. Following the reviewer's suggestion, we added additional data examining SMARCD3 mRNA and protein expression using gene profiling and IHC analysis of the G3 medulloblastoma (MB) subgroup (**Revised Fig. 1 f and g**). To interpret the weak correlation between patient survival and SMARCD3 expression levels in the G3 subgroup, we analyzed the relative dispersion of SMARCD3 mRNA expression levels among patient MB samples. We found that the levels of SMARCD3 mRNA expression are higher but with smaller variation in each sample within the G3 cohort compared with the entire MB subgroup cohorts (4.289 in all MB samples vs 0.433 in the G3) (**Revised Extended Data Fig.1d**). Statistically, a small variation in the characteristic of samples results in a low correlation coefficient, which might explain our observation of the weak correlations between SMARCD3 and patient survival in the G3 subgroup.

Additionally, we believe that the hypothesis and conclusion in this manuscript, that SMARCD3 regulates tumor metastatic dissemination, can be applied to other MB subgroups, not to the G3 subgroup exclusively. Given that G3 is the most aggressive subgroup with strong metastatic potential compared with other MB subgroups, the G3 subgroup becomes the most significant model to examine the association and regulatory relationship between SMARCD3 and tumor metastatic dissemination. Therefore, besides analysis in the G3 subgroup exclusively, we also used the datasets across all the MB subgroups when we analyzed the correlation between SMARCD3 and other related factors, such as patient outcomes, metastasis, DAB1 expression, and so on. Consistently, the levels of DAB1 mRNA expression are also higher but with smaller variation in each sample within the G3 cohort compared with all MB subgroups (10.422 in all MB samples vs 0.459 in the G3) (**Revised Extended Data Fig.4c**).

Therefore, we performed correlation analyses between SMARCD3 and DAB1 using the data from all the MB subgroups.

We added additional information about the data interpretation in the revised text including the Results and Discussion sections (**Page 4, Line 92-93; Page 5, Line 104-114; Page 10, Line 226-230; Page 20, Line 467-471**).

A conclusion on the public enhancer datasets (and how supportive this is (or not)) was not provided in the revised manuscript.

Response: We apologize for misunderstanding the comments here. To substantiate the finding of newly-identified cis-regulatory elements (CREs) at the *SMARCD3* gene locus, we firstly assessed H3K27ac histone marker levels for each tumor sample rather than pooling them together in each MB subgroup (**Revised Extended Data Fig.7c**). Second, we examined the mapping of these CREs at the *SMARCD3* gene locus in human and mouse genome using the public enhancer dataset, ENCODE and Roadmap (**Revised Extended Data Fig.7d, e**). Consistently, these results further support the newly-identified CREs of *SMARCD3* in our study.

Reviewer #3:

The authors have responded well and convincingly to my questions and criticisms.

I would strongly suggest to include the additional and important data that were generated in response to my points into

the manuscript - and not to keep it as reviewer-only figures. This applies to Rebuttal Fig. 2-5. All this data should be integrated (as Extended Data Figure panels) into the manuscript, and briefly reported and discussed there. It is not clear why this has not happened. There should be three additional Ext Data Figures possible (current number: 7).

Response: We thank the reviewer for the evaluation and appreciation of our revision. We also thank the reviewer's suggestions, which are helpful to improve the quality of the manuscript. **[REDACTED]** Therefore, we integrated all the Rebuttal figures into the revised manuscript and briefly described these data in the revised manuscript. The revision can be found in **Revised Extended Data Fig. 1g; Revised Extended Data Fig.3a, b, and e-m; Page 6, Line 124-129; Page 7, Line 164-167; Page 8-9, Line 179-203**

Decision Letter, second revision:

Date: 18th September 22 18:51:59
Last Sent: 18th September 22 18:51:59
Triggered By: Zhe Wang
From: zhe.wang@nature.com
To: baoli.hu@pitt.edu
CC: ziqian.li@nature.com,ncb@springernature.com
Subject: Your manuscript, NCB-A47467B
Message: Our ref: NCB-A47467B

18th September 2022

Dear Dr. Hu,

Thank you for submitting your revised manuscript "Hijacking a neurodevelopmental epigenomic program in metastatic dissemination of medulloblastoma" (NCB-A47467B). It has now been seen by the original referees and their comments are below. The reviewers find that the paper has improved in revision, and therefore we'll be happy in principle to publish it in Nature Cell Biology, pending minor revisions to comply with our editorial and formatting guidelines.

Thank you again for your interest in Nature Cell Biology Please do not hesitate to contact me if you have any questions.

Sincerely,

Zhe Wang, PhD
Senior Editor
Nature Cell Biology

Tel: +44 (0) 207 843 4924
email: zhe.wang@nature.com

Reviewer #2 (Remarks to the Author):

The authors have now addressed my remaining issues satisfactorily.

Final Decision Letter:

Dear Dr Hu,

I am pleased to inform you that your manuscript, "A neurodevelopmental epigenetic program mediated by SMARCD3-DAB1-Reelin signaling is hijacked to promote metastatic dissemination of medulloblastoma", has now been accepted for publication in Nature Cell Biology.

Please note that Nature Cell Biology is a Transformative Journal (TJ). Authors may publish their research with us through the traditional subscription access route or make their paper immediately open access through payment of an article-processing charge (APC). Authors will not be required to make a final decision about access to their article until it has been accepted. Find out more about Transformative Journals

Authors may need to take specific actions to achieve compliance with funder and institutional open access mandates. If your research is supported by a funder that requires immediate open access (e.g. according to Plan S principles) then you should select the gold OA route, and we will direct you to the compliant route where possible. For authors selecting the subscription publication route, the journal's standard licensing terms will need to be accepted, including self-archiving policies. Those licensing terms will supersede any other terms that the author or any third party may assert apply to any version of the manuscript.

If you have not already done so, we strongly recommend that you upload the step-by-step protocols used in this manuscript to the Protocol Exchange (www.nature.com/protocolexchange), an open online resource established by Nature Protocols that allows researchers to share their detailed experimental know-how. All uploaded protocols are made freely available, assigned DOIs for ease of citation and are fully searchable through nature.com. Protocols and Nature Portfolio journal papers in which they are used can be linked to one another, and this link is clearly and prominently visible in the online versions of both papers. Authors who performed the specific experiments can act as primary authors for the Protocol as they will be best placed to share the methodology details, but the Corresponding Author of the present research paper should be included as one of the authors. By uploading your Protocols to Protocol Exchange, you are enabling researchers to more readily reproduce or adapt the methodology you use, as well as increasing the visibility of your protocols and papers. You can also establish a dedicated page to collect your lab Protocols. Further information can be found at www.nature.com/protocolexchange/about

With kind regards,

Zhe Wang, PhD
Senior Editor
Nature Cell Biology

Tel: +44 (0) 207 843 4924
email: zhe.wang@nature.com
